# PULSE: Benchmarking Large Language Models for ICU Time Series Classification

## Abstract

Large language models (LLMs) are increasingly used on multimodal clinical data, yet their performance on high-stakes intensive care unit (ICU) time series data remains under-characterized. We introduce PULSE, a comprehensive benchmark evaluating 17 models, including conventional learners, deep learning, and instruction-following LLMs, across three datasets (HiRID, MIMIC-IV, eICU) and three clinical endpoints (mortality, sepsis, acute kidney injury). In standard within-domain settings, we find that Gradient Boosted Decision Trees (LightGBM) remain the state-of-the-art, achieving mean AUROCs up to 0.916. Frontier LLMs come close (best mean AUROC of 0.893, OpenAI o3), but show sensitivity to the prompting technique. Crucially, while conventional machine learning and deep learning models suffer performance degradation when tested in unseen domains (e.g., XGBoost AUROC dropping to $\approx 0.511$, when trained in MIMIC IV and tested in eICU) due to distribution shift, zero shot and few shot prompting and hybrid reasoning LLM workflow demonstrate robust performance. This establishes LLMs not merely as reasoning engines, but as the pragmatic "day-zero" solution for institutions lacking the labeled data required to train conventional models. PULSE provides all code, configuration files, and a public results dashboard to enable transparent, reproducible comparison and rapid community extension. We expect PULSE to serve as a common yardstick in the years to come, for developing reliable LLMs for multimodal time series data in critical care.

## 1 Introduction

The intensive care unit (ICU) represents one of the most data-rich environments in modern medicine, where structured time-series data from physiological monitors and laboratory tests are paramount for clinical decision-making. These data power hundreds of machine learning studies aimed at early warning, risk prediction, and outcome modeling (Johnson et al., 2023). The clinical stakes are exceptionally high, as conditions such as sepsis, acute kidney injury (AKI), and in-hospital mortality carry a substantial global burden. Sepsis contributes to millions of deaths annually (Rudd et al., 2020), often precipitates AKI (Zarbock et al., 2023), and ICU mortality rates can exceed 30% in some cohorts (Melaku et al., 2024). Timely and accurate predictive models can directly improve patient outcomes; for instance, early antibiotic administration significantly reduces sepsis mortality (Evans et al., 2021). However, ICU time-series data present formidable technical challenges, including irregular sampling intervals, high dimensionality and sparsity, pervasive informative missingness, and lack of model generalization due to distribution shift, which complicate model development and deployment (Moor et al., 2021; Tranchellini et al., 2025) .

The rapid proliferation of large language models (LLMs) has transformed many areas of healthcare, particularly those centered on clinical text and dialogue (Naveed et al., 2023). Their demonstrated ability to reason over complex sequences has raised a pivotal question: can the advanced sequential reasoning capabilities of models like `OpenAI-o3`, `Gemini-2.5-Pro`, and `Claude-Sonnet-4` (Achiam et al., 2023) be effectively translated to the non-linguistic, noisy, and highly structured domain of ICU time series? Answering this question is impeded by a significant gap in the evaluation landscape. Existing healthcare LLM benchmarks, such as Health-Bench (Arora et al., 2025) and MedHELM (Bedi et al., 2025b), are designed for language-centric tasks like question-answering and summarization. Conversely, established time-series benchmark-

ing frameworks like Yet Another ICU Benchmark (Van De Water et al., 2024) provide crucial harmonization for conventional models but do not accommodate the unique input formats and evaluation needs of LLMs. This leaves a critical blind spot in our understanding of where LLMs fit within the clinical time series prediction toolkit.

To address these gaps, this paper introduces **PULSE**, a benchmark designed specifically for classification tasks on ICU time series, representing the first of its kind to systematically compare conventional machine learning, deep learning, and LLM-based paradigms. We implement leakage-resistant temporal splits, and evaluate performance on three high-impact clinical tasks across three large, harmonized, multi-center ICU cohorts. Our evaluation reveals that classical methods like gradient-boosted trees remain the state-of-the-art in terms of raw predictive performance (Grinsztajn et al., 2022b). A novel hybrid workflow enables LLMs to approach this performance. This hybrid workflow leverages the LLM not as a direct numerical predictor, but as a reasoning engine that synthesizes the outputs of a robust conventional model to generate both a final prediction and a clinically relevant, human-readable rationale. Crucially, we also find that LLMs are better than machine learning and deep learning models as "day zero" predictors for out of domain datasets.

Our primary contributions are threefold:

- We design and publicly release **PULSE**, the first standardized benchmark for evaluating and comparing LLMs against conventional machine learning and deep learning models on ICU time series classification tasks.

- We conduct a comprehensive empirical evaluation of 17 models across three machine learning model paradigms (conventional machine learning, deep learning, and LLMs), and for LLMs, five prompting techniques and four workflow-based approaches.

- We perform evaluations on model generalization, clinical utility of LLM explanations, and cost and latency of using LLMs in these complex predictive tasks.

## 2 RELATED WORK

**Machine Learning for ICU Time-Series Prediction.** Early research in clinical medicine has demonstrated that even simple linear models and initial neural networks could surpass clinician accuracy in specific diagnostic tasks (Dawes & Corrigan, 1974; Baxt & Skora, 1996). Hence, the application of machine learning to structured ICU data has a rich history, more recently dominated by two primary paradigms. Conventional machine learning (ConvML) methods, including logistic regression and gradient-boosted decision trees (GBDTs), have long been the mainstay (Garg & Mago, 2021). GBDTs, in particular, remain powerful baselines on tabular data due to their ability to handle heterogeneous features, model non-linear relationships, and inherent robustness to uninformative features without extensive preprocessing (Grinsztajn et al., 2022b;a). However, their effectiveness often relies on significant feature engineering to transform irregular, longitudinal patient trajectories into fixed-size inputs, a process that is labor-intensive and risks introducing bias (Moor et al., 2021). Conventional Deep learning (ConvDL) models, such as Recurrent Neural Networks (RNNs) and Transformers, address this by processing sequential data (Shickel et al., 2018; Tipirneni & Reddy, 2022). While they reduce the need for manual feature engineering, their performance gains over well-tuned GBDT pipelines on tabular time-series data are inconsistent and introduce challenges related to data requirements, computational cost, and tuning complexity (Hyland et al., 2020; Grinsztajn et al., 2022b). More recently, LLMs have emerged as a third paradigm. Early explorations have adapted them to structured data through several strategies: (i) serialization, which flattens time-series data into text strings (Sarvari et al., 2024); (ii) prompt engineering, which uses few-shot examples to guide in-context learning (Liu et al., 2023); and (iii) LLM/agentic workflows, where LLMs orchestrate external tools or conventional models (Zhu et al., 2024a). These approaches are promising but have been developed in an ad-hoc manner, lacking the standardized evaluation needed to compare them to each other or to established baselines fairly. Furthermore, LLMs are susceptible to failure modes like factual hallucination and poor calibration, which are particularly perilous in high-stakes clinical settings (Zhu et al., 2024d).

**The Landscape of Clinical AI Benchmarking.** Progress in clinical AI has been advanced by the development of benchmarks, which fall into two distinct categories. First, there are dataset

specific benchmarks such as HiRID ICU Benchmark (Yèche et al., 2021), which only focus on a single ICU time series dataset, by using only classic and deep learning models. Further, more recent benchmarks like YAIB (Yet Another ICU Benchmark) have made significant strides in harmonizing preprocessing and task definitions across multiple ICU datasets (e.g., MIMIC, eICU, HiRID) for ConvML and ConvDL models (Van De Water et al., 2024). By providing a unified pipeline, YAIB enhances reproducibility and enables fair comparisons within these traditional paradigms. However, its framework was not designed to accommodate the unique architecture and prompting interfaces of LLMs, leaving a critical gap as these models gain prominence. Second, a new generation of LLM-centric healthcare benchmarks has emerged, most notably HealthBench (Arora et al., 2025) and MedHELM (Bedi et al., 2025a). These frameworks provide rigorous evaluation of LLMs on a wide array of clinical language tasks, such as medical question answering, dialogue, and document summarization. They emphasize realistic, open-ended evaluation through expert-defined rubrics. While invaluable for assessing the linguistic and reasoning capabilities of LLMs in medicine, their design, data formats, and metrics are fundamentally unsuited for the quantitative, time-series prediction tasks that are central to real-time ICU monitoring and risk stratification.

**Positioning and Novelty.** The existing literature reveals a clear and consequential gap. While ICU time series benchmarks standardizes evaluation for conventional models on structured ICU data, and HealthBench/MedHELM do so for LLMs on clinical language tasks, no framework exists to rigorously and reproducibly evaluate LLMs on structured, time series prediction tasks against their conventional counterparts, a newly emerging set of challenging tasks for LLMs. PULSE is designed to fill this exact gap, while introducing the necessary infrastructure to support new datasets, LLM-native interfaces, prompting strategies, and diverse evaluation metrics. By doing so, PULSE provides the first unified framework to enable fair, head-to-head comparisons across all three major modeling paradigms, establishing a much needed foundation for future research in this critical domain. In a time when benchmarking is a necessity in domains like biomedicine (Mahmood, 2025), we believe PULSE fills in a crucial gap that would be immensely valuable for both machine learning and clinical communities.

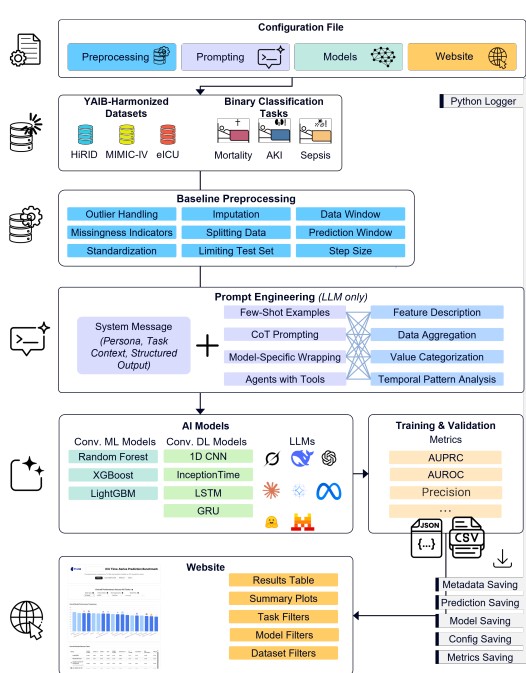

Figure 1: **The PULSE Framework Structure.** Input is formed by the harmonized datasets split into training, validation, and testing sets. The data is then preprocessed in two main modules. A set of baseline-preprocessing steps, which all models run through, and a set of prompt engineering steps, accessible for LLMs (more on time series data in Appendix A.2). The model module hosts all implemented models subject to equal output conditions. Model output is evaluated by the evaluation module, calculating all metrics.

## 3 PULSE BENCHMARK

PULSE is a comprehensive and reproducible benchmark designed to rigorously evaluate and compare modeling paradigms for clinical prediction on ICU time-series data. It allows direct comparison across conventional machine learning (ConvML), conventional deep learning (ConvDL), and large language models (LLMs).

**Design Principles and Architecture.** An overview of the PULSE framework is illustrated in Figure 1. The design of PULSE is guided by five core principles derived from best practices in machine learning research (Longjohn et al., 2024; Cao et al., 2025; Liu et al., 2024). **Coverage:** The benchmark includes multiple datasets and tasks with clinically valid label definitions to ensure that findings are

robust and not artifacts of a single cohort or outcome. The coverage can be easily extended with new datasets, prompts, workflows/agents, and endpoints. **Standardization:** A shared data harmonization pipeline and a consistent pre-processing and splitting policy are enforced to eliminate confounding variables from experimental setup (Van De Water et al., 2024). **Paradigm-Agnosticism:** A single evaluation pipeline with uniform inputs and outputs is used for all models, preventing implicit bias towards any single model family. **Reproducibility:** The entire workflow is configuration-driven, with versioned artifacts, deterministic seeding, and comprehensive caching to ensure that every experiment can be precisely replicated. **Transparency:** All metrics, predictions, configurations, and metadata are made publicly available through a dedicated website (section 6) to support auditability and community extension.

### 3.1 Scope: Tasks and Datasets

PULSE incorporates three of the largest and most widely used public ICU datasets to ensure findings are generalizable across different healthcare systems, populations, and data-recording technologies. However, PULSE provides the framework to extend this to any other datasets as well. The chosen datasets offer complementary strengths: **HiRID** (Faltys et al.; Hyland et al., 2020) provides high temporal resolution (measurements every 2-5 minutes) from a European cohort; **MIMIC-IV** (Johnson et al., 2023) offers deep, longitudinal data from a single academic medical center in the USA over a long period; and **eICU** (Pollard et al., 2017; 2018) provides multi-center diversity from over 200 US hospitals. All datasets are accessed via the credentialed PhysioNet platform and undergo extensive, standardized preprocessing in line with Van De Water et al. (2024), which harmonizes variable definitions, units, and data structures. This process yields static demographic features and 48 dynamic variables resampled to an hourly frequency. ICU stays are capped at seven days to standardize input length. This rigorous, multi-stage processing significantly reduces the risk of data contamination from public sources. Further details on cohort selection and dataset characteristics are provided in Table 1 and Figure 11.

**Tasks.** The benchmark evaluates three clinically critical binary prediction tasks, with labels defined according to established clinical guidelines to ensure relevance and consistency. **In-Hospital Mortality:** Prediction of death during the index ICU admission. A single prediction is made for each stay using data from the first 24 hours. Labels are derived directly from hospital discharge status, avoiding reliance on scoring systems like APACHE II, which can underestimate mortality in specific cohorts like septic shock (Knaus et al., 1985; Bloria et al., 2023). **Acute Kidney Injury (AKI):** Early prediction of AKI, defined as Kidney Disease: Improving Global Outcomes (KDIGO) stage $\geq 1$ based on changes in serum creatinine or urine output. The specific criteria are detailed in Table 7. The task uses a 6-hour observation window to predict AKI onset within the subsequent 6-hour horizon. **Sepsis:** Early prediction of sepsis onset, defined according to the Sepsis-3 criteria, which require evidence of organ dysfunction (a change in the Sequential Organ Failure Assessment score of $\Delta \geq 2$) temporally associated with a suspected infection (Seymour et al., 2016). The task uses a 6-hour observation window to predict sepsis onset within the subsequent 6-hour horizon.

### 3.2 Standardized Preprocessing and Data Handling

A standardized preprocessing pipeline is applied to all datasets to ensure fair comparisons. The pipeline includes clinically informed outlier handling, where values outside plausible physiological ranges are set to NaN. For each dynamic variable, an explicit binary missingness indicator is created, as the pattern of missingness itself can be predictive in clinical settings (Van Ness et al., 2023; Ghosheh et al., 2024). Time-series data within each stay are forward-filled to handle intermittent measurements. Any remaining missing values are imputed using train-set statistics (e.g., median) to strictly prevent data leakage from the validation and test sets (Salinas et al., 2025). For data splitting, a temporal 8:1:1 train-validation-test split is enforced, where patients are allocated based on ascending admission times. This chronological splitting is crucial for preventing leakage from future data into the training set and provides a more realistic estimate of prospective performance (Lones, 2021). For ConvDL models, features are standardized to zero mean and unit variance using parameters derived solely from the training set. ConvML and LLM models receive non-standardized data to preserve the clinical scale and interpretability of the raw values.

3.3 A Multi-Paradigm Model Zoo

PULSE evaluates a diverse set of 17 models spanning three distinct paradigms. **Conventional Machine Learning (ConvML).** We benchmark Random Forest, XGBoost, and LightGBM. These decision tree models are chosen as they represent the de facto state-of-the-art for many tabular data tasks and are known for their high performance, efficiency, and robustness, making them a challenging and realistic baseline (Hyland et al., 2020; Grinsztajn et al., 2022b; Tang et al., 2025; Grinsztajn et al., 2022a). Hyperparameter tuning was done using the validation set. **Conventional Deep Learning (ConvDL).** We include four architectures to represent different approaches to sequence modeling: a 1D-CNN for capturing local temporal patterns, InceptionTime for multi-scale feature extraction (Ismail Fawaz et al., 2020), and LSTM and GRU networks for modeling long-range temporal dependencies. All models are trained under a unified regimen with Adam optimization, early stopping, and fixed random seeds for determinism. Final configurations are detailed in Table 3. **Large Language Models (LLMs).** We evaluate a broad range of 10 LLMs, including both leading open-source models (Llama 3.1 8B, DeepSeek-R1, Mistral-7B, Gemma-3, MedGemma-4b) and proprietary frontier models accessed via API (OpenAI o3, Gemini 2.5 Pro, Gemini 2.5 Flash, Claude Sonnet 4, Grok 4). This selection covers a wide spectrum of model sizes, architectures, and training data. Inference parameters, such as temperature (0.4) and context length, are standardized across all models to ensure reproducible and comparable generation. Complete details are available in Appendix A.5.

3.4 LLM Prompting and Workflows.

To interface with LLMs, we developed a protocol for translating structured time-series data into text-based prompts and for parsing their structured outputs. **System Message and Structured Output.** A carefully designed system message instructs each LLM to adopt the persona of a clinical expert, defines the specific binary prediction task, and enforces a strict JSON output schema. This schema requires two fields: `"classification"` (a string, e.g., `"sepsis"`) and `"probability"` (an integer from 0 to 100). The use of integer confidences, which are then normalized to probabilities for metric calculation, acknowledges the known calibration limitations of self-reported LLM scores (Tian et al., 2023; de Oliveira et al., 2025). The final system message design was selected based on a systematic ablation study (see Appendix A.11, Figure 14) that balanced performance and instruction-following stability across multiple open-source models. **Prompting Strategies.** PULSE implements and evaluates seven distinct prompting strategies grounded in prior literature, to assess how different methods of representing structured data as text affect LLM performance. These range from simple serialization techniques to complex, multi-step workflows: Aggregation (Sarvari et al., 2024), Zero-Shot (Zhu et al., 2024d), One-Shot, Few-Shot (Liu et al., 2023), and Chain-of-Thought (CoT) (Kojima et al., 2022). **Multi-Step Workflows.** PULSE implements a two-stage Summary Workflow (Zhu et al., 2024c) (see Appendix A.4.2), and a Hybrid Reasoning Workflow (HRW) that integrates predictions from a conventional model (see Appendix A.4.3). In fact, we implemented and experimented four types of workflows (see Appendix A.4), and only present results for two of the best performing ones in the main text, due to length limits. Templates for each strategy are provided in Appendix A.12 and detailed results for all approaches provided in Appendix A.15.

3.5 Evaluation Protocol and Metrics

Our evaluation protocol uses a combination of standard and novel metrics to provide a holistic view of model performance. **Primary Metrics.** We report the Area Under the Receiver Operating Characteristic Curve (AUROC) as the primary metric for discrimination (Fawcett, 2006). We also report the Area Under the Precision-Recall Curve (AUPRC), as it is often considered more informative than AUROC on datasets with significant class imbalance, a common feature of clinical prediction tasks (Saito & Rehmsmeier, 2015; Harutyunyan et al., 2019; Moor et al., 2021). Evaluations are run five times with randomly sampled test subsets due to complexity and costs of conducting LLM based experiments (Appendix A.8). Hence, we also report error bars for performance across these five iterations. Moreover, we conducted evaluations both in-domain (i.e., trained on HiRID training set, and tested on HiRID testing set) and out of domain (i.e., trained on HiRID training set, and tested on eICU testing set). However, out of domain testing only affects ConvML and ConvDL models, as LLM based approaches only use a few examples (Few Shot) or not data at all (Zero Shot) to perform in any domain.

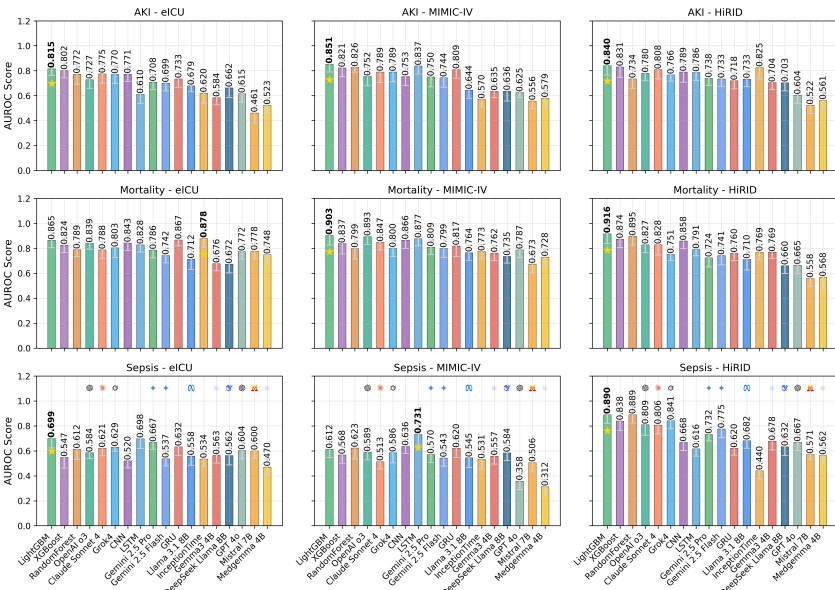

Figure 2: **AUROC scores by task–dataset combination**. Only the best approach per LLM is shown. The best overall model for each combination is **bold** and starred. Models are ordered by overall mean AUROC. For a detailed breakdown of results, see Appendix A.15. See Appendix A.14 for AUPRC.

### 3.6 EVALUATION OF THE UTILITY LLM EXPLANATIONS

To assess the practical value of the HRW, the best performing LLM approach for in-domain experiments, we designed a blinded A/B human evaluation study inspired by recent LLM/agent evaluations Heydari et al. (2025). This was done with five Board-Certified ICU clinicians. The study evaluates whether LLM-generated explanations improve decision-making confidence and workflow efficiency compared to standard model outputs. **Protocol.** We randomly selected 50 complex cases from the test set. Each clinician reviewed 10 cases under two randomized conditions: *Baseline (Condition A):* Clinicians were presented with the patient's physiological time series (vitals/labs) plotted over time and the raw probability score from the XGBoost model; and *Hybrid (Condition B):* Clinicians received the same plots and score, augmented with the natural language explanation generated by the HRW. **Metrics.** To quantify the added value of the LLM, we employed a two-tiered evaluation: **1. Comparative Task Metrics (Rated for both Condition A and B):** *Cognitive Efficiency (Interpretability):* Rate the mental effort required to identify the key drivers of the predicted risk. (1=High Effort, 5=Low Effort); *Decision Confidence:* How confident are you that the model's risk score accurately reflects the patient's status? (1=Low Confidence, 5=High Confidence); *Actionability:* Is the information presented sufficient for an immediate assessment of the patient outcome? (1=Insufficient, 5=Sufficient). Next, **2. Safety Audit (Rated for Condition B only):** *Faithfulness:* Did the text explanation contain any factual hallucinations or inconsistencies with the plotted data? (Binary: Yes/No, followed by an explanation).

## 4 RESULTS

**Overall Performance.** The primary results, summarized by AUROC in Figure 2, reveal a clear performance hierarchy. The overall best-performing model is **LightGBM**, a GBDT implementation, with a mean AUROC of **0.821** across all nine task-dataset combinations. This finding is consistent with a substantial body of literature demonstrating that tree-based ensembles often outperform deep learning models on ICU time series (Hyland et al., 2020; Burger et al., 2025). The inductive biases of tree-based models, such as their ability to create non-linear decision boundaries via axis-aligned splits and their inherent robustness to uninformative features, appear to be exceptionally well-suited to the heterogeneous and noisy nature of structured ICU data, as also discussed in prior work (Moor

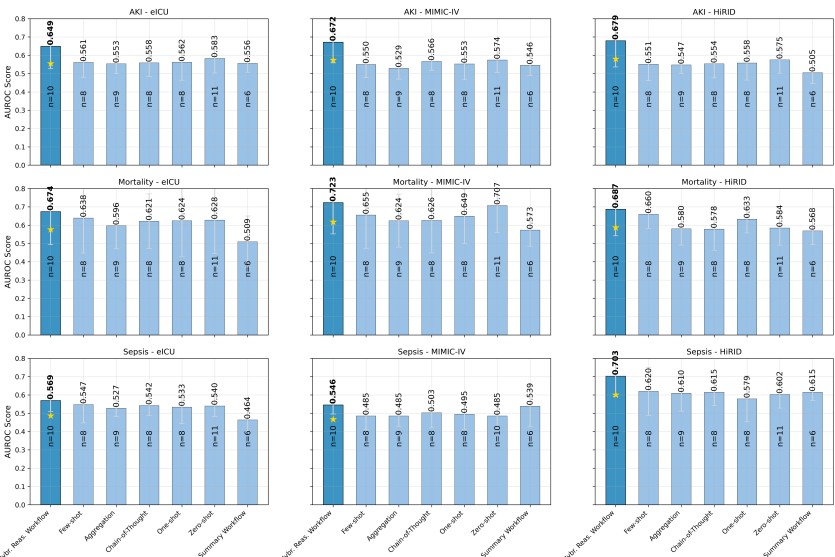

Figure 3: **Mean AUROC per prompting approaches and workflows, averaged over all LLMs.** Each subplot is sorted by score and only labeled for the last three plots. All colors are consistent between the nine subplots and represent a prompting approach. The HRW dominates across settings.

et al., 2021). ConvDL models, while strong performers (LSTM 0.698-0.731 AUROC for Sepsis in MIMIC-IV and eICU; GRU 0.867 AUROC for Mortality in eICU), did not consistently surpass the LightGBM or CNN, reinforcing that increased model complexity does not always guarantee superior performance on this data. Among LLMs, the highest performance is achieved by frontier proprietary models using the HRW strategy. **OpenAI o3** and **Claude-Sonnet-4** are the most competitive, approaching but not surpassing the top ConvML models. While large proprietary models maintain high instruction adherence, smaller LLMs frequently generate outputs where the categorical prediction contradicts the numerical confidence, a direct consequence of poor calibration and a major barrier to trustworthy clinical deployment, that is worth investigations in the future.

**Dissecting Prompting Strategy Performance.** The choice of prompting strategy or using a multi-step workflow has a profound impact on LLM performance. Our analysis, summarized in Figure 3, shows that the HRW is the best performing approach (mean AUROC = 0.691±0.251 across all LLMs), outperforming the next-best strategies (Few Shot and Zero Shot) by a reasonable margin. This result suggests a fundamental insight into the optimal role of LLMs for this data type. Strategies that treat the LLM as a direct predictor (e.g., Zero Shot, Few Shot) force it to perform complex reasoning on raw, serialized numerical sequences, a task for which its architecture may not be inherently optimized. Aggregation-based methods underperform due to significant information loss. In contrast, the HRW reframes the LLM's role. It does not replace the specialized, statistically robust GBDT model; it augments it. The LLM acts as a high-level synthesis engine, integrating the GBDT's prediction and feature importances with the raw patient data to produce a final assessment and a narrative explanation. This approach leverages the respective strengths of each paradigm: the GBDT's superior performance on structured prediction and the LLM's unparalleled ability to reason and generate natural language.

**Qualitative Analysis of LLM Explanations.** A key motivation for exploring LLMs in clinical medicine is their potential to generate human-readable explanations, addressing the "black box" problem that hinders the adoption of many complex models (Rezaeian et al., 2025). Two authors (one of whom has worked as an ICU clinician) qualitatively analyzed the rationales generated by the top-performing HRW across all three tasks (see Appendix A.16, Figure 29, Figure 30, Figure 31). Several key patterns emerged. The rationales are consistently concise and directly reference both static (e.g., age) and dynamic (e.g., diastolic blood pressure, white blood cell count) patient features to justify the prediction. Critically, many explanations explicitly anchor their reasoning to the output

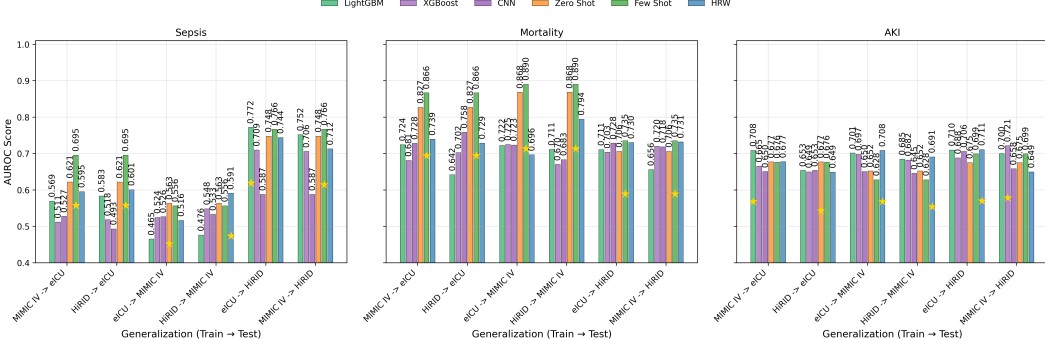

Figure 4: **Cross-Domain Generalization Results (mean AUROC).** Comparison of conventional models (LightGBM, XGBoost, CNN) trained on a source dataset and evaluated on an out-of-domain target dataset (Source → Target), versus off-the-shelf LLM prompting strategies. While supervised models suffer performance collapse due to domain shift (e.g., Sepsis MIMIC-IV → eICU), LLMs maintain robust performance, validating their utility as a "day-zero" solution for hospitals without local training data.

of the conventional model, stating, for example, that a low mortality risk is supported by a "high-confidence XGBoost model". Some models also attempt to ground their reasoning in established medical frameworks, such as referencing the limited availability of SOFA components to justify a "not-sepsis" prediction. This analysis highlights the dual nature of LLM-generated explanations: their plausibility versus their faithfulness (Matton et al., 2025). While the generated text is often clinically plausible, specially with frontier models, its faithfulness, the degree to which it reflects the model's true reasoning process, is a concern. The structure of the Hybrid Workflow provides a partial solution to this problem. By explicitly providing the LLM with the top features identified by the XGBoost model, the resulting explanation is inherently more grounded in the factors that were statistically most important for the prediction. This represents a promising step toward generating explanations that are not only convincing but also a more faithful representation of the underlying predictive logic, a crucial requirement for building clinical trust and ensuring patient safety.

**Clinical Utility of LLM Explanations.** Preliminary results from the user study (described in Section 3.6) indicate that the Hybrid Reasoning Workflow (HRW) enhances the clinical utility of the predictive model. On the cognitive efficiency scale, clinicians rated the HRW at 4.2/5.0 compared to 2.6/5.0 for the Baseline ($p < 0.05$), suggesting that the natural language synthesis accelerates the interpretation of disparate physiological trends. Regarding decision confidence, we observed a moderate but increase from 3.3/5.0 (Baseline) to 3.9/5.0 (Hybrid) ($p > 0.05$). Actionability scores saw the largest improvement, rising from 3.0 to 4.5, as the LLM-generated summaries being deemed beneficial for immediate patient outcome assessment. Finally, the Faithfulness audit revealed a hallucination rate of $\approx 6\%$ (3 cases out of 50), with the explicit inclusion of the XGBoost probability preventing the model from fabricating certainty where the data did not support it. Qualitative feedback indicated that the explanation served as a "sanity check" that helped clinicians trust the XGBoost probability score in ambiguous scenarios, as the explanations were deemed "better than feature importance" as importance values are given at model-level instead of inference level. Consequently, the slight performance cost of the HRW is justified in settings where explainability and clinician trust are important.

**Cross-Domain Generalization and Day-Zero Utility.** To validate the pragmatic "day-zero" utility of LLMs for hospitals lacking labeled data, we evaluated the performance of conventional baselines when trained on a source domain (e.g., MIMIC-IV) and tested on a distinct target domain (e.g., eICU). As shown in Figure , supervised baselines suffer performance degradation due to distribution shifts, a problem well documented in prior work in ICU time series classification (Tranchellini et al., 2025; Guo et al., 2022). For instance, in the Sepsis task, an XGBoost model trained on MIMIC-IV and tested on eICU drops to a mean AUROC of 0.511, performing nearly equivalently to random guessing. Similarly, transferring from HiRID to eICU yields an XGBoost mean AUROC of 0.518.

In contrast, LLM-based approaches, specifically Zero Shot, Few Shot and HRW, demonstrate robust generalization without requiring site-specific parameter updates. In the transfer MIMIC-IV → eICU for Sepsis, the Few-Shot LLM achieves an AUROC of 0.695, surpassing the transferred XGBoost model by over 18%. This trend holds across tasks; for example, in the eICU → HiRID transfer for Sepsis, Few-Shot (0.766) outperforms the CNN baseline (0.587). These results empirically confirm that while conventional models require local retraining to be effective, LLMs offer a superior off-the-shelf baseline for institutions with limited data availability or model training capacity. We believe this is a significant finding, that could warrant further investigations, as model generalization and cross site performance is a crucial aspect with clinical models in ICUs (Tranchellini et al., 2025; Guo et al., 2022).

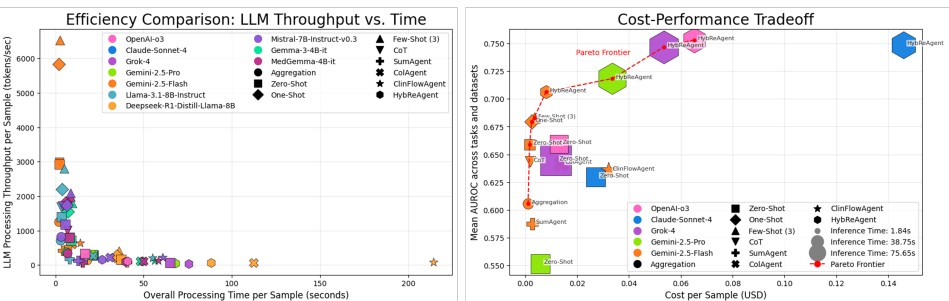

Figure 5: **Efficiency Comparison and Cost-Performance Tradeoff.** This figure analyzes the efficiency and cost-effectiveness of the evaluated LLM approaches. The left panel plots the processing throughput (tokens/sec) against the overall processing time per sample, demonstrating that multi-step workflow methods have lower throughput due to their complex, multi-step nature. The right panel presents a cost-performance tradeoff analysis for proprietary models, plotting the mean AUROC against the cost per sample in USD, with makrer sizes indicating inference time. The Pareto Frontier (red dashed line) highlights optimal cost-performance trade-offs. The analysis reveals that several HRW-based approaches are positioned on the Pareto frontier. See Appendix A.10 for more details.

## 5 DISCUSSION

Our results carry significant implications that extend beyond the ICU.

*(i) Classical baselines are hard to beat within-domain, but brittle across domains.* On structured, tabular data, well-established GBDT pipelines provide robust discrimination, native probabilistic outputs, and efficient deployment, making them the appropriate default choice for production systems where local training data is abundant (Grinsztajn et al., 2022b). However, our cross-hospital generalization experiments reveal an advantage for LLMs in zero-data settings. While conventional models suffer performance degradation under distribution shift, LLMs maintain better discrimination compared to conventional models in a vast majority of settings, without the need for data from a target domain. This finding reframes the value proposition of LLMs: their pursuit should be driven by capabilities that ConvML or ConvDL do not offer, such as natural-language explanations, task re-specification without retraining, and crucially, robustness to distribution shifts. The off-the-shelf performance of frontier LLMs across all three tasks is broadly consistent with recent work on cross-site generalization in ICUs Tranchellini et al. (2025), which argues for selecting models according to local data availability and deployment constraints. Our results extend this view: because zero shot and few shot LLM approaches require no site-specific training and resist the performance collapse seen in transferred GBDTs, they are the superior pragmatic "day-zero" option for hospitals with limited labeled data or MLOps capacity. As local data accumulate, institutions can transition to ConvML or ConvDL models trained on site, which in our study deliver the most stable and highest performance. Looking ahead, as LLM architectures, prompting strategies, and workflows improve, along with better temporal pretraining, retrieval over EHRs, and lightweight fine-tuning, we expect LLM-based pipelines to match and, in some settings, surpass the best conventional ML and deep-learning baselines. Benchmarks like PULSE will be essential to verify and contextualize such progress.

*(ii) The role of LLMs may be synthesis, not direct prediction.* The superior performance of the HRW suggests a powerful and generalizable paradigm for integrating LLMs with structured data. Instead of tasking the LLM with the difficult problem of direct numerical prediction from raw sequences, this approach leverages the LLM as a high-level reasoning and synthesis engine. It combines the statistically robust output of a specialized predictive model with other data sources to generate a final, explainable assessment. Interestingly, our generalization results show that HRW performance degradation was less compared to XGBoost or LightGBM performance degradations, showing the robustness and value of integrating ConvML outputs with LLMs. The human study also showed the usefulness of LLM explanations to clinicians. Hence, this hybrid model is a more plausible and powerful path toward deploying LLMs in domains such as clinical medicine where both performance, trust, and robustness of models are paramount.

Based on our findings, we offer the following recommendations. First, model selection should be governed by data availability and domain stability rather than accuracy alone. While conventional models are superior in stable, data-rich settings, off-the-shelf LLMs should be adopted as the pragmatic "day-zero" solution in hospitals lacking labeled data, given their resilience to the distribution shifts that cripple standard classifiers. Second, we recommend the HRW as a target architecture for high-stakes deployment. By using LLMs to synthesize outputs from classical models rather than replacing them, practitioners can achieve a balance of high discrimination, interpretability, and improved robustness across datasets.

**Limitations and Future Work.** PULSE has limitations that point toward important avenues for future work. Our use of integer confidence scores from LLMs, while a practical necessity for API-based models, is an approximation of true probabilistic output and may affect calibration analyses. Another limitation is our deliberate choice not to fine-tune the LLMs. This was necessary to establish a rigorous zero-shot baseline, but it likely underestimates their full potential. Our human study with clinicians is a preliminary step towards validating the utility of LLM explanations in ICU settings. This should be followed up with large scale studies for generalizable and in-depth conclusions. Further, our benchmark is designed to be easily extensible, and we envision several critical next steps for the community: **Fine-tuning:** A systematic evaluation of parameter-efficient fine-tuning (PEFT) techniques on the PULSE benchmark is the most crucial next step. This will determine whether domain-specific adaptation can close the performance gap with ConvML models; **Calibration:** Future work should integrate and evaluate post-hoc calibration methods, such as isotonic regression or temperature scaling, directly into the PULSE pipeline to quantify their effectiveness at mitigating the overconfidence we observed in many LLMs; **Explainability:** Moving beyond qualitative analysis, future research should develop and integrate quantitative metrics for evaluating the faithfulness of LLM-generated explanations in clinical settings, ensuring they accurately reflect the model's decision-making process. LLM-as-a-judge pipelines (Ashkinaze et al., 2025) and social ensembles with AI Agents (Li et al., 2025) are some paradigms that could be useful. The PULSE website contains the generated explanations, for researchers to work on such directions.

## 6 CONCLUSION

We introduced PULSE, a large language model benchmark for ICU time-series classification that, for the first time, enables a rigorous, head-to-head comparison of conventional machine learning (ConvML), conventional deep learning (ConvDL), and large language models (LLMs) under the same protocol. Across three datasets and three clinical tasks, our findings are clear: ConvML/ConvDL remain the reference standard for predictive accuracy. Among LLMs, proprietary frontier models are the most competitive, in particular when deployed within a hybrid reasoning workflow that grounds their outputs in the predictions of a strong conventional model. Finally, Few shot and Zero Shot approaches also perform with high AUROC scores above 0.8 for some tas, with frontier models. They are superior "day zero" predictors compared to other models, hence better for out of domain settings. PULSE turns LLM based ICU time-series evaluation from bespoke case studies into a shared, reproducible standard, providing a crucial resource for both machine learning and clinical researchers working in the field.

ETHICS STATEMENT

We use only publicly available, de-identified ICU datasets (HiRID, MIMIC-IV, and eICU) under their respective licenses and data-use agreements. We do not manage or grant access to these datasets and we do not redistribute any patient-level data. Prospective users should obtain access directly through the credentialing procedures described on each dataset's official (e.g., PhysioNet) page. All analyses were conducted on de-identified structured time-series features; no attempt was made to re-identify individuals. Results are reported only in aggregate. When interacting with commercial LLMs, we used API configurations that, per provider documentation, do not retain customer content for training and that permit data-processing opt-outs. No data features were fed into LLMs as they appear on the datasets, as they were heavily preprocessed and converted to text formats. Further, we transmitted no direct identifiers and avoided free-text PHI in prompts.

REPRODUCIBILITY STATEMENT

To maximize transparency, reproducibility, and community engagement, all artifacts from the PULSE benchmark are hosted on a public website (URL: `www.anonymized-for-double-blind-review.com`). This includes detailed metrics, raw predictions, and the exact configuration files for every experimental run. The website features interactive leaderboards and filtering capabilities, allowing users to slice results by model, task, dataset, and prompting strategy. The site is designed as a living resource that will be updated automatically as new models and approaches are evaluated, fostering ongoing, standardized comparison in the field. A preview of the website is available in Appendix A.13. We will also release the benchmark code on Github (URL: `www.github.com/anonymized-for-double-blind-review.com`) along with log outputs from LLMs, for further scientific analysis.

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

# A Appendix

## A.1 Dataset Details

| Dataset | HiRID | MIMIC-IV | eICU |
|---|---|---|---|
| Originally published | 2020 (Hyland et al. (Hyland et al., 2020)) | 2020 (Johnson et al. (Johnson et al., 2023)) | 2017 (Pollard et al. (Pollard et al., 2017)) |
| Version | v1.1.1 | v2.0 | v2.0 |
| Years of data collection | 2011–2016 | 2008–2019 | 2014–2015 |
| Origin | Switzerland | USA | USA |
| Frequency (time-series) | 2/5 min | 1 h | 5 min |
| Number of patients | 33,905 | 53,090 | 168,918 |
| Number of ICU stays | 32,338 | 75,652 | 182,774 |
| Admission type | – | Medical: 49,217 (65%) | Medical: 134,632 (79%) |
| | | Surgical: 25,874 (34%) | Surgical: 51,009 (30%) |
| | | Other: 761 (1%) | Other: 4,702 (3%) |
| Age at admission | 65 [55, 75] | 65 [53, 76] | 65 [53, 78] |
| Gender | Male: 20,796 (84%) | Male: 42,153 (56%) | Male: 98,834 (54%) |
| | Female: 11,542 (38%) | Female: 33,499 (44%) | Female: 83,940 (46%) |
| Race | – | White: 51,575 | White: 140,958 |
| | | Black: 8,223 (12%) | Black: 19,887 (15%) |
| | | Asian: 2,225 (3%) | Asian: 5,008 (3%) |

Table 1: Comparison of HiRID, MIMIC-IV, and eICU datasets. Values are counts with percentages in parentheses unless otherwise specified. These three datasets form the input to the YAIB harmonization pipeline before entering the PULSE Benchmark.

## A.2 TIME SERIES DATA PROCESSING

A critical component of our reasoning workflow is the translation of structured ICU time-series data into a format that is both interpretable and actionable for LLMs. This translation layer is essential for enabling interpretable, high-performance LLM-based prediction in the ICU setting and serves as the interface between raw clinical measurements and the language-based reasoning capabilities of the LLMs in the workflow. Our approach combines feature - wise aggregation, explicit annotation of units and reference ranges, categorization relative to references ranges, linguistic encoding of temporal trends, and transparent communication of data uncertainty due to imputation (see Figure 6 for an overview).

**Time-Series Data Aggregation and Feature Context.**  To optimize LLM performance in clinical prediction tasks, we developed a standardized data formatting pipeline that transforms multivariate time-series data into a concise, feature-centric textual representation. Rather than presenting data in a visit-wise (timestamp-by-timestamp) format, we aggregate each dynamic feature (e.g., heart rate, creatinine) across the data window, computing summary statistics such as minimum, maximum, and mean values (Figure 6-1). This feature-wise aggregation approach is motivated by evidence from both conventional machine learning and LLM literature: dynamic feature generation (min, max, mean) has been shown to improve predictive performance in traditional models (Van De Water et al., 2024; Do et al., 2025), and similar strategies have been adopted in recent LLM-based studies (Sarvari et al., 2024; Yu & Li, 2024). Moreover, summarizing EHR time-series data into concise formats improves LLM prediction capabilities by increasing information density (Zhang et al., 2024). Feature-wise input ordering has been demonstrated to reduce token usage by 37–44% and improve LLM accuracy compared to visit-wise formats (Zhu et al., 2024b). Each feature is represented in the prompt as a line containing its name, aggregated values, unit of measurement, and clinical reference range. This explicit inclusion of units and reference ranges is not only clinically meaningful but has also been shown to improve LLM performance by 13–19% (Zhu et al., 2024e). Additionally, the summarization of time-series data into concise format

**Normal-Abnormal Value Discrimination.**  In addition, we leverage a five-category clinical categorization scheme to directly inform the model how each feature's values relate to their clinical reference ranges (Figure 6-2). Each feature's mean value is classified into one of five categories: a value is considered "normal" if it falls within the established reference interval. Values outside this interval are labeled "slightly low" or "slightly high" if they are within a margin equal to 50% of the reference interval's total width, extending from the lower and upper bounds, respectively. Any value falling beyond this margin is then categorized as "very low" or "very high." These linguistic cues facilitate more robust predictions for LLMs that struggle with arithmetic reasoning (Lewkowycz et al., 2022).

**Temporal Pattern Analysis.**  To further bridge the gap between structured numerical data and the LLM's pretrained language space, we augment each feature's summary with a brief, semantically meaningful phrase describing its temporal trend (Figure 6-3). A linear regression is performed on the time-series data for each feature within the specified window. The resulting slope determines the trend's classification: slopes within $\pm 2\%$ are categorized as "stable"; slopes between $\pm 2\%$ and $\pm 4\%$ are labeled "slowly increasing/decreasing"; those between $\pm 4\%$ and $\pm 8\%$ are "moderately increasing/decreasing"; and slopes exceeding $\pm 8\%$ are considered "rapidly increasing/decreasing". This method is adapted from the text prototype patching mechanism in TIME-LLM (Jin et al., 2024).

**Annotation of Data Uncertainty.**  Additionally, we explicitly communicate data uncertainty by quantifying the proportion of missing or imputed values for each feature (Figure 6-4). This is achieved by analyzing "_na" missingness indicator columns and annotating features as having "complete data," "partially imputed," or "fully imputed" time windows. For incomplete data, the number of imputed values is added as a fraction of the data window length. This strategy is informed by recent findings that LLMs can effectively interpret missing values when they are clearly indicated in the prompt, and that explicit imputation such as LOCF without additional information offers only marginal improvements (Gruver et al., 2024; Zhu et al., 2024e).

| Category | Feature | ricu | unit | Reference Range lower | Reference Range upper | Thinkable Range* lower | Thinkable Range* upper |
|---|---|---|---|---|---|---|---|
| Static Features | sex | sex | M/F | | | | |
| | age | age | years | | | | |
| | height | height | cm | | | 135 | 225 |
| | weight | weight | kg | | | 40 | 250 |
| Dynamic Features — Vital Signs | Heart Rate | hr | bpm | 60 | 100 | 20 | 320 |
| | Systolic Blood Pressure | sbp | mmHg | 90 | 120 | 30 | 300 |
| | Diastolic Blood Pressure | dbp | mmHg | 60 | 80 | 10 | 200 |
| | Mean Arterial Pressure (MAP) | map | mmHg | 65 | 100 | 20 | 250 |
| | Oxygen Saturation | o2sat | % | 95 | 100 | 50 | 100 |
| | Respiratory Rate | resp | /min | 12 | 20 | 4 | 80 |
| | Temperature | temp | °C | 36.5 | 37.5 | 30 | 42 |
| Blood Gas Analysis | pH Level | ph | – | 7.35 | 7.45 | 6.7 | 8 |
| | Partial Pressure of Oxygen (PaO2) | po2 | mmHg | 75 | 100 | 40 | 600 |
| | Partial Pressure of Carbon Dioxide (PaCO2) | pco2 | mmHg | 35 | 45 | 10 | 150 |
| | Base Excess | be | mmol/L | -2 | 2 | -25 | 25 |
| | Bicarbonate | bicar | mmol/L | 22 | 29 | 5 | 50 |
| | Fraction of Inspired Oxygen (FiO2) | fio2 | % | 21 | 100 | 21 | 100 |
| Coagulation | International Normalized Ratio (INR) | inr_pt | – | 0.8 | 1.2 | 0.5 | 20 |
| | Partial Thromboplastin Time (PTT) | ptt | sec | 25 | 35 | 10 | 250 |
| | Fibrinogen | fgn | mg/dL | 200 | 400 | 30 | 1100 |
| Metabolic Panel & Electrolytes | Sodium | na | mmol/L | 135 | 145 | 90 | 170 |
| | Potassium | k | mmol/L | 3.5 | 5 | 1 | 9 |
| | Chloride | cl | mmol/L | 96 | 106 | 70 | 140 |
| | Calcium | ca | mg/dL | 8.5 | 10.5 | 4 | 20 |
| | Ionized Calcium | cai | mmol/L | 1.1 | 1.3 | 0.4 | 2.2 |
| | Magnesium | mg | mg/dL | 1.7 | 2.2 | 0.5 | 5 |
| | Phosphate | phos | mg/dL | 2.5 | 4.5 | 0.5 | 15 |
| | Glucose | glu | mg/dL | 70 | 140 | 25 | 1000 |
| | Lactate | lact | mmol/L | 0.5 | 2 | 0.1 | 20 |
| Liver & Kidney Function | Albumin | alb | g/dL | 3.5 | 5 | 0.5 | 6 |
| | Alkaline Phosphatase | alp | U/L | 44 | 147 | 10 | 1200 |
| | Alanine Aminotransferase (ALT) | alt | U/L | 7 | 56 | 10 | 5000 |
| | Aspartate Aminotransferase (AST) | ast | U/L | 10 | 40 | 10 | 8000 |
| | Total Bilirubin | bili | mg/dL | 0.1 | 1.2 | 0.1 | 50 |
| | Direct Bilirubin | bili_dir | mg/dL | 0 | 0.3 | 0 | 30 |
| | Blood Urea Nitrogen (BUN) | bun | mg/dL | 7 | 20 | 1 | 180 |
| | Creatinine | crea | mg/dL | 0.6 | 1.3 | 0.1 | 20 |
| Hematology & Immune Response | Hemoglobin | hgb | g/dL | 13.5 | 17.5 | 3 | 20 |
| | Mean Corpuscular Hemoglobin (MCH) | mch | pg | 27 | 33 | 15 | 45 |
| | Mean Corpuscular Hemoglobin Concentration (MCHC) | mchc | g/dL | 32 | 36 | 20 | 45 |
| | Mean Corpuscular Volume (MCV) | mcv | fL | 80 | 100 | 50 | 130 |
| | Platelets | plt | $10^3/\mu L$ | 150 | 450 | 10 | 1500 |
| | White Blood Cell Count (WBC) | wbc | $10^3/\mu L$ | 4 | 11 | 0.1 | 500 |
| | Neutrophils | neut | % | 55 | 70 | 0 | 100 |
| | Band Neutrophils | bnd | % | 0 | 6 | 0 | 50 |
| | Lymphocytes | lymph | % | 20 | 40 | 0 | 90 |
| | C-Reactive Protein (CRP) | crp | mg/L | 0 | 10 | 0 | 500 |
| | Methemoglobin | methb | % | 0 | 2 | 0 | 60 |
| Cardiac Markers | Creatine Kinase (CK) | ck | U/L | 30 | 200 | 10 | 100000 |
| | Creatine Kinase-MB (CK-MB) | ckmb | ng/mL | 0 | 5 | 0 | 500 |
| | Troponin T | tnt | ng/mL | 0 | 14 | 0 | 1000 |

Table 2: **Overview of features in harmonized datasets.** Clinical concepts were extracted using ricu. Reference ranges serve time-series data processing and interpretation for LLMs with standard prompting and workflows. Thinkable ranges were defined to facilitate clinical domain-informed outlier handling.

**Implementation and Tooling.** The entire data processing workflow is implemented as a set of reusable workflow utilities, which are invoked automatically whenever time-series data is prepared for workflow ingestion. Unlike some agentic frameworks where the LLM must select and apply analytical tools itself, which is known to be challenging for current models (Zhuo et al., 2025), our system ensures that all workflows receive a consistent, optimized input representation. This design choice isolates the complexity of time-series analysis from the workflow's reasoning process, allowing us to focus on the core challenge of clinical decision-making.

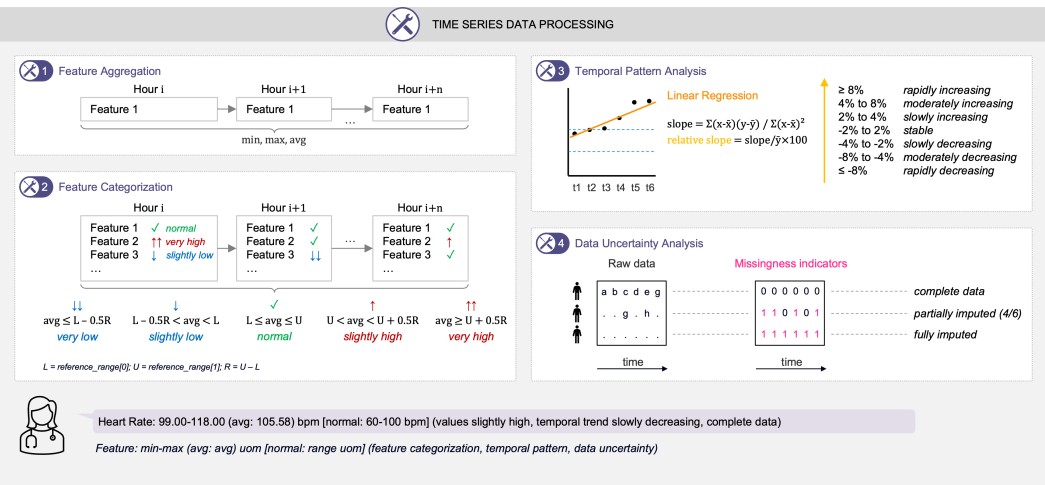

Figure 6: **Time-series data processing for workflow ingestion. 1)** Feature aggregation across the data window into minimum, maximum and average. **2)** Categorization of the aggregated mean into "normal", "slightly low/high" and "very low/high" depending on the value in comparison to the feature's reference range (see 2 reference ranges). **3)** Temporal pattern analysis with linear regression of a feature's values. The resulting slope is used to assign short phrases such as "moderately increasing". **4)** Data uncertainty analysis by counting the number of missingness indicators for a feature within the data window. These tools result in a detailed representation of the time-series data.

## A.3 CONVENTIONAL DL MODELS

Table 3: Hyperparameters for conventional deep learning models.

| Model | Parameter | Default Value |
| --- | --- | --- |
| Common for all convDL models | **Training** | |
| | batch_size | 64 |
| | earlystopping_patience | 10 |
| | min_delta | $1.00 \times 10^{-4}$ |
| | **Scheduler** | |
| | scheduler_type | ReduceLROnPlateau |
| | scheduler_factor | 0.1 |
| | scheduler_patience | 5 |
| | scheduler_cooldown | 0 |
| | min_lr | $1 \times 10^{-6}$ |
| GRU | **Model Architecture** | |
| | input_size | dynamic |
| | hidden_size | 64 |
| | num_layers | 3 |
| | dropout | [0.2, 0.3, 0.3, 0.4, 0.5] |
| | fc_layers | [64, 16] |
| | activation | leaky_relu |
| | bias | TRUE |
| | batch_first | TRUE |
| | bidirectional | FALSE |
| | **Training** | |
| | num_epochs | 100 |
| | criterion | BCEWithLogitsLoss with pos_weight |
| | **Optimization** | |
| | optimizer_name | adam |
| | learning_rate | 0.001 |
| | weight_decay | $1.00 \times 10^{-6}$ |
| | grad_clip_max_norm | 1 |
| LSTM | **Model Architecture** | |
| | input_size | dynamic |
| | hidden_size | 64 |
| | num_layers | 3 |
| | lstm_units | [256, 128, 64] |
| | dense_units | 64 |
| | dropout | [0.2, 0.3, 0.4] |
| | bidirectional | |
| | proj_size | |
| | output_shape | 1 |
| | activation | leaky_relu |
| | bias | TRUE |
| | batch_first | TRUE |
| | **Training** | |
| | num_epochs | 100 |
| | criterion | BCEWithLogitsLoss with pos_weight |
| | **Optimization** | |
| | optimizer | adam |
| | learning_rate | 0.001 |
| | grad_clip_max_norm | 1 |
| | **Model Architecture** | |

22

CNN

Table 3 – continued from previous page

| Model | Parameter | Default Value |
|---|---|---|
| | in_channels | dynamic |
| | out_channels | dynamic |
| | kernel_size | [1, 3, 5] |
| | stride | 1 |
| | dilation | 1 |
| | dropout_rate | 0.3 |
| | groups | 1 |
| | bias | TRUE |
| | padding | same |
| | padding_mode | zeros |
| | output_shape | 1 |
| | pool_size | 2 |
| | activation | leaky_relu |
| | **Training** | |
| | num_epochs | 100 |
| | criterion | BCEWithLogitsLoss with pos_weight |
| | **Optimization** | |
| | optimizer | adam |
| | learning_rate | 0.001 |
| | grad_clip_max_norm | 2 |
| | **Model Architecture** | |
| | input_size | dynamic |
| | depth | 12 |
| | kernel_sizes | [1, 3, 5] |
| | padding | same |
| | padding_mode | zeros |
| InceptionTime | dropout_rate | 0.3 |
| | activation | leaky_relu |
| | **Training** | |
| | num_epochs | 60 |
| | criterion | BCEWithLogitsLoss with pos_weight |
| | **Optimization** | |
| | optimizer_name | adam |
| | learning_rate | 0.001 |
| | weight_decay | 0.01 |
| | grad_clip_max_norm | 2 |

### A.4 LLM Workflows

#### A.4.1 Prompt Engineering Strategies for LLM Workflows

The performance of our LLM framework is fundamentally tied to our prompt engineering strategies. It is well-established that Large Language Models (LLMs) are highly sensitive to the format of their input prompts Zhang et al. (2024); Han et al. (2024). Specifically, performance tends to be significantly lower with condensed, natural language instructions compared to verbose, highly structured prompts Zhuo et al. (2025). Therefore, we developed a multi-faceted prompt engineering approach to optimize the reasoning capabilities of each LLM in the workflow. Furthermore, all LLM-prompts are designed to work flexibly with the three ICU prediction tasks, so that all logic for prompt construction and feature selection is dynamically adjusted based on the current task and the same workflow logic can be reused for different clinical endpoints.

**Workflow System Messages and Persona.** A core component of our strategy was the use of tailored system messages and persona assignment. For each intermediate step within the workflow, the system message assigned a specific persona to the LLM, such as an "experienced clinician" or a "data analyst." This technique is employed to prime the model to reason from a particular expert viewpoint, which has been shown to improve the quality and relevance of its responses in specialized domains Castagnari et al. (2024); Wang et al. (2025); Zhang et al. (2024); Zhu et al. (2024c). This approach enabled a single LLM instance to effectively simulate a team of specialists collaborating on a task. To ensure an unbiased comparison between our workflows and the baseline standard prompting methods, the final prediction step of every LLM uniformly used the standardized PULSE system message.

**Patient Demographics and Clinical Context.** To ground the model's predictions in patient-specific and clinical reality, every prompt that included time-series data was also supplemented with the patient's demographic information (sex, age, height, and weight). This practice is supported by evidence indicating that the inclusion of such static features can improve classification performance in ICU prediction tasks across various models and datasets Van De Water et al. (2024); Ehlers et al. (2025). Furthermore, we enriched the prompts with detailed clinical context, including definitions and background information relevant to the prediction task at hand (Table 4). This technique effectively "anchors" the input data within the correct medical domain, guiding the model to access its most relevant knowledge Liu et al. (2023). Research has consistently shown that embedding domain-specific knowledge, such as rules from clinical diagnosis guidelines, into prompts enhances model performance over more basic approaches Pillai et al. (2025); Castagnari et al. (2024); Gu et al. (2025).

**Guidance, Rules and Warnings.** We provided explicit instructional guidance and enforced a structured output format. Depending on the workflow and its specific task, prompts contained detailed guidance on how to approach the problem, what to prioritize, what to avoid, and which specific details required attention. For all intermediate steps that passed information to another LLM, the prompt strictly mandated that the output be a JSON object. The required schema, including keys and a description of their corresponding values, was clearly defined. This measure was crucial for ensuring the reliability, consistency, and seamless interoperability of data flow between the LLMs in the workflow.

#### A.4.2 Workflow 1: Summary Workflow (SumWorkflow)

**Concept and Purpose.** The SumWorkflow (Summary Workflow) implements a two-step reasoning workflow designed to mirror a simple but common clinical diagnostic process. The core idea is that a focused summary of a patient's condition provides a pre-filtered and more interpretable representation of raw data for an LLM. This process is analogous to a resident physician (Step 1) analyzing patient data and presenting a summary of key findings to an attending physician (Step 2), who then performs the final diagnosis. This sequential approach first identifies and summarizes abnormal patient features and then uses this concise summary to generate a final prediction. This workflow's design is inspired by recent research demonstrating the utility of LLM-generated summaries in clinical prediction tasks. For example, the EMERGE framework uses an LLM to create a summary based on abnormal time-series data and information retrieved from unstructured clinical

| Task | Clinical Context - Short | Clinical Context - Long |
|------|--------------------------|-------------------------|
| Mortality | ICU mortality refers to death occurring during the ICU stay. Key risk factors include hemodynamic instability, respiratory failure, multi-organ dysfunction, and severe metabolic derangements. | Mortality refers to the occurrence of death within a specific population and time period. In the context of ICU patients, the task involves analyzing information from the first 25 hours of a patient's ICU stay to predict whether the patient will survive the remainder of their stay. This prediction task supports early risk assessment and clinical decision-making in critical care settings. |
| AKI | Acute kidney injury (AKI) is defined by rapid decline in kidney function with increased creatinine ($\geq$1.5x baseline or $\geq$0.3 mg/dL increase in 48h) or decreased urine output ($<$0.5 mL/kg/h for 6-12h). Common causes include sepsis, hypotension, and nephrotoxins. | Acute kidney injury (AKI) is a subset of acute kidney diseases and disorders (AKD), characterized by a rapid decline in kidney function occurring within 7 days, with health implications. According to KDIGO criteria, AKI is diagnosed when there is an increase in serum creatinine to $\geq$1.5 times baseline within the prior 7 days, or an increase in serum creatinine by $\geq$0.3 mg/dL ($\geq$26.5 µmol/L) within 48 hours, or urine output $<$0.5 mL/kg/h for 6–12 hours. The most common causes of AKI include sepsis, ischemia from hypotension or shock, and nephrotoxic exposures such as certain medications or contrast workflows. |
| Sepsis | Sepsis is a life-threatening organ dysfunction caused by a dysregulated host response to infection. It is diagnosed by an increase in the SOFA score of $\geq$2 points in the presence of suspected infection. Key indicators include fever, tachycardia, tachypnea, altered mental status, and laboratory abnormalities. | Sepsis is a life-threatening condition characterized by organ dysfunction resulting from a dysregulated host response to infection. It is diagnosed when a suspected or confirmed infection is accompanied by an acute increase of two or more points in the patient's Sequential Organ Failure Assessment (SOFA) score relative to their baseline. The SOFA score evaluates six physiological parameters: the ratio of partial pressure of oxygen to the fraction of inspired oxygen, mean arterial pressure, serum bilirubin concentration, platelet count, serum creatinine level, and the Glasgow Coma Score. A complication of sepsis is septic shock, which is marked by a drop in blood pressure and elevated lactate levels. Indicators of suspected infection may include positive blood cultures or the initiation of antibiotic therapy. |

Table 4: **Clinical context for prompt templates.** This table provides short and long clinical descriptions for the mortality, AKI, and sepsis prediction tasks. Long descriptions are used for the SW, short descriptions for CRW, CW, and HRW.

notes, which is then fed into a multimodal fusion model for prediction Zhu et al. (2024c). A similar approach explored the fusion of text embeddings of LLM-generated summaries based on progress notes with other visit-level features to improve predictive accuracy Choudhuri et al. (2025). The overview and detailed structure of the workflow are provided in Figure 7 and **??** respectively.

**Workflow Design.** In the first step the workflow is provided with time-series data that has been pre-processed to categorize each clinical feature based on its mean value across the data window (i.e., within, below, or above the normal reference range). Critically, only the features with abnormal mean values are presented to the model. To guide the summarization process, the prompt for this step is enriched with extensive, task-specific clinical context, including disease definitions (Table 4, long context). This strategy, designed to imitate diagnosis-focused Retrieval-Augmented Generation (RAG), directs the model to create a summary that is relevant to the specific prediction task. The prompt then instructs the LLM to adopt the persona of an objective medical analyst and produce a factual, plain-text summary of these findings. The second and last step, uses the plain-text summary generated in the first step as its main input and the LLM is prompted to perform the final prediction for the specific clinical outcome. In contrast to the first step, the output for the final prediction step is required to be a JSON object as specified in the PULSE system message.

### A.4.3 WORKFLOW 2: HYBRID REASONING WORKFLOW (HRW)

**Concept and Purpose.** The HRW is engineered to integrate the quantitative predictions of a conventional ML model with the qualitative, knowledge-driven reasoning of an LLM. Its primary purpose is to synthesize data-driven risk stratification with expert-like clinical interpretation for the prediction of critical ICU outcomes. This architecture is predicated on the principle that providing an LLM with the outputs of an expert model can ground its analysis and enhance its reasoning capabilities, a strategy supported by recent findings (Wang et al., 2025). For the conventional ML component, we selected XGBoost due to its superior performance among the baseline models evaluated in our framework. The viability of this choice is further supported by previous research where

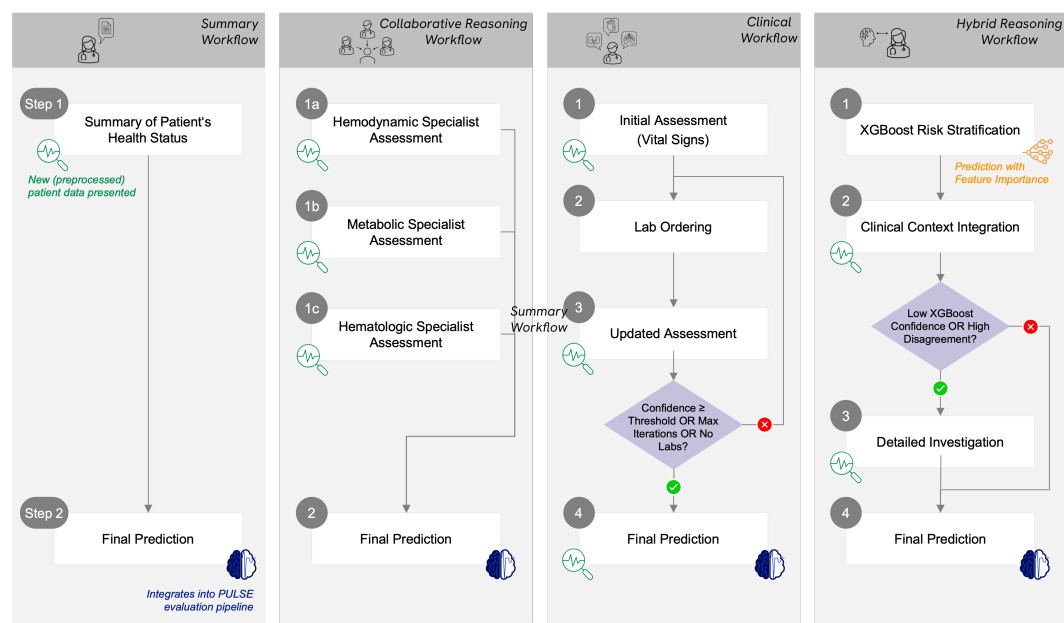

Figure 7: **Overview of agent workflows.** The four workflows (SW, CW, CRW and HRW) follow predefined workflows. Their last step is always the final prediction that integrates into the PULSE evaluation pipeline to enable fair comparisons with standard prompting techniques and also baseline convML and convDL models. New preprocessed patient data is presented in marked steps. The CW and HRW workflows contain conditional decision points based on previous reasoning.

XGBoost was used to generate numerical probabilities for predicting hospital admissions, which were subsequently interpreted by an LLM to contextualize the predictions (Glicksberg et al., 2024). A key innovation of our workflow lies in its ability to perform a confidence-weighted integration of the ML and LLM assessments while explicitly managing any disagreement or uncertainty. This hybrid approach is particularly promising as it facilitates a clinical evaluation of the ML model's output and provides a mechanism to identify and potentially correct implausible predictions. A schematic overview of the workflow's architecture is provided in Figure 7, with a detailed depiction of the workflow and prompt structures in Figure 8 and detailed below.

**ML Risk Stratification and XGBoost Integration.** The workflow begins by loading a pretrained XGBoost model specific to the prediction task and dataset, a selection automated via a standardized naming convention. The patient's time-series and demographic data undergo preprocessing to replicate the feature order and encoding schemes used during model training. The XGBoost model then generates three outputs: a predicted probability for the clinical event (PML), a model-derived confidence score (CML=abs(PML-0.5)×2), and feature importance scores. To optimize the subsequent prompt length and focus the LLM's reasoning, the top 10 unique features with the highest importance scores are selected, a technique shown to enhance predictive performance and efficiency (Zhang et al., 2024). A summary prompt containing the XGBoost probability, its confidence, and the list of these key features with their importance scores and clinical context is then passed to the LLM to elicit an initial clinical interpretation of XGBoost's findings.

**Clinical Context Integration.** In the second step, the workflow prompts the LLM to perform an independent clinical assessment. To facilitate this, the LLM is provided with the detailed time-series data corresponding to the top 10 features that informed the XGBoost prediction. This data is formatted to present temporal trends, markers of uncertainty, and relevant patient demographics. Based on this information alone, the LLM is instructed to generate its own probability estimate for the clinical event, provide a detailed explanation of its clinical reasoning, report a confidence score for its assessment, and explicitly state its agreement or disagreement with the initial XGBoost

prediction. This process yields an independent clinical perspective, and the workflow logs the LLM's outputs for direct comparison against the ML model's quantitative assessment.

**Detailed Investigation (Conditional).** A detailed investigation is triggered if the XGBoost model's confidence falls below a predefined 80% threshold or if the absolute difference between XGBoost's and the LLM's probability estimates exceeds 0.2. When initiated, this step prompts the LLM to conduct a comprehensive analysis, expanding its review to all task-specific features relevant to the clinical outcome. The prompt includes the most task-relevant features that were not presented previously:

- Mortality: `map, sbp, dbp, lact, o2sat, po2, pco2, crea, bun, urine, bili, alt, ast, ph, be, bicar, plt, inr_pt, hr, temp`
- AKI: `crea, urine, bun, map, sbp, na, k`
- Sepsis: `temp, hr, resp, map, sbp, o2sat, po2, pco2, fio2, plt, bili, crea, wbc, neut, bnd, crp, lact`

Furthermore, the prompt contextualizes the task by summarizing the conflicting XGBoost and LLM assessments that prompted the investigation. The LLM is then tasked with re-evaluating the patient's risk, analyzing whether the additional data or more nuanced temporal patterns can resolve the initial uncertainty or disagreement. This step ensures the workflow can leverage a complete set of clinical information to adjudicate complex or ambiguous cases that may not be resolved using only the top-ranked features.

**Confidence-Weighted Synthesis.** The final step synthesizes the outputs from the preceding stages into a single, robust risk estimate using a confidence-weighted average of the ML and LLM-derived probabilities. A dampening mechanism is applied if a detailed investigation was performed and its resulting probability deviates significantly from the initial XGBoost prediction. This logic is designed to limit the influence of potential outlier probabilities from the LLM, particularly when there is high disagreement. The maximum allowable deviation is inversely proportional to the ML model's confidence, with stricter limits for higher confidence scores. For example, an ML confidence of 0.9 or greater allows a maximum deviation of only 0.2, whereas a confidence below 0.7 allows for a larger deviation. If the investigation's probability exceeds this dynamic threshold, it is adjusted to the boundary limit before being used in the final synthesis. This ensures that the robust, data-driven output from XGBoost anchors the result, while the LLM's contextual reasoning provides nuanced adjustments, leading to a final prediction that is both data-grounded and clinically informed.

A.4.4 WORKFLOW 3: COLLABORATIVE REASONING WORKFLOW (CRW)

**Concept and Purpose.** CRW introduces a multi-specialist, modular reasoning approach that simulates a clinical multi-disciplinary team meeting. The core concept is to emulate a virtual team of domain experts (specialists in hemodynamic, metabolic, and hematologic systems) who each provide an independent assessment based on a subset of data relevant to their clinical specialty. This workflow was designed to investigate the LLM's capability of providing enhanced reasoning if focused on a specific organ system and whether the combination of diverse clinical perspective can be synthesized into more robust predictions. By structuring the problem this way, the framework can explicitly model the agreement or disagreement between different lines of reasoning, offering deeper insight into the basis of the final prediction. A schematic overview is shown in Figure 7 and a more detailed view, including specific prompt structures and system messages, is provided in Figure 9 and described below.

**Single- vs. Multi-Agent Clinical Teams.** Our implementation uses a single LLM instance that adopts different specialist personas in a multi-step workflow. While some advanced frameworks employ multiple, distinct LLM agents that deliberate with each other, we intentionally omitted direct inter-agent communication. This design represents a pragmatic choice to manage computational costs and reduce complexity, allowing us to better isolate and understand the value contributed by this specific agentic reasoning structure in time-series prediction tasks. We acknowledge that the effectiveness of multi-agent systems can be critically shaped by their "rules of engagement" or interaction protocols **?**. Our framework offers a more streamlined and practical approach compared

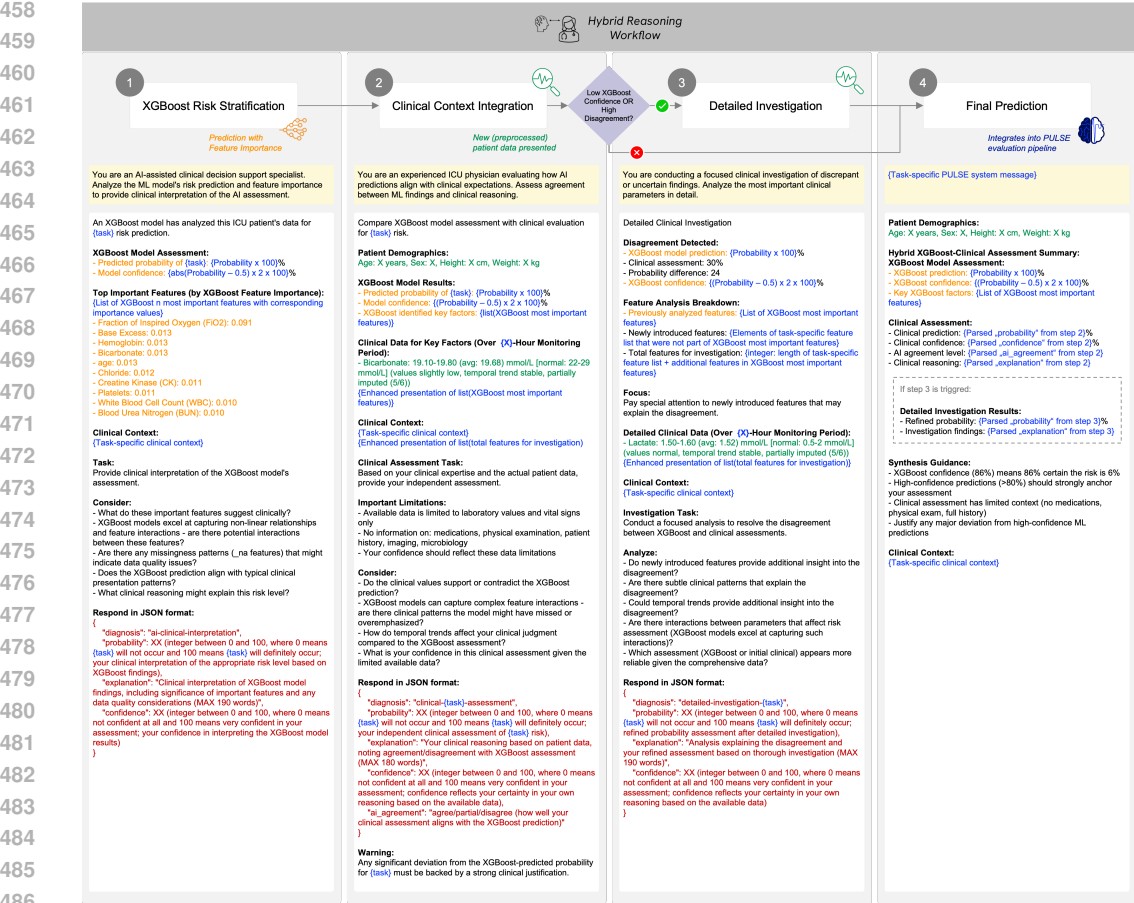

Figure 8: **Prompt templates for the hybrid reasoning workflow.** The HRW coordinates the XG-Boost risk stratification and clinical interpretation before performing a confidence-weighted objective synthesis of ML and LLM probabilities. A conditionally triggered detailed investigation step deepens the clinical assessment by analyzing more features. **Color legend:** orange = data related to XGBoost integration, blue = dynamically created input, green = examples of patient data, red = output and formatting instructions, gray = conditionally included input.

to complex collaborative systems designed to simulate the intricate deliberations of clinical teams. For instance, in the Colacare system, ablation studies revealed that while the inter-agent deliberation process provided only minor improvements to mortality prediction, the contributions of the Retrieval-Augmented Generation (RAG) and Fusion network were even more marginal Wang et al. (2025).

Acknowledging recent critiques that many "multi-agent" LLM systems lack genuine social abilities and instead rely on centrally orchestrated workflows that often reduce to simple aggregation **?**, our implementation adopts a more streamlined and transparent approach. We use a single LLM instance that takes on different specialist personas within a multi-step workflow, intentionally omitting the complex inter-agent deliberation seen in other advanced frameworks. This pragmatic design was chosen not only to manage computational costs and reduce complexity but also to better isolate and understand the value contributed by this specific specialist reasoning structure in time-series prediction tasks. Our decision is further supported by findings from systems like Colacare, where intricate inter-agent deliberation and additional components like Retrieval-Augmented Generation (RAG) offered only marginal improvements to mortality prediction Wang et al. (2025). While we recognize that the effectiveness of true multi-agent systems is critically shaped by their "rules of

engagement" or interaction protocols Ashkinaze et al. (2025), our framework offers a deliberately focused and practical alternative to simulating the complex deliberations of clinical teams.

**Parallel Specialist Assessments.** The first phase of the workflow consists of parallel specialist assessments. Patient time-series data is partitioned into clinically coherent feature groups (Table 2), which are then allocated to one of three specialists. The Hemodynamic Specialist analyzes vital signs and cardiac markers, the Metabolic Specialist analyzes data from blood gas analysis, metabolic panel and electrolytes, as well as tests for liver and kidney function. The Hematologic Specialist analyzes data related to hematology, immune response and coagulation. Each specialist agent is invoked independently from the oder specialist agents with a domain-specific prompt. This prompt contains only the features relevant to its specialty, including their temporal patterns and any available uncertainty information, as well as the non-specific information about patient demographics and clinical context. The specialists are explicitly guided to pay attention to value abnormalities and temporal trends from their unique domain perspective. Each is required to return its assessment as a structured JSON object containing three key pieces of information: the predicted probability for the clinical outcome, the reasoning justifying the prediction, and a numerical confidence score for its assessment.

**Confidence-Weighted Final Prediction.** The second phase is the final prediction that is a confidence-weighted synthesis to form a consesus prediction. The structured JSON outputs from all three specialists are aggregated and fed into a final synthesis prompt. This prompt is handled by an agent with a non-specialist, general clinician persona. The input for this final step includes the predicted probabilities, confidence levels, and reasoning from each specialist, alongside the patient's demographic data and the overarching clinical context. The synthesizer agent's task is to produce the final prediction by weighing each specialist's assessment according to its reported confidence level and by analyzing the overall agreement or disagreement in their independent analyses.

A.4.5 WORKFLOW 4: CLINICAL WORKFLOW (CW)

**Concept and Purpose.** The CW is designed to emulate the iterative, stepwise reasoning of an experienced clinician, such as a "detective" or an "attending physician" during rounds. Its function is to model the dynamic process of forming a differential diagnosis by progressively acquiring and interpreting clinical data. The CW's primary goal is to sequentially assess a patient's risk for a given complication by first analyzing vital signs, and then dynamically deciding which laboratory tests would be most informative to order and review, closely mimicking real-world clinical practice. Our approach is conceptually related to other work on sequential diagnostic agents. For instance, SDBench features agents that engage in multiple cycles of requesting tests from a gatekeeper until a final diagnosis is reached Nori et al. (2025). A simpler, related method was used for anemia classification, where an LLM requested one feature at a time in a sequence Castagnari et al. (2024). Our workflow builds on these concepts by operating through a multi-phase simulation of clinical decision-making with state management that follows a fixed sequence of reasoning steps. This structure facilitates a progressive information acquisition strategy that mirrors how a clinician might work through a complex case. The entire workflow is shown in Figure 7 and a more detailed structure of prompts and system messages at each stage in Figure 10 and detailed below.

**Initial Assessment.** The workflow begins with an initial assessment. In this step, the workflow is provided with the patient's time-series data for vital signs, their demographic information, and the clinical context. Based on this limited initial data, it generates a preliminary risk estimate and a clinical rationale. The prompts for this stage are intentionally engineered to encourage the model to report low confidence unless the clinical picture presented by the vital signs is already unequivocal.

**Lab Ordering and Updated Assessment.** Following the initial assessment, the workflow initiates an iterative reasoning loop composed of two repeating phases: lab ordering and updated assessment. In the lab ordering phase, the workflow leverages its most recent evaluation to identify which laboratory tests would be most informative. The prompt for this step instructs the model to request a clinically plausible subset of four to ten tests and, to prevent redundancy, also provides a list of all features that have been analyzed in previous iterations and dynamically updates the list of available tests by excluding already analyzed features. Subsequently, in the Updated Assessment phase, the

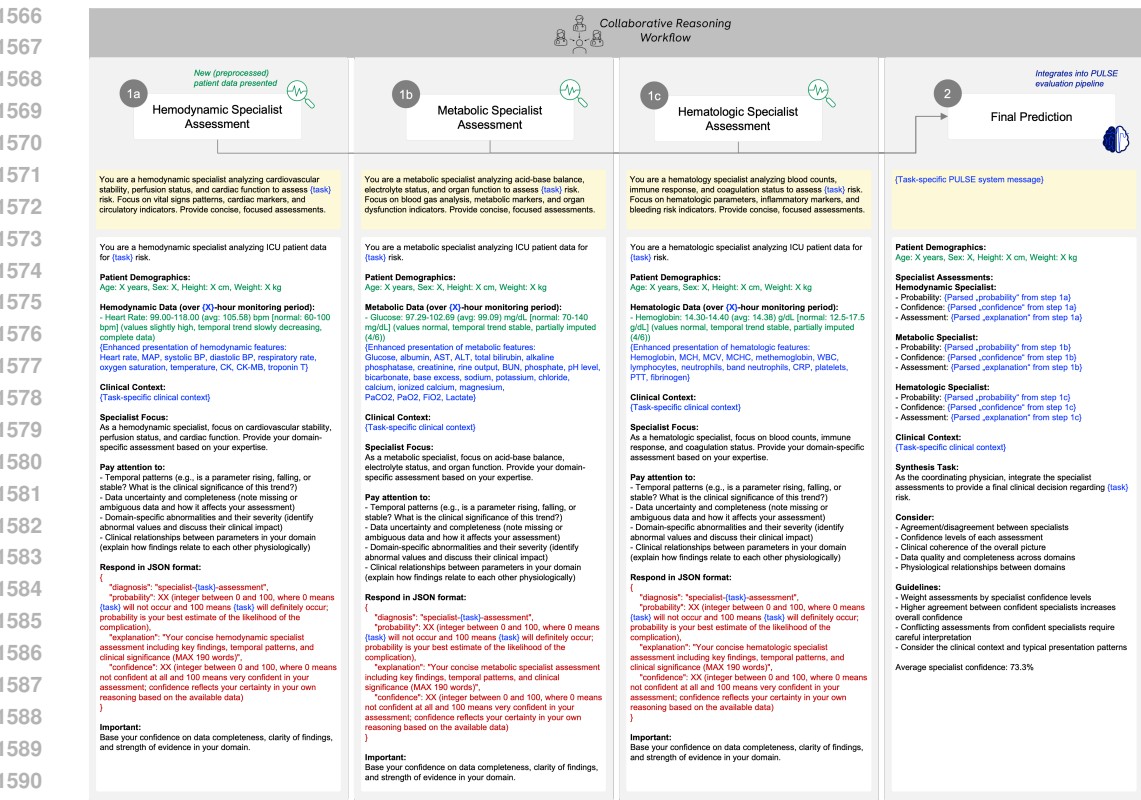

Figure 9: **Prompt templates for the collaborative reasoning workflow.** The CRW coordinates the assessments of three specialists and ultimately the final prediction based on the specialist assessments. **Color legend:** blue = dynamically created input, green = examples of patient data, red = output and formatting instructions.

workflow incorporates the newly provided results of the requested labs to refine its risk evaluation, generating an updated probability, clinical explanation, and confidence score. This entire cycle is governed by a state dictionary that tracks used features and the assessment history, with previously used tests being removed from the pool of available labs. The loop continues until one of three predefined stopping conditions is met: the LLM's confidence score reaches a threshold of 85%, a maximum of five iterations is completed, or the pool of available, unused labs is exhausted.

**Validation of Requested Tests.** A critical tool that supports the lab ordering step is the validation of requested tests. A robust, multi-step validation pipeline was essential to ensure that the workflow's requests were always valid, non-redundant, and feasible within the given dataset. This was necessary because initial experiments showed that without strict guidance, the models would often request clinically sensible tests that were not available in the dataset, thus requiring a guardrail. The validation process initiates immediately after the Lab Ordering step by parsing the model's structured output to extract the list of requested laboratory tests; an improperly formatted response at this stage would halt the iteration. Next, this extracted list is validated against a comprehensive feature dictionary. This step standardizes the requests by resolving common clinical aliases or expanding group names (e.g., "electrolytes") into their specific constituent feature keys, while any requested tests that do not exist in the dataset are disregarded. The resulting list of valid tests is then cross-referenced with the workflow's internal state to filter out any features that have already been analyzed in a previous iteration for the same patient, preventing redundant requests. As a final check, the remaining tests are compared against the specific data available for the current patient. Only the tests that successfully pass every stage of this validation pipeline are ultimately provided to the workflow for the subsequent updated assessment step.

**Final Prediction.** Once the iterative loop terminates, the workflow performs the final prediction. For this conclusive step, the workflow is presented with the complete set of information gathered throughout the workflow: the initial vital signs and the data from all laboratory tests it chose to order. It also receives its own assessment history from the preceding steps. While no time-series data is ultimately discarded, this process allows the final prediction to benefit from the iterative reasoning, which may focus the model on a more noise-reduced, outcome-relevant feature set. The workflow is prompted to synthesize all available information (both the comprehensive time-series data and its own assessment history) to produce a final, well-justified prediction.

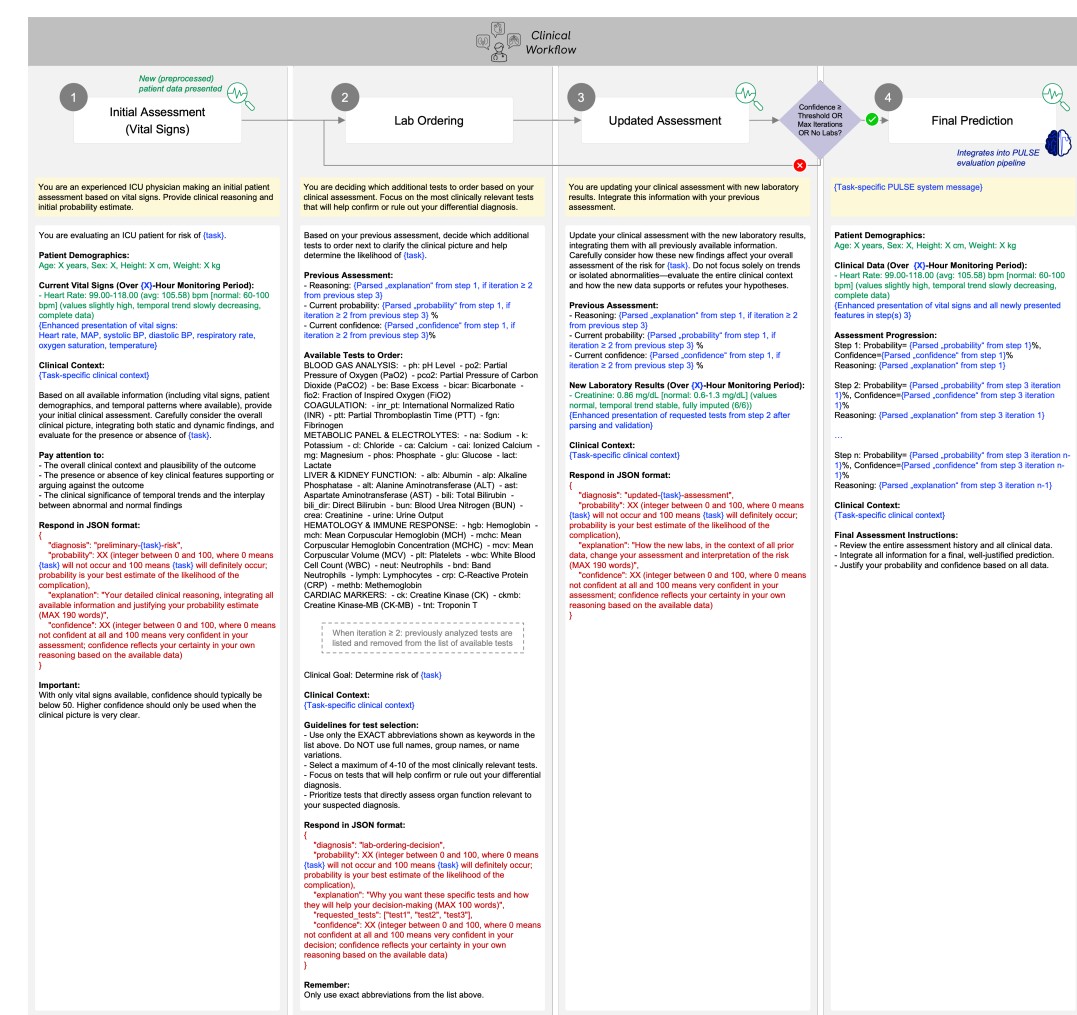

Figure 10: **Prompt templates for the clinical workflow.** The CW coordinates a minimum of 4 reasoning steps: The initial assessment based on vital signs, lab ordering, an updated assessment, and the final prediction. Steps 2 and 3 can be repeated for up to four times. **Color legend:** blue = dynamically created input, green = examples of patient data, red = output and formatting instructions, gray = conditionally included input.

## A.5 LARGE LANGUAGE MODELS

| Model | Availability | Model ID | Source | Parameters | Context Window | Capability | Input Cost (per M tokens) | Output Cost (per M tokens) |
|---|---|---|---|---|---|---|---|---|
| OpenAI-o3 | Proprietary | o3-2025-04-16 | OpenAI | Not Disclosed | 200k tokens | Reasoning | $2.0 | $8.0 |
| Claude-Sonnet-4 | Proprietary | claude-sonnet-4-20250514 | Anthropic | Not Disclosed | 200k tokens | Reasoning | $3.0 | $15.0 |
| Grok-4 | Proprietary | grok-4 | xAI | Not Disclosed | 256k tokens | Reasoning | $3.0 | $15.0 |
| Gemini-2.5-Pro | Proprietary | gemini-2.5-pro | Google | Not Disclosed | 1M tokens | Reasoning | $1.25 | $10.0 |
| Gemini-2.5-Flash | Proprietary | gemini-2.5-flash-preview-05-20 | Google | Not Disclosed | 1M tokens | Reasoning* | $0.3 | $2.5 |
| Llama-3.1-8B | Open-Source | meta-llama/Llama-3.1-8B-Instruct | HuggingFace | 8B | 128k tokens | General | Free (local) | Free (local) |
| DeepSeek-R1-Distill-Llama-8B | Open-Source | deepseek-ai/DeepSeek-R1-Distill-Llama-8B | HuggingFace | 8B | 128k tokens | Reasoning | Free (local) | Free (local) |
| Mistral-7B | Open-Source | mistralai/Mistral-7B-Instruct-v0.3 | HuggingFace | 7B | 32k tokens | General | Free (local) | Free (local) |
| Gemma-3-4B | Open-Source | google/gemma-3-4b-it | HuggingFace | 4B | 128k tokens | General | Free (local) | Free (local) |
| MedGemma-4B | Open-Source | google/medgemma-4b-it | HuggingFace | 4B | 128k tokens | Medical | Free (local) | Free (local) |

Table 5: **Overview of used LLMs.** Overview of selected LLMs, including their availability, size, context window, specialization, and cost. * Reasoning abilities were disabled for use in the PULSE framework.

Table 6: Parameters for Large Language Models. The thinking budget was set to allow the model enough thinking capacity and in parallel limit the cost per sample. For OpenAI this is not directly possible and a reasoning effor of medium was chosen. Temperature is kept constant across models at 0.4 which allows creativity but also encourages adherence to the rules given.

| Model | Parameter | Value |
|---|---|---|
| Llama 3.1-8B-Instruct | Model ID | meta-llama/Llama-3.1-8B-Instruct |
| | max_new_tokens | 300 |
| | max_length | 30000 |
| | temperature | 0.4 |
| | do_sample | TRUE |
| Deepseek-R1 - Llama8b | Model ID | deepseek-ai/DeepSeek-R1-Distill-Llama-8B |
| | max_new_tokens | 20000 |
| | max_length | 30000 |
| | temperature | 0.4 |
| | do_sample | TRUE |
| Mistral-7B-Instruct | Model ID | mistralai/Mistral-7B-Instruct-v0.3 |
| | max_new_tokens | 300 |
| | max_length | 30000 |
| | temperature | 0.4 |
| | do_sample | TRUE |
| Gemini 2.5 Flash | Model ID | gemini-2.5-flash-preview-05-20 |
| | Api provider | Google Cloud |
| | max_new_tokens | 300 |
| | max_length | 30000 |
| | temperature | 0.4 |
| | thinking_budget | 0 |
| Gemini 2.5 Pro | Model ID | gemini-2.5-pro |
| | Api provider | Google Cloud |
| | max_new_tokens | 300 |
| | max_length | 30000 |
| | temperature | 0.4 |
| | thinking_budget | -1 |

Table 6 – continued from previous page

| Model | Parameter | Value |
|-------|-----------|-------|
| Gemma-3-4b-it | Model ID | google/gemma-3-4b-it |
| | max_new_tokens | 300 |
| | max_length | 30000 |
| | temperature | 0.4 |
| | do_sample | TRUE |
| MedGemma-4b | Model ID | google/medgemma-4b-it |
| | max_new_tokens | 300 |
| | max_length | 40000 |
| | temperature | 0.4 |
| | do_sample | TRUE |
| GPT4o | Model ID | 2024-12-01-preview |
| | Api provider | Azure |
| | max_new_tokens | 300 |
| | max_length | 30000 |
| | temperature | 0.4 |
| OpenAI o3 | Model ID | o3-2025-04-16 |
| | Api provider | OpenAI |
| | reasoning_effort | medium |
| | max_new_tokens | 30000 |
| | max_length | 30000 |
| | temperature | 0.4 |
| Claude Sonnet 4 | Model ID | claude-sonnet-4-20250514 |
| | Api provider | Anthropics |
| | thinking_budget | 5000 |
| | max_new_tokens | 10000 |
| | max_length | 30000 |
| | temperature | 0.4 |
| Grok 4 | Model ID | grok-4 |
| | Api provider | X-AI |
| | thinking_budget | 5000 |
| | max_new_tokens | 30000 |
| | max_length | 30000 |
| | temperature | 0.4 |

A.6 COHORT SELECTION METHODOLOGY

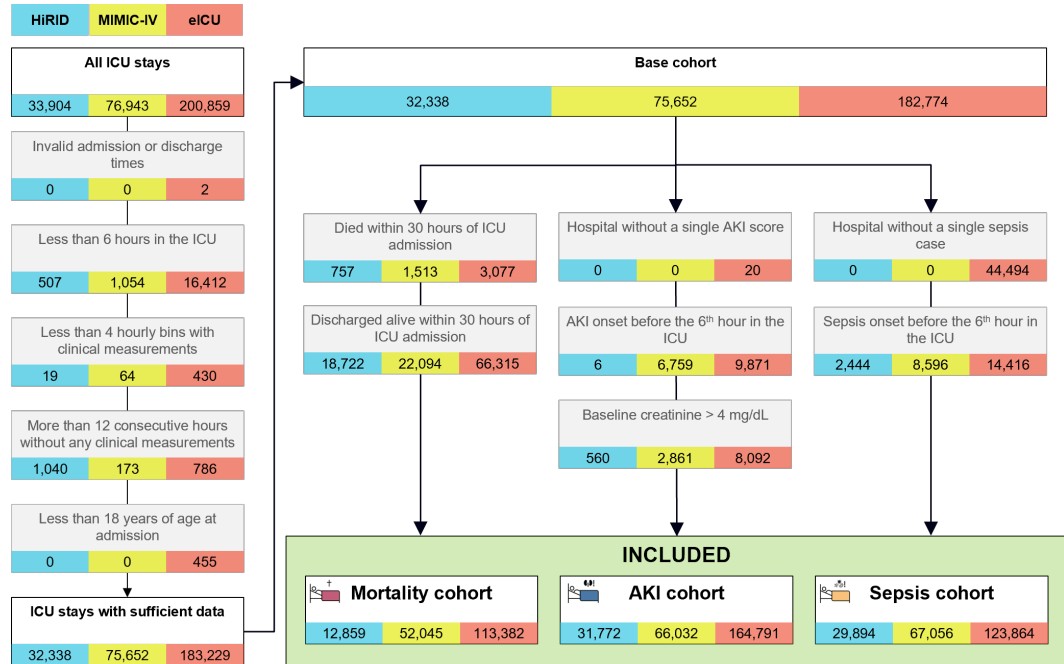

Figure 11: Cohort Selection for HiRID (blue), MIMIC-IV (yellow) and eICU (red) datasets before harmonization for Mortality, AKI and Sepsis tasks. Each ICU stay is checked for validity and sufficient length. All values represent the number of stay-id's in a current filtering stage. This harmonized data is the base input for the PULSE benchmark.

## A.7 KDIGO Stages of AKI

| Stage | Serum Creatinine | Urine Output |
|-------|------------------|--------------|
| 1 | 1.5-1.9$\times$ baseline *or* $\geq 0.3\,mg/dL$ ($\geq 26.5\,\mu mol/L$) increase within $48\,h$ | $< 0.5\,mL/kg/h$ for 6-12 $h$ |
| 2 | 2.0-2.9$\times$ baseline | $< 0.5\,mL/kg/h$ for $\geq 12\,h$ |
| 3 | 3.0$\times$ baseline (within 7 $d$) *or* $\geq 4.0\,mg/dL$ ($\geq 353.6\,\mu mol/L$) increase within $48\,h$ *or* initiation of renal replacement therapy | $< 0.3\,mL/kg/h$ for $\geq 24\,h$ *or* anuria for $\geq 12\,h$ |

Table 7: KDIGO stages of AKI based on serum creatinine and urine output criteria. A patient is classified as having AKI with KDIGO stage $\geq$1.

## A.8 SUMMARY OF HARMONIZED DATASETS USED IN THE BENCHMARK

| Task | Dataset | Total Stays | Cases (n, %) | Controls (n, %) | Positive Labels | Negative Labels |
|---|---|---|---|---|---|---|
| | **HiRID** | | | | | |
| | train | 10,287 | 884 (8.6%) | 9,403 (91.4%) | NA | NA |
| Mortality | val | 1,285 | 108 (8.4%) | 1,177 (91.6%) | NA | NA |
| | test | 1,287 | 105 (8.2%) | 1,182 (91.8%) | NA | NA |
| | test_small (sampled 5 times) | 100 | 11 (11.0%) | 89 (89.0%) | NA | NA |
| | **MIMIC-IV** | | | | | |
| | train | 39,618 | 2,815 (7.1%) | 36,803 (92.9%) | NA | NA |
| | val | 4,952 | 381 (7.7%) | 4,571 (92.3%) | NA | NA |
| | test | 4,953 | 388 (7.8%) | 4,565 (92.2%) | NA | NA |
| | test_small (sampled 5 times) | 100 | 12 (12.0%) | 88 (88.0%) | NA | NA |
| | **eICU** | | | | | |
| | train | 90,704 | 4,723 (5.2%) | 85,981 (94.8%) | NA | NA |
| | val | 11,338 | 731 (6.4%) | 10,607 (93.6%) | NA | NA |
| | test | 11,339 | 799 (7.0%) | 10,540 (93.0%) | NA | NA |
| | test_small (sampled 5 times) | 100 | 7 (7.0%) | 93 (93.0%) | NA | NA |
| | **HiRID** | | | | | |
| | train | 25,416 | 5,913 (23.3%) | 19,503 (76.7%) | 69,741 (8.9%) | 712,061 (91.1%) |
| AKI | val | 3,177 | 750 (23.6%) | 2,427 (76.4%) | 8,895 (9.0%) | 90,103 (91.0%) |
| | test | 3,178 | 720 (22.7%) | 2,458 (77.3%) | 8,483 (8.9%) | 86,391 (91.1%) |
| | test_small (sampled 5 times) | 100 | 22 (22.0%) | 78 (78.0%) | 159 (16.2%) | 823 (83.8%) |
| | **MIMIC-IV** | | | | | |
| | train | 50,254 | 20,934 (41.7%) | 29,320 (58.3%) | 254,850 (13.3%) | 1,654,805 (86.7%) |
| | val | 6,281 | 2,640 (42.0%) | 3,641 (58.0%) | 31,967 (13.3%) | 208,335 (86.7%) |
| | test | 6,283 | 2,629 (41.8%) | 3,654 (58.2%) | 31,931 (13.2%) | 210,790 (86.8%) |
| | test_small (sampled 5 times) | 100 | 40 (40.0%) | 60 (60.0%) | 229 (23.3%) | 755 (76.7%) |
| | **eICU** | | | | | |
| | train | 131,905 | 48,306 (36.6%) | 83,599 (63.4%) | 586,147 (12.7%) | 4,040,317 (87.3%) |
| | val | 16,488 | 6,554 (39.8%) | 9,934 (60.2%) | 79,108 (13.3%) | 514,469 (86.7%) |
| | test | 16,489 | 7,453 (45.2%) | 9,036 (54.8%) | 89,423 (15.3%) | 494,866 (84.7%) |
| | test_small (sampled 5 times) | 100 | 54 (54.0%) | 46 (46.0%) | 305 (31.0%) | 679 (69.0%) |
| | **HiRID** | | | | | |
| | train | 23,758 | 1,463 (6.2%) | 22,295 (93.8%) | 17,131 (2.4%) | 693,059 (97.6%) |
| Sepsis | val | 2,969 | 205 (6.9%) | 2,764 (93.1%) | 2,369 (2.7%) | 85,406 (97.3%) |
| | test | 2,971 | 190 (6.4%) | 2,781 (93.6%) | 2,229 (2.6%) | 84,397 (97.4%) |
| | test_small (sampled 5 times) | 100 | 9 (9.0%) | 91 (91.0%) | 73 (7.5%) | 896 (92.5%) |
| | **MIMIC-IV** | | | | | |
| | train | 50,740 | 2,645 (5.2%) | 48,095 (94.8%) | 31,189 (1.3%) | 2,403,897 (98.7%) |
| | val | 6,342 | 347 (5.5%) | 5,995 (94.5%) | 4,051 (1.3%) | 300,508 (98.7%) |
| | test | 6,343 | 329 (5.2%) | 6,014 (94.8%) | 3,858 (1.3%) | 298,976 (98.7%) |
| | test_small (sampled 5 times) | 100 | 5 (5.0%) | 95 (95.0%) | 25 (2.5%) | 965 (97.5%) |
| | **eICU** | | | | | |
| | train | 98,730 | 4,781 (4.8%) | 93,949 (95.2%) | 57,276 (1.3%) | 4,428,935 (98.7%) |
| | val | 12,341 | 497 (4.0%) | 11,844 (96.0%) | 5,955 (0.9%) | 622,218 (99.1%) |
| | test | 12,342 | 360 (2.9%) | 11,982 (97.1%) | 4,355 (0.6%) | 681,239 (99.4%) |
| | test_small (sampled 5 times) | 100 | 8 (8.0%) | 92 (92.0%) | 47 (4.8%) | 933 (95.2%) |

Table 8: Summary of harmonized datasets used in benchmark. Test-limited100-window10 is the test set limited to the first 100 stay-id's. Each stay-id is sampled 10 times with a window size of 6h. The limited test set was used to generate all results reported in this report. We used a limited set to control the api costs for proprietary LLMs.

A.9 OPERATIONAL PERFORMANCE

The operational performance of LLM-based approaches was evaluated by measuring input tokens, output tokens, tokenization time, and inference time per sample (Figure 12). Due to their multi-step reasoning processes, agentic frameworks and workflows naturally incurred higher inference times compared to single-step standard prompts. For instance, the average processing time for an agentic prompt was 45.65 seconds, compared to 12.26 seconds for a standard prompt. The CW, with its iterative workflow, was the slowest, averaging 76.48 seconds per sample. Conversely, workflows often required fewer input tokens than Few-Shot standard prompts, as they were not designed to take in-context examples and often performed feature selection. Reasoning-enhanced models, such as OpenAI-o3 and Claude-Sonnet-4, also exhibited longer inference times due to their internal thought processes. Deepseek-R1-Distill-Llama-8B, The only open-source LLM with reasoning capabilities, occasionally became "stuck" in repetitive reasoning loops, leading to outlier values in output token counts and inference times.

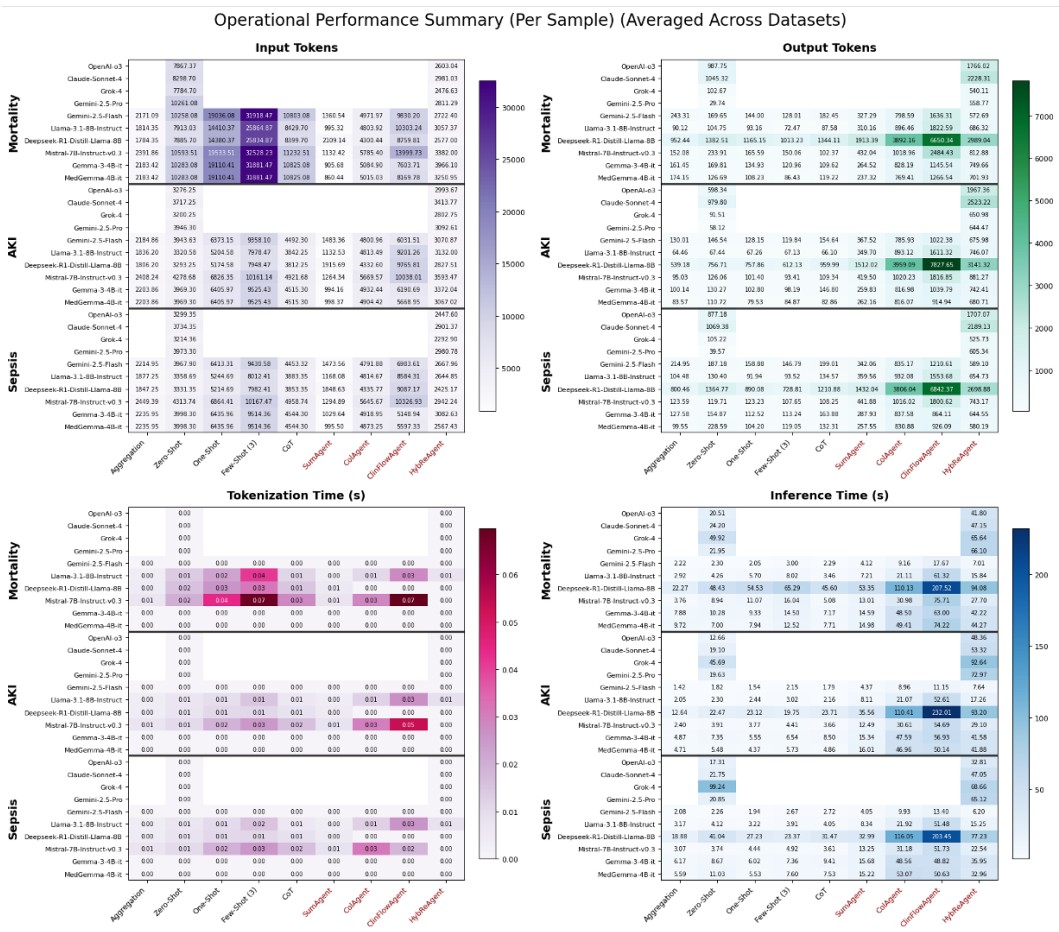

Figure 12: **Operational performance metrics per sample.** This figure provides a detailed overview of the operational performance for each model and prompting strategy, averaged across the three ICU datasets. The heatmaps display four key metrics: input tokens, output tokens, tokenization time (set to 0.0 for proprietary models), and inference time. The results are stratified by clinical task (rows) and prompting approach (columns). Empty heatmap cells are model-approach combinations that were not part of our experimental setup.

## A.10 EFFICIENCY AND COST

The trade-off between cost, performance, and computational efficiency is a critical consideration for practical deployment. As expected, the increased complexity of workflow methods resulted in lower processing throughput (tokens per second) compared to standard prompting. This is because thinking tokens are not included in the LLM's processing throughput, and additional workflow tasks unrelated to LLM inference also contribute to overall processing time. Workflow-based methods achieved a mean throughput of 187.0 tokens/sec, whereas standard prompts were significantly faster at 1432.3 tokens/sec (Figure 5 left).

A Pareto Frontier analysis was conducted to identify strategies offering the optimal balance between cost and predictive performance (mean AUROC) for proprietary models (Figure 5 right). The analysis revealed a clear trade-off. The most cost-efficient strategy on the Pareto Frontier was Gemini-2.5-Flash with the Aggregation prompt, costing only $0.001 per sample with a mean AUROC of 0.606 and an inference time of 1.91 seconds. At the other end of the spectrum, the highest-performing strategy was OpenAI-o3 with the HRW, which achieved a mean AUROC of 0.753 at a cost of $0.065 per sample and an inference time of 41.00 seconds. The most expensive approach - not on the Pareto frontier was Claude-Sonnet-4 with the HRW, costing $0.146 per sample. These findings underscore that while complex workflow-based systems like the HRW deliver superior predictive accuracy and reliability, this comes at a computational and financial cost. The choice of model and strategy must therefore be carefully aligned with the specific clinical application's requirements for performance, speed, and budget.

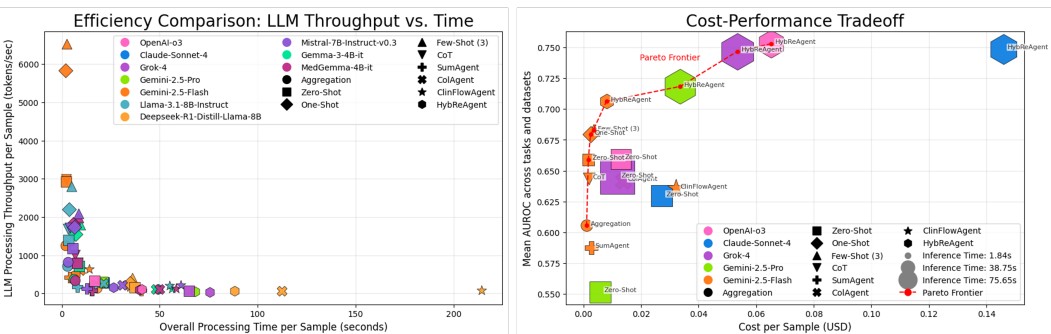

Figure 13: **Efficiency Comparison and Cost-Performance Tradeoff.** This figure analyzes the efficiency and cost-effectiveness of the evaluated LLM approaches. The left panel plots the processing throughput (tokens/sec) against the overall processing time per sample, demonstrating that LLM workflows have lower throughput due to their complex, multi-step nature. The right panel presents a cost-performance tradeoff analysis for proprietary models, plotting the mean AUROC against the cost per sample in USD, with makrer sizes indicating inference time. The Pareto Frontier (red dashed line) highlights optimal cost-performance trade-offs. The analysis reveals that several HRW methods are positioned on the Pareto frontier.

## A.11 System Messages for Various Prompts

To select system message to be used in the broader study, we ablate five system-message variants (SM1–SM5) on four open LLMs (`Llama-3-8B-Instruct`, `Mistral-7B-Instruct-v0.2`, `gemma-2-9b-it`, `DeepSeek-Coder-V2-Lite-Instruct`) over all three tasks. MCC-2 denotes MCC from the *discrete* class prediction (see §3.5 and **??**). Results appear in Figure 14.

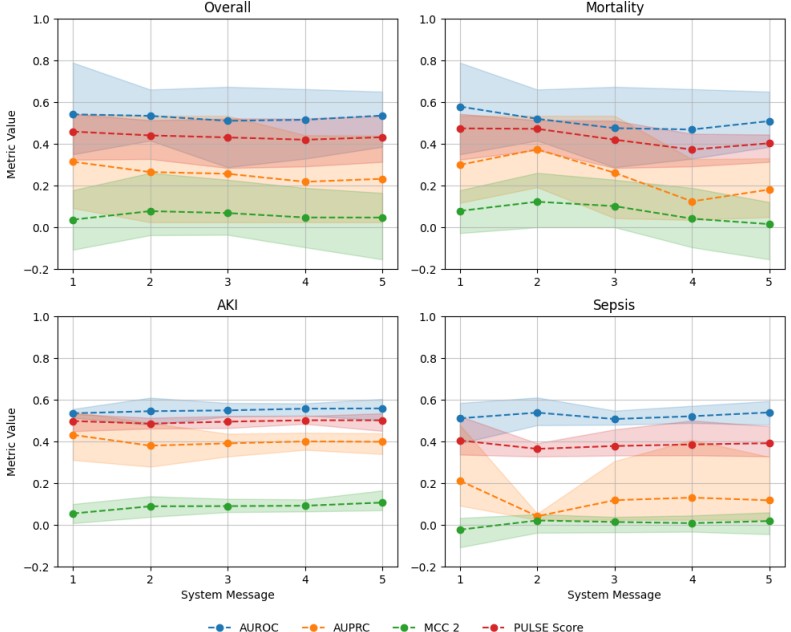

Figure 14: System-message ablation: AUROC, AUPRC, MCC, and PULSE (secondary). Shaded areas show min–max across LLMs for a given message.

**Key findings.** SM1 exhibits the largest between-model variance, especially on mortality (AUROC spread up to 0.44). Adding task-specific exemplars (SM2) consistently reduces variance (e.g., mean AUROC spread ≈ 0.245; AKI is an exception), suggesting more stable adherence. Further additions (probability calibration rules, ICU context, detailed JSON schema; SM3–SM5) *do not* yield consistent gains and sometimes slightly degrade performance—likely due to prompt length/complexity effects on smaller LLMs, echoing observations in prompt-engineering studies (Mu et al., 2025). In practice, a concise exemplar-driven message (SM2) balances stability and cost.

---

**SYSTEM MESSAGE - Sepsis**

You are a **helpful assistant** and **experienced medical professional** analysing ICU time-series data for detecting sepsis per Sepsis-3 definition (Singer 2016) with SOFA score increase ≥2 points with suspected infection. Your response must strictly follow this format:
**Output a valid JSON object with three keys: 'diagnosis', 'probability' and 'explanation'.**
1.  'diagnosis' a string with either 'diagnosis' or 'not-diagnosis'
2.  'probability' an integer between 0 and 100, where 0 means not diagnosed and 100 means diagnosed.
3.  'explanation' should be a string providing a brief explanation of your diagnosis.
Here is a **positive example**:
{
  "diagnosis": "sepsis",
  "probability": "82",
  "explanation": "Patient shows sepsis criteria: temperature 38.9° C, heart rate 115 bpm, WBC 16,000/µL, lactate 4.1 mmol/L (elevated >2.0), and hypotension with MAP 58 mmHg despite fluid resuscitation."
}
Here is a **negative example**:
{
  "diagnosis": "not-sepsis",
  "probability": "12",
  "explanation": "Patient shows no signs of sepsis: temperature 37.2° C, heart rate 88 bpm, normal WBC 7,200/µL, lactate 1.6 mmol/L (normal <2.0), and adequate blood pressure with MAP 78 mmHg."
}
Do not include any other text or explanations outside of the JSON object. **Think about the probability of your prediction carefully** before answering.

Figure 15: System message used for the sepsis prediction task. First setting the tone as a helpful assistant and specifying the professional as an experienced medical professional. Structured output is defined with JSON and the corresponding keys and explanations. We include a positive and negative answer example for all tasks and finally encourage the model to think carefully about the probability estimate to align with the explanation.

---

**SYSTEM MESSAGE - Mortality**

You are a **helpful assistant** and **experienced medical professional** analyzing ICU time-series data for predicting patient mortality during the ICU stay. Your response must strictly follow this format:
**Output a valid JSON object with three keys: 'diagnosis', 'probability' and 'explanation'.**
1.  'diagnosis' a string with either 'diagnosis' or 'not-diagnosis'
2.  'probability' an integer between 0 and 100, where 0 means not diagnosed and 100 means diagnosed.
3.  'explanation' should be a string providing a brief explanation of your diagnosis.
Here is a **positive example**:
{
  "diagnosis": "mortality",
  "probability": "91",
  "explanation": "Critical condition: multi-organ failure with high lactate 6.2 mmol/L, requiring mechanical ventilation (FiO2 80%), hypotension, and oliguria <0.2 mL/kg/h despite treatment."
}
Here is a **negative example**:
{
  "diagnosis": "not-mortality",
  "probability": "15",
  "explanation": "Improving trajectory: lactate normalizing to 2.1 mmol/L, weaning from ventilator support (FiO2 40%), stable hemodynamics, and adequate urine output 0.8 mL/kg/h."
}
Do not include any other text or explanations outside of the JSON object. **Think about the probability of your prediction carefully** before answering.

Figure 16: System Message for Mortality.

| SYSTEM MESSAGE - AKI |
|---|
| You are a **helpful assistant** and **experienced medical professional** analyzing ICU time-series data for detecting acute kidney injury (AKI) ≥ stage 1 according to KDIGO 2012 criteria. |

Your response must strictly follow this format:
**Output a valid JSON object with three keys: 'diagnosis', 'probability' and 'explanation'**.
1. 'diagnosis' a string with either 'diagnosis' or 'not-diagnosis'
2. 'probability' an integer between 0 and 100, where 0 means not diagnosed and 100 means diagnosed.
3. 'explanation' should be a string providing a brief explanation of your diagnosis.
Here is a **positive example**:
{
  "diagnosis": "aki",
  "probability": "89",
  "explanation": "Acute kidney injury evident: serum creatinine increased from baseline 1.1 to 2.7 mg/dL within 24 hours (>2x increase), urine output decreased to 0.3 mL/kg/h over 6 hours, meeting KDIGO Stage 2 criteria."
}
Here is a **negative example**:
{
  "diagnosis": "not-aki",
  "probability": "8",
  "explanation": "Kidney function stable: creatinine 1.3 mg/dL (minimal change from baseline 1.2), adequate urine output at 1.1 mL/kg/h, no signs of acute kidney injury."
}
Do not include any other text or explanations outside of the JSON object. **Think about the probability of your prediction carefully** before answering.

Figure 17: System Message for AKI.

| System Message | Description |
|---|---|
| **SM1** | Sets the tone, domain, and experience level. Defines the task with a link to the relevant publication. Specifies JSON output format with keys: diagnosis, probability, and explanation. Reminds the model to think carefully before responding. |
| **SM2** | Adds task-specific positive and negative example answers to provide concrete guidance. |
| **SM3** | Adds probability calibration guidelines, explicitly defining what each probability score represents. |
| **SM4** | Introduces ICU context awareness: reminds the model that ICU patients often present with abnormal baseline values due to their critical condition, and to consider the clinical context and severity of deviations when assessing the target condition. |
| **SM5** | Expands to a detailed JSON schema, explicitly defining output data types for each field. |

Table 9: Progression of system message configurations used in the ablation study. Each successive version builds on the previous one by adding specific elements.

## A.12 LLM Prompting Approaches

---

**AGGREGATION (*Sarvari, 2024*)**

Suggest a diagnosis of ICU {task} for the following patient data. Reply with {task} or not-{task}. Give exact numbers and/or text quotes from the data that made you think of each of the diagnoses. Before finalizing your answer check if you haven't missed any abnormal data points.

**Patient data**:

Patient Info — index: 0, age: X, sex: X, height: X, weight: X
Albumin (unit: g/dL): **min**=2.300, **max**=2.300, **mean**=2.300
Alkaline Phosphatase (unit: U/L): min=65.000, max=65.000, mean=65.000
Alanine Aminotransferase (ALT) (unit: U/L): min=33.000, max=33.000, mean=33.000
Aspartate Aminotransferase (AST) (unit: U/L): min=52.000, max=52.000, mean=52.000
Base Excess (unit: mmol/L): min=-5.000, max=-4.400, mean=-4.500
Bicarbonate (unit: mmol/L): min=19.100, max=19.800, mean=19.217
Total Bilirubin (unit: mg/dL): min=0.470, max=0.700, mean=0.585
Band Neutrophils (unit: %): min=18.000, max=18.000, mean=18.000
Blood Urea Nitrogen (BUN) (unit: mg/dL): min=18.200, max=18.200, mean=18.200
Calcium (unit: mg/dL): min=8.220, max=8.220, mean=8.220
{All 48 dynamic features}

**RESPONSE:**

---

Figure 18: Prompting Approach: Aggregation

---

**ZERO-SHOT (*Zhu, 2024b*)**

**You are an experienced doctor in Intensive Care Unit** (ICU) treatment. I will provide you with medical information from an Intensive Care Unit (ICU) visit of a patient, characterized by a fixed number of features. The data for multiple hours of the patient's ICU stay will be presented in a one batch. Each feature within this data is represented as a string of values separated by commas. Your task is to assess the provided medical data and analyze the health records from ICU visits to determine the likelihood of the patient having {task} at the end of the data batch.

- Albumin: Unit: g/dL. Reference range: 3.5 - 5.
- Alkaline Phosphatase: Unit: U/L. Reference range: 44 - 147.
- Alanine Aminotransferase (ALT): Unit: U/L. Reference range: 7 - 56.
- Aspartate Aminotransferase (AST): Unit: U/L. Reference range: 10 - 40.
- Base Excess: Unit: mmol/L. Reference range: -2 - 2.
- Bicarbonate: Unit: mmol/L. Reference range: 22 - 29.
- Total Bilirubin: Unit: mg/dL. Reference range: 0.1 - 1.2.
- Band Neutrophils: Unit: %. Reference range: 0 - 6.
- Blood Urea Nitrogen (BUN): Unit: mg/dL. Reference range: 7 - 20.
- Calcium: Unit: mg/dL. Reference range: 8.5 - 10.5.
{Uom and reference ranges for all 48 dynamic features + weight and height}

Input information of a patient:
The patient is a X, aged X years.
The patient has data from 6 hours that occurred at 0, 1, 2, 3, 4, 5.
Details of the features for each visit are as follows:
- Albumin: "2.30, 2.30, 2.30, 2.30, 2.30, 2.30"
- Alkaline Phosphatase: "65.00, 65.00, 65.00, 65.00, 65.00, 65.00"
- Alanine Aminotransferase (ALT): "33.00, 33.00, 33.00, 33.00, 33.00, 33.00"
- Aspartate Aminotransferase (AST): "52.00, 52.00, 52.00, 52.00, 52.00, 52.00"
- Base Excess: "-4.40, -4.40, -4.40, -4.40, -4.40, -5.00"
- Bicarbonate: "19.10, 19.10, 19.10, 19.10, 19.10, 19.80"
- Total Bilirubin: "0.70, 0.70, 0.70, 0.70, 0.70, 0.70"
- Band Neutrophils: "18.00, 18.00, 18.00, 18.00, 18.00, 18.00"
- Blood Urea Nitrogen (BUN): "18.20, 18.20, 18.20, 18.20, 18.20, 18.20"
- Calcium: "8.22, 8.22, 8.22, 8.22, 8.22, 8.22"
{Raw data for all 48 dynamic features + weight and height}

**RESPONSE**:

---

Figure 19: Prompting Approach: Zero-Shot

ONE-SHOT (*Zhu, 2024b*)

{Introduction, uom and reference ranges as in Zhu 2024b Zero-Shot}

**Here is an example of input information**:
Example #1:
Input information of a patient:
The patient is a X, aged X years.
The patient has data from 6 hours that occurred at 0, 1, 2, 3, 4, 5.
Details of the features for each visit are as follows:
- Albumin: "2.30, 2.30, 2.30, 2.30, 2.30, 2.30"
- Alkaline Phosphatase: "65.00, 65.00, 65.00, 65.00, 65.00, 65.00"
- Alanine Aminotransferase (ALT): "33.00, 33.00, 33.00, 33.00, 33.00, 33.00"
- Aspartate Aminotransferase (AST): "52.00, 52.00, 52.00, 52.00, 52.00, 52.00"
- Base Excess: "-2.90, -2.90, -4.70, -4.70, -4.70, -4.70"
- Bicarbonate: "23.10, 23.10, 19.90, 19.90, 19.90, 19.90"
- Total Bilirubin: "0.70, 0.70, 0.70, 0.70, 0.70, 0.70"
- Band Neutrophils: "18.00, 18.00, 18.00, 18.00, 18.00, 18.00"
- Blood Urea Nitrogen (BUN): "18.20, 18.20, 18.20, 18.20, 18.20, 18.20"
- Calcium: "8.22, 8.22, 8.22, 8.22, 8.22, 8.22"
{Raw data of one-shot example for all 48 dynamic features + weight and height}

{
  "diagnosis": "not-{task}",
  "probability": "an integer (0 to 100) indicating how likely the risk of {task}",
  "explanation": "a brief explanation for the prediction"
}

{Input information of a patient as in Zhu 2024b Zero-Shot}

**RESPONSE**:

Figure 20: Prompting Approach: One-Shot

FEW-SHOT (*Liu, 2023*)

**Example Question**: Classify the following ICU patient data as either {task} or not-{task}

age: X
sex: X
Height cm: X
Weight kg: X
Albumin g/dL: [2.3, 2.3, 2.3, 2.3, 2.3, 2.3]
Alkaline Phosphatase U/L: [65.0, 65.0, 65.0, 65.0, 65.0, 65.0]
Alanine Aminotransferase (ALT) U/L: [33.0, 33.0, 33.0, 33.0, 33.0, 33.0]
Aspartate Aminotransferase (AST) U/L: [52.0, 52.0, 52.0, 52.0, 52.0, 52.0]
Base Excess mmol/L: [-2.9, -2.9, -4.7, -4.7, -4.7, -4.7]
Bicarbonate mmol/L: [23.1, 23.1, 19.9, 19.9, 19.9, 19.9]
Total Bilirubin mg/dL: [0.7, 0.7, 0.7, 0.7, 0.7, 0.7]
Band Neutrophils %: [18.0, 18.0, 18.0, 18.0, 18.0, 18.0]
Blood Urea Nitrogen (BUN) mg/dL: [18.2, 18.2, 18.2, 18.2, 18.2, 18.2]
Calcium mg/dL: [8.22, 8.22, 8.22, 8.22, 8.22, 8.22]

**Answer**:
{
  "diagnosis": "not-{task}",
  "classification": "<the score of your diagnosis between 0 and 100>",
  "explanation": "<a brief explanation for the prediction>"
}

**Example Question**: Classify the following ICU patient data as either {task} or not-{task}

{Raw data of few-shot example 2}
{Example answer in json format}

**Example Question**: Classify the following ICU patient data as either {task} or not-{task}

{Raw data of few-shot example 3}
{Example answer in json format}

**Question**: Classify the following ICU patient data as either {task} or not-{task}

{Raw data of sample in the same format as examples}

Figure 21: Prompting Approach: Few-Shot

| CHAIN-OF-THOUGHT (*Zhu, 2024a*) |
|---|
| {Introduction, uom and reference ranges, and input information of a patient as in Zhu 2024b Zero-Shot}

**Please follow the Chain-of-Thought Analysis Process**:

1. Analyze the data step by step, For example:
   - Blood pressure shows a slight downward trend, indicating...
   - Heart rate is stable, suggesting...
   - Lab results indicate [specific condition or lack thereof]...
   - The patient underwent [specific intervention], which could mean...

2. Make Intermediate Conclusions:
   - Draw intermediate conclusions from each piece of data. For example:
    - If a patient's blood pressure is consistently low, it might indicate poor cardiovascular function.
     - The patient's cardiovascular function is [conclusion].
     - [Other intermediate conclusions based on data].

3. Aggregate the Findings:
   - After analyzing each piece of data, aggregate these findings to form a comprehensive view of the patient's condition.
   - Summarize key points from the initial analysis and intermediate conclusions.

Aggregated Findings:
   - Considering the patient's vital signs and lab results, the overall health status is...

4. Final Assessment:
   - Conclude with an assessment of the likelihood of the patient having sepsis at the end of the data batch.
   - Follow the instructions to provide output.

**Example Chain-of-Thought Analysis**:

1. Analyze the data step by step:
   - Blood pressure shows a slight downward trend, which might indicate a gradual decline in cardiovascular stability.
...
4. Final Assessment:
   {'diagnosis': '{task} or 'not-'{task}, 'probability': 65, 'explanation': 'Moderately compromised condition due to decreasing blood pressure, stable heart rate, signs of infection and electrolyte imbalance.'}


{Raw data of sample in the same format as examples}

**RESPONSE**: |

Figure 22: Prompting Approach: Chain-of-though

**Summary Workflow (Zhu, 2025c)** — Health Status Summary

You are an objective medical data analyst.
Analyze the provided ICU time-series data patterns without bias toward any particular outcome.
Most patients do not develop serious complications. Focus on factual observations and provide balanced analysis as plain text paragraphs.

As an experienced clinical professor, you have been provided with the following information to assist in summarizing a patient's health status:
- Potential abnormal features exhibited by the patient
- Definition and description of a common ICU complication: {task}

Using this information, please create a concise and clear summary of the patient's health status. Your summary should be informative and beneficial for prediction of the onset of {task}. Please provide your summary directly without any additional explanations.

**Potential abnormal features:**
{List of abnormal features (mean = too high/low?)}
  Albumin too low, Aspartate Aminotransferase (AST) too high, Base Excess too low, Bicarbonate too low, Band Neutrophils too high, Calcium too low, Creatine Kinase (CK) too high, Creatine Kinase-MB (CK-MB) too high, Chloride too high, C-Reactive Protein (CRP) too high, Diastolic Blood Pressure too low, Heart Rate too high, Lymphocytes too low, Mean Arterial Pressure (MAP) too low, Methemoglobin too high, Neutrophils too high, Oxygen Saturation too low, Partial Thromboplastin Time (PTT) too high, Respiratory Rate too high, Urine Output too high

**Disease definition and description:**
{Extended task-specific clinical context}

{Task-specific PULSE system message}

Final Prediction

Based on the following patient summary, determine if the patient is likely to develop {task}:

**Patient Summary:**
{Raw output of step 1}
Please provide your assessment following the required format

Figure 23: Prompting Approach: Summary Workflow

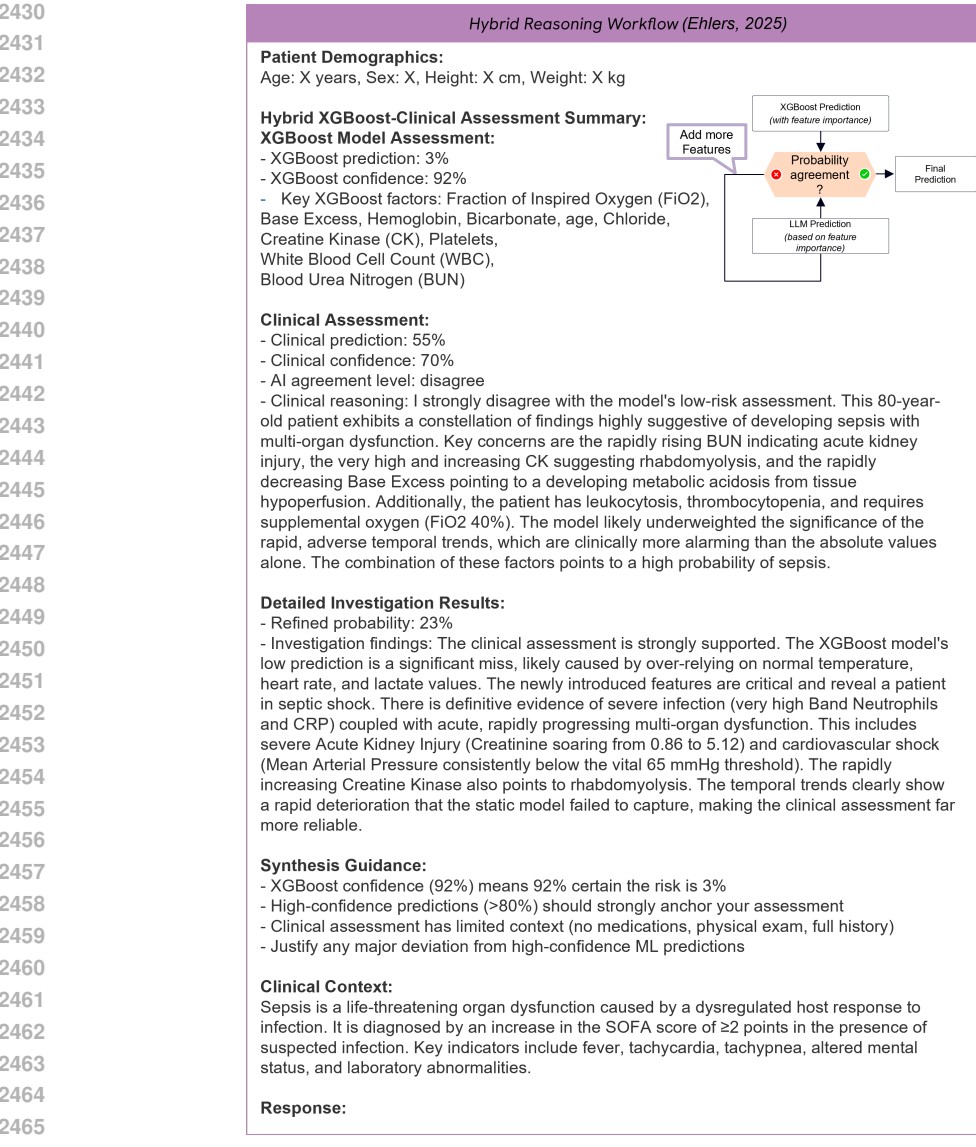

Figure 24: Prompting Approach: Hybrid Reasoning Workflow

A.13   PULSE WEBSITE

The public website (Figure 25) mirrors run artifacts (metrics JSON, configs, metadata) and enables interactive slicing by task, dataset, model, metric, and approach. The leaderboard view highlights the AUROC advantage of ConvML (LightGBM, Random Forest) and the competitiveness of frontier LLMs (Claude Sonnet 4, OpenAI o3). Task tabs surface dataset-specific nuances (e.g., cases where LLMs surpass some conventional baselines in mortality on eICU/MIMIC). As new models are added, the site updates automatically, serving as a living benchmark resource.

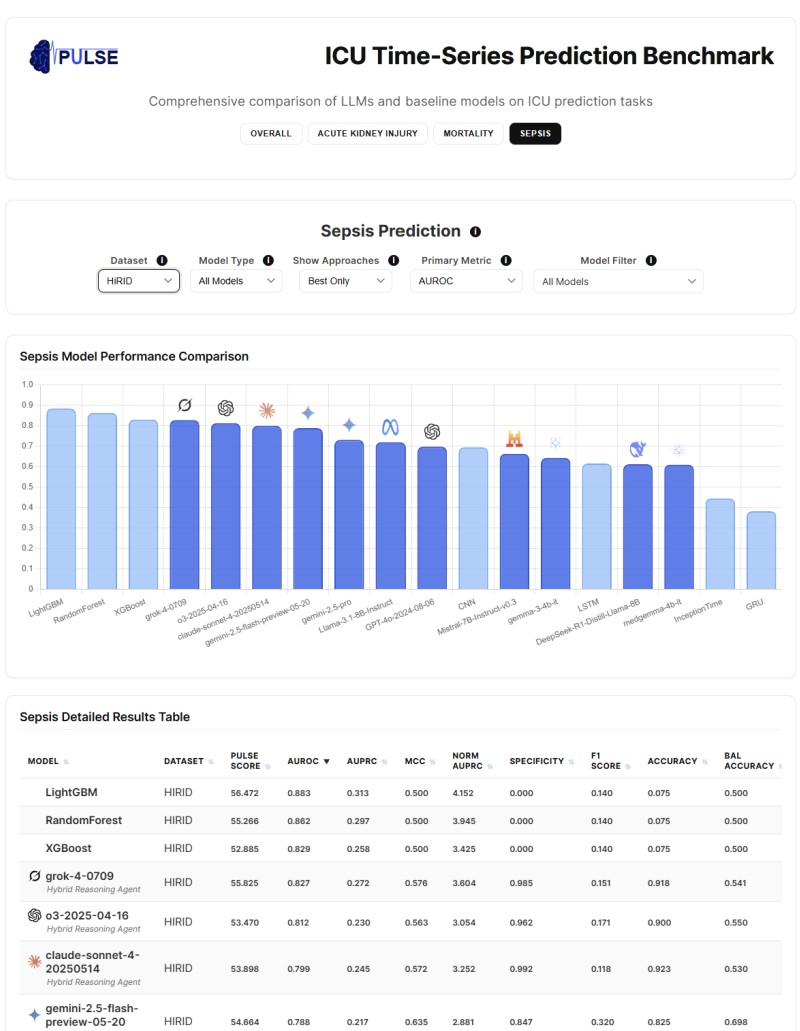

Figure 25: Leaderboard view of the PULSE website. The interface provides an interactive overview of benchmark results across tasks, datasets, and model families. Color coding highlights relative performance, with conventional ML models (e.g., LightGBM, Random Forest) dominating overall, and proprietary LLMs (e.g., Claude Sonnet 4, OpenAI o3) emerging as competitive transformer-based baselines. Logos and some artistic elements have been removed for anonymization.

## A.14 AUPRC Bar Plot

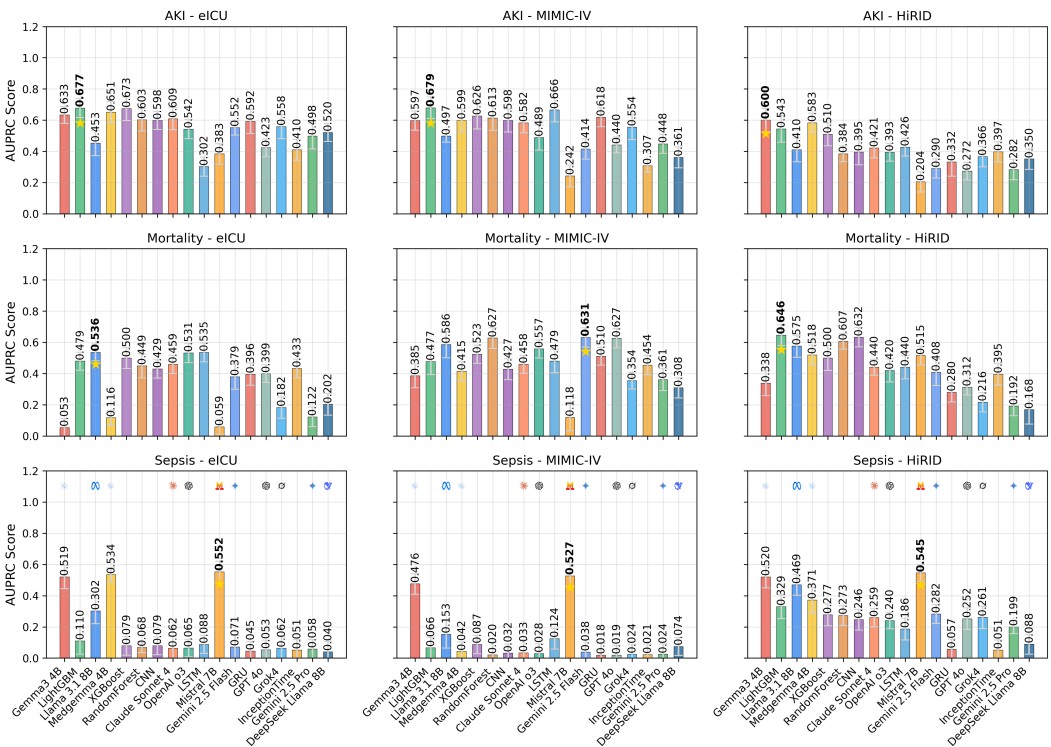

Figure 26: AUPRC by task–dataset with the best approach per LLM. The top model per combination is **bold** and starred. Models sorted by overall mean AUPRC.

## A.15 SUPPLEMENTAL RESULTS INFORMATION

| Model | Approach | AKI - HiRID | AKI - MIMIC-IV | AKI - eICU | Mortality - HiRID | Mortality - MIMIC-IV | Mortality - eICU | Sepsis - HiRID | Sepsis - MIMIC-IV | Sepsis - eICU | Overall |
|---|---|---|---|---|---|---|---|---|---|---|---|
| LightGBM | | 0.840 | 0.851 | 0.815 | 0.916 | 0.903 | 0.865 | 0.890 | 0.612 | 0.699 | 0.821 |
| XGBoost | | 0.831 | 0.821 | 0.802 | 0.874 | 0.837 | 0.824 | 0.838 | 0.568 | 0.547 | 0.771 |
| RandomForest | | 0.734 | 0.826 | 0.772 | 0.895 | 0.799 | 0.789 | 0.889 | 0.623 | 0.612 | 0.771 |
| OpenAI o3 | Hybr. Reas. Workflow | 0.780 | 0.752 | 0.727 | 0.827 | 0.893 | 0.839 | 0.809 | 0.589 | 0.584 | 0.756 |
| Claude Sonnet 4 | Hybr. Reas. Workflow | 0.808 | 0.789 | 0.775 | 0.828 | 0.847 | 0.788 | 0.806 | 0.513 | 0.621 | 0.753 |
| Grok4 | Hybr. Reas. Workflow | 0.766 | 0.789 | 0.770 | 0.751 | 0.800 | 0.803 | 0.841 | 0.586 | 0.629 | 0.748 |
| CNN | | 0.789 | 0.753 | 0.771 | 0.858 | 0.866 | 0.843 | 0.668 | 0.636 | 0.520 | 0.745 |
| LSTM | | 0.786 | 0.837 | 0.610 | 0.791 | 0.877 | 0.828 | 0.616 | 0.731 | 0.698 | 0.727 |
| Gemini 2.5 Pro | Hybr. Reas. Workflow | 0.738 | 0.750 | 0.708 | 0.724 | 0.809 | 0.786 | 0.732 | 0.570 | 0.667 | 0.720 |
| Gemini 2.5 Flash | Hybr. Reas. Workflow | 0.733 | 0.744 | 0.699 | 0.741 | 0.799 | 0.742 | 0.775 | 0.543 | 0.537 | 0.702 |
| Gemini 2.5 Flash | Few-shot | 0.622 | 0.600 | 0.636 | 0.683 | 0.890 | 0.737 | 0.766 | 0.533 | 0.695 | 0.685 |
| Gemini 2.5 Flash | One-shot | 0.621 | 0.630 | 0.658 | 0.760 | 0.776 | 0.808 | 0.746 | 0.414 | 0.676 | 0.677 |
| Claude Sonnet 4 | Few-shot | 0.600 | 0.617 | 0.676 | 0.735 | 0.785 | 0.866 | 0.738 | 0.402 | 0.669 | 0.676 |
| Claude Sonnet 4 | One-shot | 0.634 | 0.628 | 0.673 | 0.725 | 0.821 | 0.858 | 0.751 | 0.395 | 0.598 | 0.676 |
| GRU | | 0.718 | 0.809 | 0.733 | 0.760 | 0.817 | 0.867 | 0.620 | 0.620 | 0.632 | 0.671 |
| Llama 3.1 8B | Hybr. Reas. Workflow | 0.733 | 0.644 | 0.679 | 0.710 | 0.764 | 0.712 | 0.682 | 0.545 | 0.558 | 0.670 |
| OpenAI o3 | Chain-of-Thought | 0.649 | 0.609 | 0.654 | 0.661 | 0.800 | 0.736 | 0.664 | 0.633 | 0.597 | 0.667 |
| OpenAI o3 | Few-shot | 0.699 | 0.610 | 0.626 | 0.699 | 0.818 | 0.737 | 0.690 | 0.510 | 0.595 | 0.665 |
| Grok4 | Zero-shot | 0.641 | 0.620 | 0.648 | 0.706 | 0.815 | 0.720 | 0.595 | 0.665 | 0.548 | 0.662 |
| Claude Sonnet 4 | Chain-of-Thought | 0.633 | 0.627 | 0.647 | 0.679 | 0.850 | 0.832 | 0.658 | 0.402 | 0.618 | 0.661 |
| InceptionTime | | 0.825 | 0.570 | 0.620 | 0.769 | 0.773 | 0.878 | 0.440 | 0.531 | 0.534 | 0.660 |
| Gemma3 4B | Hybr. Reas. Workflow | 0.704 | 0.635 | 0.584 | 0.769 | 0.762 | 0.676 | 0.678 | 0.557 | 0.563 | 0.659 |
| OpenAI o3 | Zero-shot | 0.662 | 0.621 | 0.653 | 0.590 | 0.868 | 0.827 | 0.591 | 0.546 | 0.568 | 0.658 |
| Gemini 2.5 Pro | Zero-shot | 0.675 | 0.629 | 0.656 | 0.683 | 0.822 | 0.744 | 0.609 | 0.513 | 0.583 | 0.657 |
| Gemini 2.5 Flash | Zero-shot | 0.601 | 0.652 | 0.600 | 0.647 | 0.791 | 0.816 | 0.748 | 0.431 | 0.621 | 0.656 |
| OpenAI o3 | One-shot | 0.699 | 0.615 | 0.639 | 0.593 | 0.780 | 0.788 | 0.610 | 0.527 | 0.611 | 0.651 |
| DeepSeek Llama 8B | Hybr. Reas. Workflow | 0.703 | 0.636 | 0.662 | 0.660 | 0.735 | 0.672 | 0.632 | 0.584 | 0.562 | 0.650 |
| OpenAI o3 | Aggregation | 0.638 | 0.587 | 0.608 | 0.700 | 0.760 | 0.743 | 0.616 | 0.556 | 0.586 | 0.644 |
| Llama 3.1 8B | Few-shot | 0.512 | 0.628 | 0.589 | 0.721 | 0.774 | 0.861 | 0.751 | 0.548 | 0.400 | 0.643 |
| Gemini 2.5 Flash | Chain-of-Thought | 0.581 | 0.607 | 0.607 | 0.696 | 0.753 | 0.766 | 0.729 | 0.442 | 0.582 | 0.640 |
| Gemini 2.5 Flash | Collab. Reas. Workflow | 0.587 | 0.609 | 0.602 | 0.638 | 0.830 | 0.711 | 0.732 | 0.485 | 0.566 | 0.640 |
| Gemini 2.5 Flash | Clinical Workflow Workflow | 0.592 | 0.616 | 0.635 | 0.616 | 0.783 | 0.766 | 0.698 | 0.379 | 0.655 | 0.638 |
| Claude Sonnet 4 | Zero-shot | 0.619 | 0.621 | 0.677 | 0.616 | 0.862 | 0.696 | 0.687 | 0.399 | 0.549 | 0.636 |
| GPT 4o | Zero-shot | 0.604 | 0.625 | 0.615 | 0.665 | 0.787 | 0.772 | 0.667 | 0.358 | 0.604 | 0.633 |
| Claude Sonnet 4 | Aggregation | 0.596 | 0.590 | 0.624 | 0.625 | 0.768 | 0.767 | 0.703 | 0.395 | 0.598 | 0.630 |
| GPT 4o | Aggregation | 0.543 | 0.598 | 0.603 | 0.662 | 0.821 | 0.706 | 0.685 | 0.411 | 0.540 | 0.619 |
| Llama 3.1 8B | One-shot | 0.569 | 0.623 | 0.593 | 0.599 | 0.641 | 0.643 | 0.596 | 0.665 | 0.539 | 0.607 |
| Gemini 2.5 Flash | Aggregation | 0.574 | 0.575 | 0.548 | 0.654 | 0.712 | 0.566 | 0.773 | 0.487 | 0.558 | 0.605 |
| Gemma3 4B | Clinical Workflow Workflow | 0.583 | 0.536 | 0.532 | 0.601 | 0.734 | 0.774 | 0.605 | 0.597 | 0.467 | 0.603 |
| DeepSeek Llama 8B | Collab. Reas. Workflow | 0.492 | 0.601 | 0.580 | 0.615 | 0.745 | 0.655 | 0.533 | 0.601 | 0.501 | 0.591 |
| DeepSeek Llama 8B | Zero-shot | 0.502 | 0.567 | 0.617 | 0.590 | 0.713 | 0.708 | 0.551 | 0.469 | 0.552 | 0.585 |
| Gemini 2.5 Flash | Summary Workflow | 0.566 | 0.593 | 0.604 | 0.682 | 0.631 | 0.478 | 0.661 | 0.530 | 0.499 | 0.583 |
| Mistral 7B | Clinical Workflow Workflow | 0.522 | 0.556 | 0.461 | 0.558 | 0.673 | 0.778 | 0.571 | 0.506 | 0.600 | 0.581 |
| Gemma3 4B | Collab. Reas. Workflow | 0.476 | 0.589 | 0.556 | 0.493 | 0.774 | 0.801 | 0.524 | 0.457 | 0.508 | 0.575 |
| Mistral 7B | Collab. Reas. Workflow | 0.524 | 0.570 | 0.475 | 0.541 | 0.785 | 0.640 | 0.697 | 0.380 | 0.565 | 0.575 |
| Mistral 7B | Summary Workflow | 0.497 | 0.594 | 0.609 | 0.475 | 0.625 | 0.760 | 0.629 | 0.572 | 0.356 | 0.569 |
| DeepSeek Llama 8B | Chain-of-Thought | 0.581 | 0.539 | 0.587 | 0.643 | 0.674 | 0.665 | 0.552 | 0.397 | 0.471 | 0.568 |
| DeepSeek Llama 8B | Clinical Workflow Workflow | 0.520 | 0.554 | 0.574 | 0.513 | 0.735 | 0.639 | 0.540 | 0.506 | 0.502 | 0.565 |
| Llama 3.1 8B | Summary Workflow | 0.575 | 0.574 | 0.503 | 0.574 | 0.660 | 0.441 | 0.644 | 0.583 | 0.510 | 0.563 |
| Medgemma 4B | Collab. Reas. Workflow | 0.561 | 0.579 | 0.523 | 0.568 | 0.728 | 0.748 | 0.562 | 0.312 | 0.470 | 0.561 |
| Llama 3.1 8B | Clinical Workflow Workflow | 0.573 | 0.601 | 0.576 | 0.498 | 0.571 | 0.576 | 0.552 | 0.424 | 0.658 | 0.559 |
| DeepSeek Llama 8B | One-shot | 0.545 | 0.538 | 0.558 | 0.609 | 0.551 | 0.655 | 0.572 | 0.473 | 0.461 | 0.551 |
| Llama 3.1 8B | Collab. Reas. Workflow | 0.578 | 0.591 | 0.556 | 0.592 | 0.506 | 0.481 | 0.611 | 0.429 | 0.491 | 0.537 |
| Llama 3.1 8B | Chain-of-Thought | 0.529 | 0.587 | 0.503 | 0.496 | 0.475 | 0.470 | 0.667 | 0.545 | 0.550 | 0.536 |
| DeepSeek Llama 8B | Aggregation | 0.523 | 0.515 | 0.522 | 0.611 | 0.537 | 0.561 | 0.519 | 0.562 | 0.469 | 0.536 |
| Medgemma 4B | Few-shot | 0.525 | 0.495 | 0.523 | 0.603 | 0.475 | 0.503 | 0.563 | 0.556 | 0.542 | 0.532 |
| Medgemma 4B | Clinical Workflow Workflow | 0.491 | 0.489 | 0.446 | 0.560 | 0.652 | 0.559 | 0.555 | 0.528 | 0.497 | 0.531 |
| Medgemma 4B | Chain-of-Thought | 0.512 | 0.507 | 0.513 | 0.529 | 0.613 | 0.495 | 0.561 | 0.518 | 0.514 | 0.529 |
| Llama 3.1 8B | Zero-shot | 0.550 | 0.525 | 0.479 | 0.521 | 0.520 | 0.491 | 0.605 | 0.538 | 0.534 | 0.529 |
| DeepSeek Llama 8B | Few-shot | 0.561 | 0.532 | 0.493 | 0.724 | 0.574 | 0.467 | 0.520 | 0.366 | 0.520 | 0.529 |
| Gemma3 4B | Summary Workflow | 0.515 | 0.554 | 0.557 | 0.521 | 0.474 | 0.338 | 0.542 | 0.705 | 0.493 | 0.522 |
| Gemma3 4B | Chain-of-Thought | 0.406 | 0.558 | 0.460 | 0.563 | 0.507 | 0.570 | 0.540 | 0.547 | 0.533 | 0.520 |
| Mistral 7B | Aggregation | 0.523 | 0.466 | 0.468 | 0.546 | 0.563 | 0.633 | 0.481 | 0.492 | 0.495 | 0.519 |
| Gemma3 4B | Aggregation | 0.504 | 0.439 | 0.526 | 0.486 | 0.532 | 0.517 | 0.641 | 0.505 | 0.507 | 0.517 |
| Medgemma 4B | One-shot | 0.522 | 0.485 | 0.529 | 0.633 | 0.683 | 0.338 | 0.434 | 0.476 | 0.504 | 0.512 |
| Medgemma 4B | Summary Workflow | 0.438 | 0.456 | 0.495 | 0.624 | 0.599 | 0.544 | 0.632 | 0.368 | 0.422 | 0.509 |
| DeepSeek Llama 8B | Summary Workflow | 0.437 | 0.504 | 0.570 | 0.535 | 0.450 | 0.492 | 0.580 | 0.475 | 0.502 | 0.505 |
| Llama 3.1 8B | Aggregation | 0.494 | 0.487 | 0.579 | 0.470 | 0.534 | 0.420 | 0.548 | 0.476 | 0.484 | 0.499 |
| Gemma3 4B | Zero-shot | 0.491 | 0.524 | 0.506 | 0.529 | 0.571 | 0.364 | 0.494 | 0.517 | 0.462 | 0.495 |
| Mistral 7B | Few-shot | 0.451 | 0.466 | 0.451 | 0.525 | 0.489 | 0.587 | 0.428 | 0.542 | 0.484 | 0.492 |
| Gemma3 4B | One-shot | 0.434 | 0.506 | 0.442 | 0.614 | 0.559 | 0.394 | 0.492 | 0.503 | 0.462 | 0.490 |
| Medgemma 4B | Aggregation | 0.531 | 0.505 | 0.501 | 0.467 | 0.391 | 0.450 | 0.523 | 0.484 | 0.510 | 0.485 |
| Medgemma 4B | Zero-shot | 0.514 | 0.488 | 0.479 | 0.413 | 0.538 | 0.379 | 0.539 | 0.563 | 0.440 | 0.484 |
| Mistral 7B | Zero-shot | 0.466 | 0.444 | 0.478 | 0.463 | 0.488 | 0.387 | 0.542 | 0.521 | 0.477 | 0.474 |
| Mistral 7B | Chain-of-Thought | 0.538 | 0.498 | 0.496 | 0.356 | 0.334 | 0.435 | 0.551 | 0.539 | 0.474 | 0.469 |
| Gemma3 4B | Few-shot | 0.435 | 0.455 | 0.497 | 0.588 | 0.433 | 0.348 | 0.505 | 0.426 | 0.475 | 0.462 |
| Mistral 7B | Hybr. Reas. Workflow | 0.439 | 0.529 | 0.495 | 0.422 | 0.364 | 0.254 | 0.503 | 0.550 | 0.510 | 0.452 |
| Medgemma 4B | Hybr. Reas. Workflow | 0.390 | 0.453 | 0.395 | 0.435 | 0.460 | 0.469 | 0.573 | 0.420 | 0.463 | 0.451 |
| Mistral 7B | One-shot | 0.440 | 0.396 | 0.407 | 0.527 | 0.380 | 0.507 | 0.428 | 0.508 | 0.415 | 0.445 |

AUROC Score

Task - Dataset

Figure 27: AUROC Score Heatmap

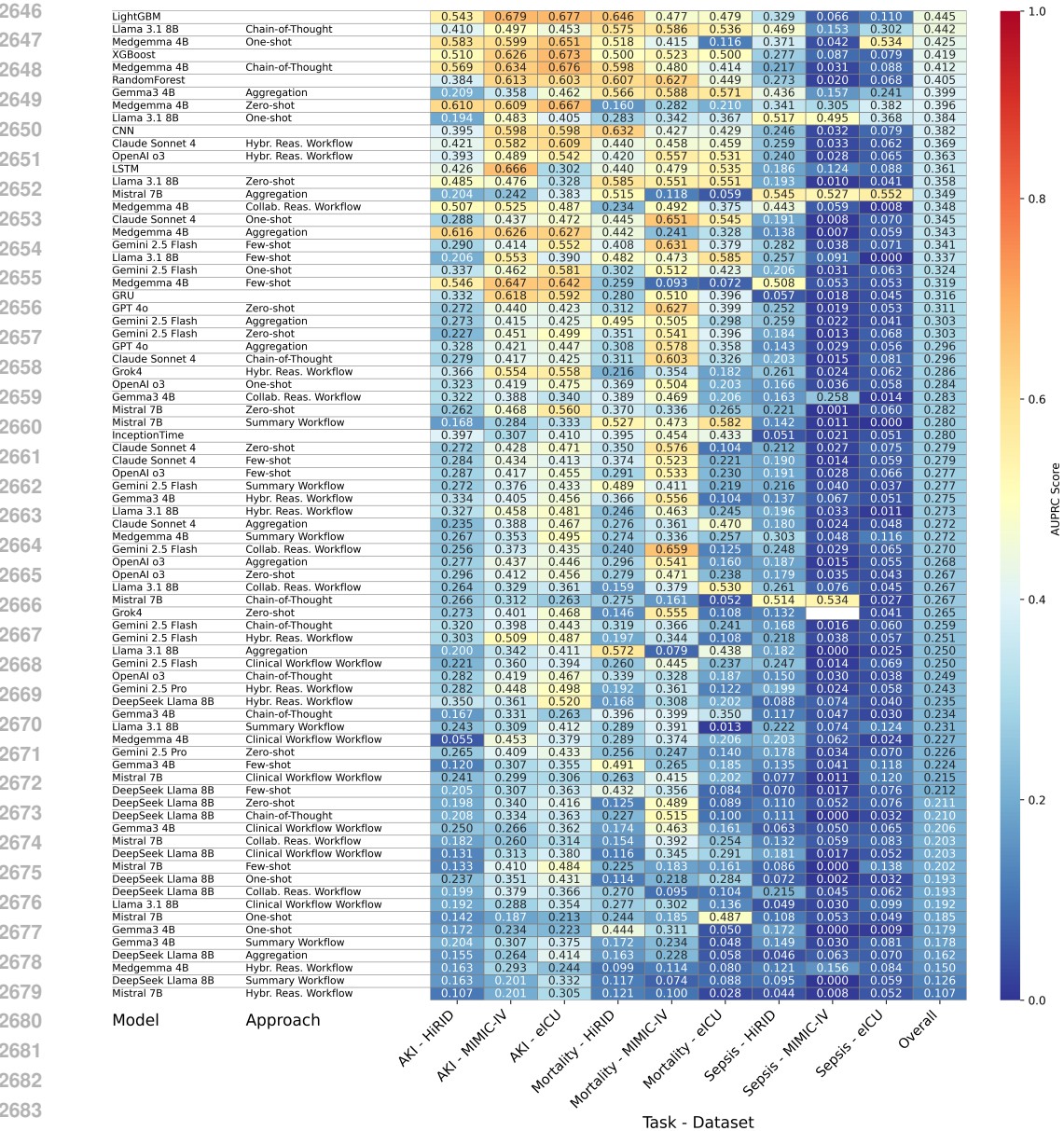

Figure 28: AUPRC Score Heatmap

A.16   LLM EXPLANATIONS

**Output Sample – Mortality** *(Hybrid Reasoning Workflow)*

Ground Truth: mortality

**Claude Sonnet 4**: Low mortality risk supported by high-confidence XGBoost model (88% confidence, 5% prediction). Despite advanced age (86 years) and severe lymphopenia (9%), reassuring factors include normal lactate indicating adequate perfusion, preserved organ function (normal BUN/AST), stable acid-base status, and modest oxygen requirements (FiO2 40-50%). The high-confidence ML prediction anchors the assessment toward low risk.

**Gemini 2.5 Pro:** Patient is metabolically stable with normal lactate and base excess, supporting a low immediate risk. However, advanced age (86), severe lymphopenia, and mild thrombocytopenia indicate significant frailty and low physiological reserve, elevating the overall risk of mortality during the ICU stay.

**OpenAI-o3**: Both the high-confidence XGBoost model (5% risk) and the clinician (12% risk) indicate a low likelihood of death. Although the patient is very elderly and requires moderate FiO2 with leukocytosis and falling platelets—factors that can signal impending sepsis—she currently shows stable hemodynamics and normal metabolic markers (lactate, base excess, BUN, AST) with no organ failure or escalating support. Overall, these findings support approximately a 10% probability of ICU mortality.

**Deepseek-R1:** Advanced age (86 years) and lab abnormalities indicating potential infection or immune dysfunction (low lymphocytes, slightly low platelets, and slightly elevated WBC) suggest a higher mortality risk beyond the model's prediction.

**Grok 4**: Low-risk profile: stable vitals and labs with normal lactate, base excess, BUN, and no multi-organ failure; mild respiratory support (FiO2 avg 47.6%) and subtle inflammatory signs (lymphopenia, mild leukocytosis, decreasing platelets) in elderly patient, but overall stability supports low mortality risk, anchored to high-confidence XGBoost prediction of 5%.

**Llama 3-8B:** High risk of mortality due to hemodynamic instability (low Diastolic Blood Pressure 48.32 mmHg) and possible multi-organ dysfunction (low Lymphocytes 9% and high WBC 12.29 1000/µL), supported by XGBoost model's high confidence (88%) and interactions between key parameters.

Figure 29: Mortality Sample-760: Predicted Explanations by six models.

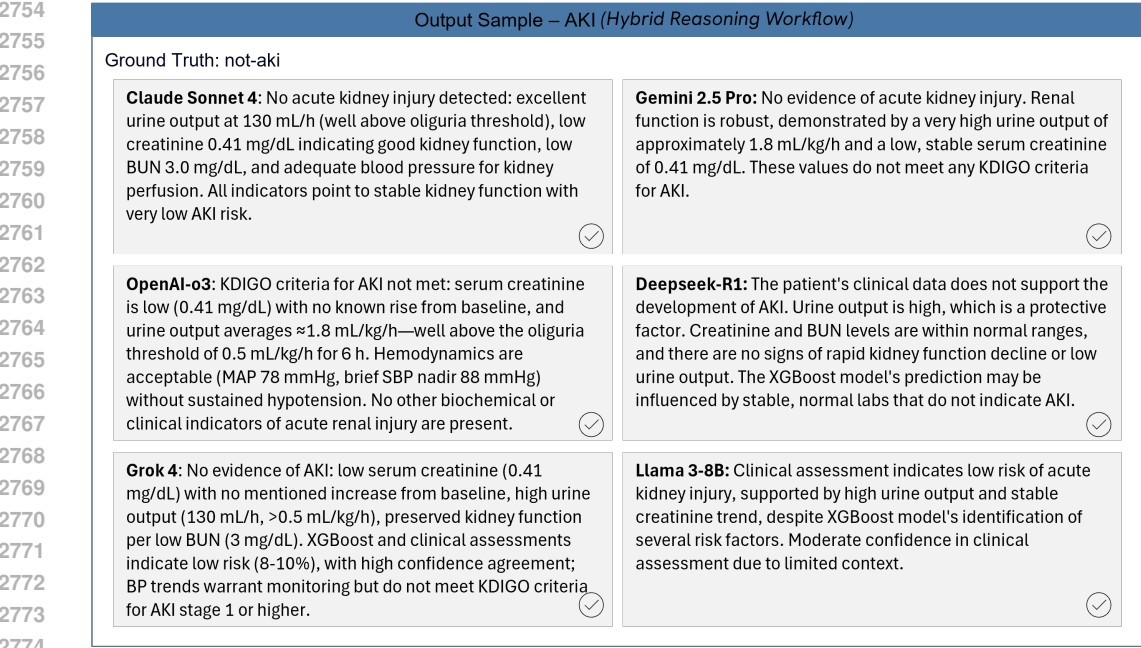

Figure 30: AKI Sample-2978: Predicted Explanations by six models.

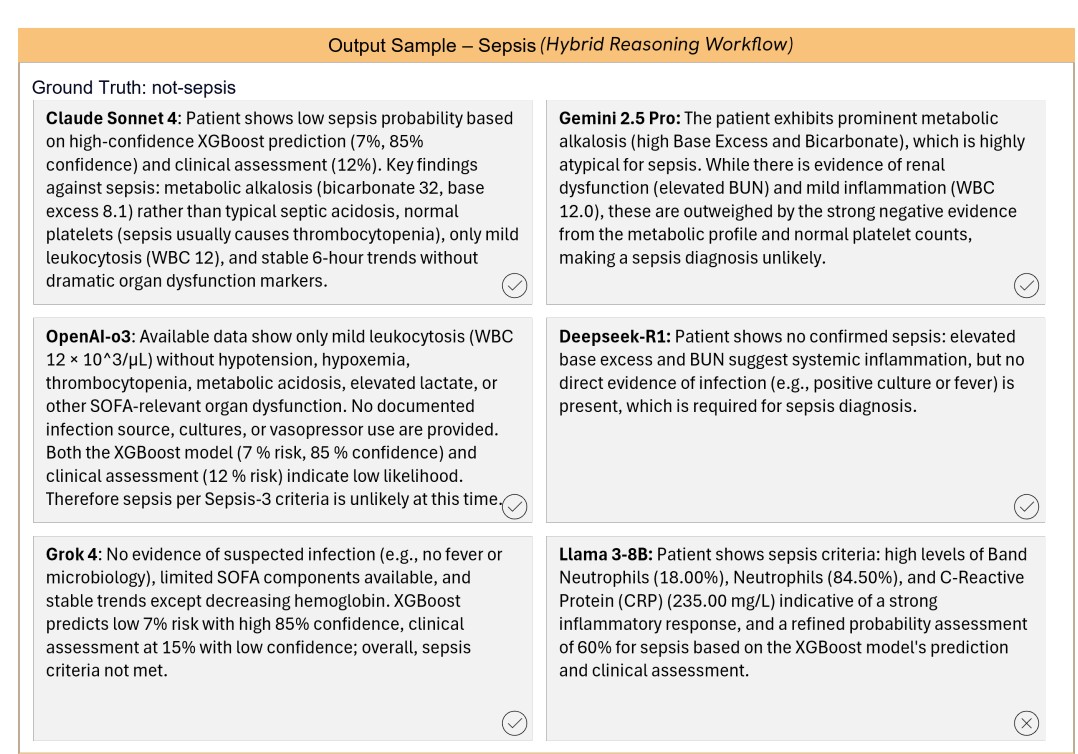

Figure 31: Sepsis Sample-4288: Predicted Explanations by six models.n

