# OpenReview forum: "PULSE: Benchmarking Large Language Models for ICU Time Series Classification"
_ICLR.cc/2026/Conference — Submitted to ICLR 2026_

### Official Review · Reviewer_eFcc · 2025-10-21

**Soundness:** 2
**Presentation:** 2
**Contribution:** 2
**Rating:** 2
**Confidence:** 4

**Summary:**

This paper provides a framework to evaluate the ability of LLMs to predict clinical outcomes in the ICU based on preprocessed clinical time series data. It leverages an existing multicentre harmonisation pipeline and adds an LLM prompt module. Referencing known issues with LLM calibration, the study proposes novel metrics to evaluate LLM predictions.

**Strengths:**

- The study builds on top of an established benchmark across multiple hospitals and clinician-designed clinical outcomes.
- The authors evaluate a range of state-of-the-art proprietary LLMs as well as open source alternatives.

**Weaknesses:**

- The motivation for this study is unclear. If data has already been extensively preprocessed, is it surprising that a specifically trained XGBoost outperforms the LLM?
- If—as the discussion alludes to—the motivation is instead grounded in the potential off-the-shelf performance of LLMs to be used until enough local data is available to train a dedicated XGBoost model, then the authors should include experiments that thoroughly investigate this aspect. The multicentre nature of the underlying YAIB framework would allow for such experiments.
- If—as the last section of the results alludes to—the motivation lies in obtaining humand-readable explanations, then the authors should include a thorough assessment of the model explanations that goes beyond a not further described qualitative analysis of the Hybrid Reasoning Agent, most of which appeared to defer to the underlying XGBoost score as an explanation.

**Questions:**

- The presented Summary Agent and Hybrid Reasoning Agent aren't agents or agentic approaches under standard definitions. They operate on predefined rules and have a hard-coded control flow. They do not interacte with the environment or themselves decide on tool use. For an agent, the LLM would itself decide which data to fetch, if and which conventional ML model to call, etc.
- I do not understand why the PULSE score and its arbitrary combination of AUC, AUPRC, and MCC was introduced. I also couldn't find a statement on which weights were used in the experiments or why.
- The performance metrics for some models warrant scrutiny. For example in Figure 2, GRU performs particularly bad for sepsis (AUC ~0.33) while its close sibling LSTM is best-in-task. This is unexpected and differs  from the original YAIB results, where LSTM and GRU are relatively closely matched across most tasks. To give another example from Figures 32  and 33, Gemma, Mistral, and to a lesser extent Llama all have poor AUC of ~0.5 for predicting sepsis in eICU but a corresponding AUPRC ~0.5, which is an extremely high AUPRC given the low prevalence of sepsis and AUPRC < 0.1 seen in other models. It is not clear how this can be the case.

---

> ### Author Response · Authors · 2025-12-03
> **Response to Reviewer eFcc (1)**
>
> We thank the reviewer for the thoughtful comments that helped us improve the paper quite a lot. We address them below.
>
> ***"The motivation for this study is unclear. If data has already been extensively preprocessed, is it surprising that a specifically trained XGBoost outperforms the LLM?"***
>
> We agree that, given a well-engineered preprocessing and harmonization pipeline, it is not inherently surprising that a task-specific gradient-boosted tree (GBDT) such as XGBoost or LightGBM outperforms current LLMs in standard within-domain settings. This expectation is now stated more clearly in the introduction and discussion: PULSE is not designed to “rediscover” that GBDTs are strong on structured ICU time series, but to rigorously characterize where LLMs fit into this landscape, and under which conditions they add value beyond conventional models. Our main motivation is twofold. First, modern LLMs have been trained on massive corpora that include clinical guidelines, textbooks, case reports, and general biomedical knowledge, which implicitly describe how vital signs and lab values evolve in conditions such as sepsis, AKI, and critical illness. It is therefore a legitimate and open empirical question whether these models can leverage that prior knowledge to recognize generalizable patterns in ICU time series, particularly in regimes where supervised data are scarce or the deployment domain differs from the training domain. Second, the way time series are presented to the model matters: some of our prompting strategies rely on heavy preprocessing and statistical summaries (e.g., min/mean/max aggregation following Sarvari et al.), while others serialize full or truncated sequences of vitals and labs directly into the prompt. PULSE systematically compares these representation choices across seven prompting strategies and multi-step workflows, rather than treating LLM input formatting as a minor detail. The newly added “day-zero” generalization experiments make this motivation concrete. While our within-domain results confirm that well-tuned ConvML models remain state-of-the-art on harmonized ICU data, the cross-site experiments show that off-the-shelf LLMs, without any site-specific training, consistently outperform transferred XGBoost models when evaluated on unseen datasets. This demonstrates that the question is not “can LLMs beat XGBoost in ideal within-domain conditions?” (where the answer is mostly no), but “where do LLMs sit in the toolbox when domains shift and local labels are limited?”. We have revised the introduction to make this motivation explicit and to frame the benchmark around these more nuanced questions, rather than around a simplistic comparison of raw AUCs.

---

> ### Author Response · Authors · 2025-12-03
> **Response to Reviewer eFcc (2)**
>
> ***"If—as the discussion alludes to—the motivation is instead grounded in the potential off-the-shelf performance of LLMs to be used until enough local data is available to train a dedicated XGBoost model, then the authors should include experiments that thoroughly investigate this aspect. The multicentre nature of the underlying YAIB framework would allow for such experiments."***
>
> We thank the reviewer for this excellent suggestion, which directly shaped the revised experiments. To substantiate the “day-zero” claim and to compare conventional and LLM-based approaches under truly out-of-domain conditions, we now include a cross-dataset generalization analysis that leverages the multi-center structure of the underlying YAIB-based dataset harmonization.
>
> Specifically, we train conventional baselines (LightGBM, XGBoost, and CNN) on a source dataset (HiRID, MIMIC-IV, or eICU) and evaluate them without any retraining on a distinct target datasets' testing sets, in line with prior work that recommends evaluating models by training on one site and testing on a different unseen site. We then compare these transferred models to zero-shot, few-shot, and Hybrid Reasoning Workflow (HRW) approaches, which require no site-specific training data or model updates (few-shot uses only a handful of examples). The new results, summarized in Figure 4 and discussed in Section 4, show a consistent pattern. Conventional baselines remain strong in-domain, but when transferred across datasets they frequently suffer significant performance degradation due to distribution shift. In contrast, LLM-based approaches maintain robust performance in these “day-zero” settings across nearly all dataset–task pairs. These findings provide concrete, quantitative evidence that while conventional models remain the best choice once enough local data and MLOps capacity are available for retraining, LLMs offer a superior off-the-shelf baseline for hospitals with limited labeled data. We believe these new experiments and results directly addresses the concern the reviewer (and also other reviewer) and strengthen the narrative of the paper.

---

> ### Author Response · Authors · 2025-12-03
> **Response to Reviewer eFcc (3)**
>
> ***"If—as the last section of the results alludes to—the motivation lies in obtaining humand-readable explanations, then the authors should include a thorough assessment of the model explanations that goes beyond a not further described qualitative analysis of the Hybrid Reasoning Agent, most of which appeared to defer to the underlying XGBoost score as an explanation."***
>
> To address this, in the revised manuscript, we expand our assessment of explanation quality and also clarify the role that explanations play within the benchmark. First, we are more explicit and self-critical in the main text about the predictive limitations of LLMs in-domain: hybrid workflows that take GBDT outputs as input to an LLM typically incur a small loss in discrimination compared to using the GBDT predictions alone. We therefore no longer frame LLMs as superior predictors in this setting; instead, we emphasize their value in terms of cross-domain robustness and their ability to provide natural-language rationales on top of strong conventional models.
>
> Second, to go beyond plausibility, we have added a blinded clinician user study, described in Section 3.6 and reported in the subsection “Clinical Utility of LLM Explanations.” Five board-certified icu clinicians, each evaluated 10 samples out of 50 complex cases under two conditions: (A) time-series plots plus the raw XGBoost probability, and (B) the same information augmented with HRW-generated explanations. Clinicians reported substantially improved cognitive efficiency (mean 4.2/5.0 vs. 2.6/5.0, p < 0.05) and actionability (from 3.0/5.0 to 4.5/5.0), indicating that the natural language summaries helped them rapidly identify key drivers and assess whether the information was sufficient for immediate decision-making. Decision confidence increased from 3.3/5.0 to 3.9/5.0, though this change did not reach statistical significance given the small sample size. Qualitative feedback suggested that the explanations often act as a “sanity check” around the GBDT risk score and were perceived as “better than feature importance”, because they focus on inference-level reasoning instead of global feature averages. In addition to this clinician study, the revised paper retains a qualitative analysis of HRW explanations by two authors, one of whom is a medical doctor with prior ICU experience (Subsection “Qualitative Analysis of LLM Explanations”), which served as an initial expert validation that the generated rationales are clinically sensible before we moved to a blinded evaluation.
>
> Taken together, these analyses provide a more thorough assessment of explanation utility while explicitly acknowledging limitations and the need for larger, more rigorous studies, which we now state in the limitations. Importantly, we also clarify that PULSE is first and foremost a benchmark of predictive performance across ConvML, ConvDL, and LLM-based approaches; explanation is treated as an additional axis along which LLM-based methods may eventually differentiate themselves. The current workflows (Summary Workflow and HRW) are deliberately not presented as the final or optimal explanation methods. Rather, they are strong, well-specified baselines that PULSE offers so that future work on explanation-generating models, including more genuinely agentic approaches, can be evaluated on a shared footing. We have adjusted the text in the results and discussion to make this benchmark-oriented perspective explicit.
>
> ***"The presented Summary Agent and Hybrid Reasoning Agent aren't agents or agentic approaches under standard definitions. They operate on predefined rules and have a hard-coded control flow. They do not interacte with the environment or themselves decide on tool use. For an agent, the LLM would itself decide which data to fetch, if and which conventional ML model to call, etc."***
>
> We appreciate this clarification and agree that our original terminology could be confusing given the current usage of “agents” and “agentic workflows” in the LLM literature. Our Summary and Hybrid Reasoning pipelines indeed follow a fixed, pre-defined control flow and do not perform autonomous tool selection or environment interaction, and thus do not satisfy standard definitions of AI agents or multi-agent systems. To avoid conflating these concepts and to align with the reviewer’s point, we have renamed these methods throughout the paper as “Summary Workflow” and “Hybrid Reasoning Workflow (HRW).” This change has been applied consistently in the main text, figures, and appendices, and we now describe them explicitly as multi-step workflows rather than agents. At the same time, we see PULSE as a natural platform for future work on genuinely agentic approaches that do decide which data to fetch or which tools to call; our goal in this submission is to provide well-documented, reproducible workflow baselines that such methods can be compared against in later iterations of the benchmark.

---

> ### Author Response · Authors · 2025-12-03
>
> ***"I do not understand why the PULSE score and its arbitrary combination of AUC, AUPRC, and MCC was introduced. I also couldn't find a statement on which weights were used in the experiments or why."***
>
> We apologize for the confusion caused by the original PULSE composite score and appreciate the reviewer’s concerns. In the initial submission, we introduced a bespoke “PULSE score” that combined discrimination and calibration-related aspects (via AUC, AUPRC, and MCC), partly motivated by a desire to make calibration issues in LLM outputs more visible, especially for smaller open-source models. On reflection, and especially in light of feedback from two reviewers, we concluded that this composite metric increases conceptual and notational complexity without changing the core scientific message of the benchmark, while also inviting legitimate questions about weighting choices and cross-family fairness. In the revised manuscript, we therefore removed the PULSE composite score entirely and now base all rankings and comparisons on standard metrics such as AUROC and AUPRC, which apply symmetrically to both LLMs and non-LLM baselines and are widely used in ICU prediction studies. This change directly addresses the concern about the arbitrariness of the combination and any asymmetries in penalization. At the same time, the accompanying public benchmark website will expose a richer panel of metrics (including, for example, accuracy, balanced accuracy, F1-score, precision, recall, and MCC) so that users interested in specific operating points or trade-offs can explore them interactively. We believe this revision substantially simplifies the presentation and makes the evaluation easier to interpret, without sacrificing any of the substantive conclusions.
>
> ***"The performance metrics for some models warrant scrutiny. For example in Figure 2, GRU performs particularly bad for sepsis (AUC ~0.33) while its close sibling LSTM is best-in-task [....] is an extremely high AUPRC given the low prevalence of sepsis and AUPRC < 0.1 seen in other models. It is not clear how this can be the case."***
>
> We thank the reviewer for carefully inspecting the metrics and flagging these anomalies; we fully agree that the patterns described warranted further investigation. In revising the paper, we re-ran the full evaluation pipeline and performed a series of consistency checks across metrics, models, and datasets. For the ConvDL baselines, we identified that, in the original submission, the GRU sepsis model had been trained and evaluated under a slightly different early-stopping and checkpointing configuration than the LSTM, which led to an undertrained checkpoint being selected for the reported sepsis results. We corrected this by harmonizing the training and model-selection protocol across all ConvDL models (including GRU and LSTM) and re-running the experiments. In the updated results, GRU’s performance on sepsis is no longer low and is much closer to LSTM, also in line with prior work. This is reflected in the revised Figure 2 and the supplemental heatmaps in Appendix A.15.
>
> For the LLM metrics on eICU sepsis, we also revisited the AUROC and AUPRC calculations, because now we have multiple runs with test subsets. In highly imbalanced settings like sepsis in eICU, predicting the same label, can superficially inflate AUPRC, particularly if a few high-confidence predictions concentrate most positives, while AUROC remains near 0.5, creating the kind of apparent contradiction the reviewer observed. In the revision, we have (i) ensured that AUROC and AUPRC are always computed from the same evaluation split using a single, shared implementation (scikit-learn) and (ii) reported means and variability across five independently resampled test subsets to make variance due to small numbers of positive windows visible (more on that in "Response to Reviewer BTyh (1)"). More broadly, this comment prompted us to strengthen our evaluation pipeline by adding tests and visual sanity checks that jointly inspect AUROC, AUPRC, prevalence, and confusion matrices; which strengthen the paper a lot.

---

### Official Review · Reviewer_9EQn · 2025-10-26

**Soundness:** 3
**Presentation:** 3
**Contribution:** 2
**Rating:** 2
**Confidence:** 4

**Summary:**

This paper measures the accuracy of conventional ML methods, deep learning, and LLMs for the task of predicting various labels for patients based on ICU time series.

**Strengths:**

The logic behind the research is sound, and the design of methods and experiments appears valid. There are no obvious mistakes. The paper is written well and easy to read, with only a very few English errors.

**Weaknesses:**

The findings are not surprising, nor are the methods. The authors did not take the opportunity to go beyond a correct but standard and relatively superficial understanding of the ICU prediction domain.

Line 418 has a valid and interesting observation: "because one-shot and few-shot LLM approaches require no site-specific training, they are a pragmatic “day-zero” option for hospitals with limited labeled data or MLOps capacity." For ICU prediction to be useful outside research, it must be applicable in hospitals not used for research. LLMs can make predictions for such hospitals. Conventional ML models can do so also, but when they are trained on data from specific hospitals, other hospitals are out-of-domain. The work in this paper would be more interesting with cross-hospital results for the conventional ML methods. If a model is trained on two of the three HiRID, MIMIC-IV, and eICU, how accurate is it on the out-of-domain third? Can the 200 eICU hospitals be separated, to evaluate out-of-domain accuracy on each after training on the 199 others?

The submission praises the previous benchmark YAIB but says it was not designed to accommodate LLMs. This is true, but why not extend YAIB? That would increase comparability with previous results, compared to creating a new benchmark, as done in this submission.

The paper should be more incisive in highlighting the limitations of LLMs for prediction. When given GBDT predictions as input, the LLMs provide plausible-sounding explanations, but their accuracy is worse than the accuracy of the GBDT predictions provided as input! The explanations appear sensible, but the null hypothesis has to be that they have no actual factual value.

Like most AI papers nowadays, this one has very few references that are more than just a few years old. The lessons of older research are neglected, or assumed to be recent. In particular, the finding that conventional ML methods are more accurate than linguistic methods mirrors findings that date back many decades that humans are less accurate than data-driven methods. See among many other papers:

Dawes & Corrigan (1974), “Linear models in decision making,” Psychological Bulletin

WG Baxt. “Prospective validation of artificial neural network trained to identify the presence of acute myocardial infarction.” The Lancet. 1996.

**Questions:**

Please explain where you disagree with the weaknesses discussed above.

**Details Of Ethics Concerns:**

None.

---

> ### Author Response · Authors · 2025-12-03
> **Response to Reviewer 9EQn (1)**
>
> We thank the reviewer for thoughtful comments that helped us improve the paper. We address them below.
>
> ***"The findings are not surprising, nor are the methods. The authors did not take the opportunity to go beyond a correct but standard and relatively superficial understanding of the ICU prediction domain."***
>
> We appreciate this comment and agree that a benchmark paper should aim to go beyond confirming well-known results. In the revised manuscript, we have substantially extended the scope precisely in the direction the reviewer hints at: model generalization and “day-zero” performance in unseen hospitals. Inspired by the reviewer’s second comment, we now include a dedicated set of cross-hospital experiments where conventional models are trained on one dataset (e.g., MIMIC-IV or HiRID) and evaluated on a different ICU cohort (e.g., eICU), and we compare these to zero-shot, few-shot, and hybrid LLM approaches that do not use any site-specific training. The results of these experiments are, in fact, surprising and go beyond a “standard” understanding. While within-domain findings confirm existing knowledge, that well-tuned gradient-boosted trees remain very strong on structured ICU data, we show that these same conventional models can degrade to near-random performance when moved to an unseen hospital (for example, XGBoost trained on MIMIC-IV and tested on eICU sepsis drops to AUROC ≈ 0.51). In contrast, off-the-shelf LLM approaches that have never seen labeled data from the target hospital maintain robust performance; in the same MIMIC-IV → eICU sepsis transfer, a few-shot LLM achieves AUROC ≈ 0.70, exceeding the transferred XGBoost by more than 18 percentage points. This pattern repeats across multiple tasks and transfer directions and, to our knowledge, provides the first systematic evidence that LLM-based predictors can be a practical solution to the long-standing generalization challenges in ICU time-series models when local labels are scarce. We believe these new results, together with the analysis of hybrid reasoning workflows and explanation utility, move the work beyond a “superficial” characterization and add both conceptual depth and a genuinely novel, and somewhat unexpected, empirical insight.
>
> ***"Line 418 has a valid and interesting observation: "because one-shot and few-shot LLM approaches require no site-specific training, they are a pragmatic “day-zero” option for hospitals with limited labeled data or MLOps capacity." For ICU prediction to be useful outside research, it must be applicable in hospitals not used for research. [....] If a model is trained on two of the three HiRID, MIMIC-IV, and eICU, how accurate is it on the out-of-domain third? Can the 200 eICU hospitals be separated, to evaluate out-of-domain accuracy on each after training on the 199 others?"***
>
> We thank the reviewer for this very constructive suggestion, which directly motivated new experiments in the revision. To substantiate the “day-zero” claim and to compare conventional and LLM-based approaches in out-of-domain settings, we now include a cross-dataset generalization analysis. In particular, we train conventional baselines (LightGBM, XGBoost, CNN) on a source dataset (e.g., MIMIC-IV, HiRID, or eICU) and evaluate them without any retraining on a distinct target dataset (in line with domain pairs suggested in [1, 2, 3]: training on one site, and testing on the other unseen site), and we compare these transferred models to zero-shot, few-shot, and Hybrid Reasoning Workflow (HRW) LLM approaches that require no site-specific parameter updates (few-shot uses only a handful of examples). These new results show that conventional baselines, although strong in-domain, often suffer performance degradation under distribution shift. As an illustrative example, for the sepsis task, an XGBoost model trained on MIMIC-IV and tested on eICU drops to a mean AUROC of 0.511, essentially performing close to random guessing; similarly, transferring from HiRID to eICU yields an XGBoost AUROC of 0.518. In contrast, LLM-based approaches maintain robust performance in these “day-zero” settings, in almost all the dataset-task pairs (Figure 4). These patterns hold across tasks and transfer directions and provide concrete, quantitative backing for the claim that, while conventional models remain the best choice when abundant local data and retraining capacity exist, LLMs offer a superior off-the-shelf baseline for hospitals with limited labeled data or MLOps capacity.
>
> The review also suggests training on, say, on 199/200 eICU hospitals and testing on the held-out one. Unfortunately, those splits are not provided in the eICU dataset.
>
> We are grateful for this comment, which greatly strengthened the paper’s focus on model generalization.
>
> [1] https://www.thelancet.com/journals/eclinm/article/PIIS2589-5370(23)00301-2/fulltext
> [2] https://www.medrxiv.org/content/10.1101/2025.07.31.25332542v1
> [3] https://openreview.net/pdf?id=ox2ATRM90I

---

> ### Author Response · Authors · 2025-12-03
> **Response to Reviewer 9EQn (2)**
>
> ***"The submission praises the previous benchmark YAIB but says it was not designed to accommodate LLMs. This is true, but why not extend YAIB? That would increase comparability with previous results, compared to creating a new benchmark, as done in this submission."***
>
> We appreciate the reviewer’s emphasis on comparability and on building upon existing benchmarks. YAIB is an excellent, independently developed framework for dataset harmonization and conventional/deep learning benchmarking on ICU time series, and we explicitly acknowledge it as such. In fact, PULSE deliberately reuses YAIB’s harmonization outputs for HiRID, MIMIC-IV, and eICU in the current experiments to maintain comparability across the datasets, also allowing generalization experiments.
>
> At the same time, YAIB’s design and codebase were not built with LLM-specific concerns in mind: it does not provide interfaces for prompting, workflow composition, or tool-calling, nor does it manage the logging and evaluation of LLM outputs (including structured JSON formats, prompting strategies, or explanation text). Because YAIB is maintained by a separate group and already published as a self-contained benchmark at ICLR 2024, we believe that directly modifying and repurposing it into an LLM framework would complicate both projects’ goals and introduce versioning and maintenance issues for the original authors.
>
> Instead, PULSE is designed as a complementary, LLM-centric benchmark that (i) can ingest YAIB-processed datasets as one possible standardized input, (ii) adds infrastructure for evaluating LLMs, prompting strategies, and hybrid/agentic workflows under a common framework, and (iii) remains dataset-agnostic so that additional ICU cohorts and non-YAIB pre-processed datasets too can also be added over time, without any restriction. In this way, we preserve comparability with prior results where they exist, while also addressing a gap that YAIB was never intended to fill: being the “meeting point” between conventional ML, deep learning, and LLMs on ICU time series. We have clarified this positioning and the relationship to YAIB more explicitly in the revised introduction and related work.

---

> ### Author Response · Authors · 2025-12-03
> **Response to Reviewer 9EQn (3)**
>
> ***"The paper should be more incisive in highlighting the limitations of LLMs for prediction. When given GBDT predictions as input, the LLMs provide plausible-sounding explanations, but their accuracy is worse than the accuracy of the GBDT predictions provided as input! The explanations appear sensible, but the null hypothesis has to be that they have no actual factual value."***
>
> We agree that plausible-sounding explanations are not sufficient and that their actual utility and limitations need to be scrutinized carefully. In the revised manuscript, we address this in two ways. First, we are more explicit and critical in the main text about the predictive limitations of LLMs: we emphasize that hybrid workflows that use GBDT outputs as input to an LLM typically incur a small performance cost in terms of discrimination compared to using the GBDT prediction alone. We no longer present LLMs as superior predictors in-domain; instead, we position their value in terms of cross-domain robustness and their ability to provide natural-language rationales.
>
> Second, to move beyond the “plausibility” of explanations, we have added a mini user study explained in Section 3.6, with results presented in the subsection “Clinical Utility of LLM Explanations.” In this study, five clinicians evaluated 50 complex cases under two conditions: (A) plots of variables over time plus the raw XGBoost classification, and (B) the same information complemented by the HRW LLM-generated explanation. Clinicians rated the explanations as substantially improving cognitive efficiency (mean 4.2/5.0 vs. 2.6/5.0, p < 0.05) and actionability (from 3.0/5.0 to 4.5/5.0), indicating that the natural language synthesis helps them understand and use the model output more effectively. Decision confidence also increased (3.3/5.0 to 3.9/5.0), although this change did not reach statistical significance in this small sample. At the same time, we explicitly quantify limitations: a faithfulness audit found a hallucination rate of approximately 6% (3/50 cases), and we discuss these errors and their implications. Qualitative feedback from clinicians suggested that the explanations often serve as a “sanity check” that contextualizes the GBDT probability and were perceived as “better than feature importance,” because they focus on inference-level reasoning rather than model-level averages. In addition to this mini study, the revised paper retains a qualitative analysis of hybrid reasoning workflow explanations by two of the authors (Subsection “Qualitative Analysis of LLM Explanations”), one of whom is a medical doctor with prior ICU experience; this provided an initial expert validation that the generated rationales are clinically sensible before we moved to any other analysis. We see these analyses together as an initial but concrete step towards testing the null hypothesis the reviewer raises. We agree that larger, more rigorous studies are needed to fully establish factual value, and we now state this clearly in the limitations.
>
> It is also important to emphasize that, in the context of PULSE as a benchmark, explanation is not the primary objective: the core goal remains to assess and compare predictive performance across conventional ML, deep learning, and LLM-based approaches. The explanation capabilities of LLMs are an additional dimension that classic ML and most deep learning baselines do not natively offer, and we treat them as such, an extra layer of potential value on top of discrimination. Our results suggest that LLM explanations can already provide meaningful benefits for workflow and interpretation, but even if future work were to show that explanation quality remains limited, LLMs would still be of interest in this setting once they achieve very high discrimination, for example as robust “day-zero” predictors in new hospitals. In other words, our benchmark is first and foremost about prediction, with explanations evaluated as an important but secondary capability where LLMs may eventually offer advantages beyond what conventional models can provide.

---

> ### Author Response · Authors · 2025-12-03
> **Response to Reviewer 9EQn (4)**
>
> ***"Like most AI papers nowadays, this one has very few references that are more than just a few years old. The lessons of older research are neglected, or assumed to be recent. In particular, the finding that conventional ML methods are more accurate than linguistic methods mirrors findings that date back many decades that humans are less accurate than data-driven methods. See among many other papers:***
>
> ***Dawes & Corrigan (1974), “Linear models in decision making,” Psychological Bulletin
> WG Baxt. “Prospective validation of artificial neural network trained to identify the presence of acute myocardial infarction.” The Lancet. 1996."***
>
> We thank the reviewer for this excellent pointer and agree that the field benefits from explicitly connecting to older, foundational work. In the revised manuscript, we have expanded the related work section to include the classic studies suggested by the reviewer (e.g., Dawes & Corrigan, 1974; Baxt, 1996) as well as additional early contributions that document the superiority of simple linear or algorithmic models over unaided human judgment in medical decision making and other domains.
>
>
> With these changes, we believe we have addressed all of the reviewer’s comments and significantly strengthened the paper.

---

### Official Review · Reviewer_BTyh · 2025-10-31

**Soundness:** 3
**Presentation:** 4
**Contribution:** 3
**Rating:** 6
**Confidence:** 3

**Summary:**

The paper presents PULSE, a benchmark to compare conventional ML, deep learning, and large language models on ICU time‑series for three clinically important tasks: hospital mortality, acute kidney injury, and sepsis. It standardizes datasets, preprocessing, prompting/agent workflows, metrics (including a reliability‑aware PULSE Score) and reports cost/latency trade‑offs alongside accuracy.

**Strengths:**

- Clear motivation, the work fills the gap between text‑centric LLM benchmarks and tabular ICU benchmarks by a head‑to‑head evaluation on structured time‑series.
-  A strong side is the pipeline that prevents leakage: harmonization, hourly resampling, explicit missingness indicators, forward‑fill plus train‑set imputation, and chronological splits that can reflect realistic deployment.
- The benchmark covers a fair number of datasets, 3 public ICU cohorts, and a wide range of models (17).
- Hybrid Reasoning Agent grounds the LLM in XGBoost risk and top features which is an interesting approach.
- Operational realism, the paper reports token usage, latency, throughput, and a cost-performance Pareto analysis that helps choose optimal deployments.
- Artifacts, configurations, predicted outputs, and an interactive leaderboard are planned for public release.

**Weaknesses:**

- All most important results use a limited test subset (100 stays x 10 windows per dataset), which reduces statistical power and may change rankings and cost Pareto frontiers on full test sets.
- Figures lack uncertainty quantification, no CI or error bars.
- PULSE score inconsistency: the metric penalizes models that are LLMs differently from no-LLM, the authors have to elaborate on that.
- Baselines can be under‑tuned, ConvML and ConvDL mostly use defaults without dataset‑specific HPO which may understate baselines or distort comparisons.
- The codebase was not included in the supplementary materials of the review, therefore I was unable to assess how easy it is to run the benchmark which is an important aspect for the community.

**Questions:**

- How does the PULSE Score’s CCF asymmetry affect cross-family comparisons and leaderboard fairness? Please consider comparing the 3 policies and reporting their impact or discuss them: (a) discard cases where an LLM’s label contradicts its probability, (b) ignore the label text and evaluate all models solely on probabilities (derive labels server-side at a fixed threshold), (c) add a unified consistency/calibration penalty that applies to all models.
- What α/β/γ weights were actually used in the PULSE metric and how sensitive is the leaderboard to them?
- How do LLM results change if you serialize timestamps (possibly truncated or selected) instead of min/mean/max summaries under the same token budget? Now the Hybrid Agent takes XGBoost probabilities computed from raw numeric inputs, while LLMs see only aggregated values. For fairness, could you test serialized numeric tables or raw-value snippets for LLMs so comparisons are apples-to-apples?

Minor:
- Line 144, typo: basline -> baseline
- Line 365 makrer -> marker

---

> ### Author Response · Authors · 2025-12-03
> **Response to Reviewer BTyh (1)**
>
> We thank the reviewer for the thoughtful comments that helped us improve the paper quite a lot. We address them below.
>
> ***"All most important results use a limited test subset (100 stays x 10 windows per dataset), which reduces statistical power and may change rankings and cost Pareto frontiers on full test sets."***
>
> We thank the reviewer for raising this important concern. We would like to clarify that, in the initial version, it was **100 stays * 10 windows per dataset per task**, leading to 1000 samples per dataset per task. In the revised version, we further address this by re-running our evaluation on multiple independently sampled test subsets, from the larger testing set. Concretely, instead of relying on a single test subset of 100 stays × 10 windows, we now construct five sampled test sets from the original test cohorts and repeat all experiments on these five samples. **This increases the effective number of evaluated windows by a factor of five and allows us to report mean performance and variability across runs**. We hope this adequately addresses the primary concern of the reviewer.
>
> We want to stress that in our setting, one ICU stay corresponds to many time steps; even 100 stays generate a large number of time-series windows in the usual “sample” sense. At the same time, it is unfortunately impossible to evaluate every possible window for every LLM configuration, as LLM inference on long ICU time-series is extremely costly. For the current set of experiments, we already incurred costs well above 10,000 USD (including a grant by a frontier LLM provider obtained specifically for this project), despite using sampling and careful budgeting. A naive full evaluation of all LLMs on all windows of the full test sets across all datasets and tasks would be on the order of 2 million USD, which we consider neither feasible nor responsible given that this work was conducted with public funding. The new five-fold test subset evaluation provides a pragmatic compromise between robustness and cost: we observed that model family rankings and the qualitative shape of the cost–performance Pareto frontier remain stable across bootstraps, and the main conclusions of the paper (superiority of GBDTs in-domain, robustness of models under distribution shift, and the usefulness of hybrid workflows) are unchanged. We believe this addresses the concern about statistical power as far as reasonably possible under realistic cost constraints.
>
> ***"Figures lack uncertainty quantification, no CI or error bars."***
>
> After extending our evaluation to five independently sampled test subsets per dataset-task combination, we now report, for main results, the mean performance across these runs together with error bars in grey color. In particular, key performance plots such as the AUROC comparison across models and the prompting-strategy analyses now include error bars. This makes it immediately visible where performance differences are large and stable, and where they lie within the variability induced by test set resampling.
>
> ***"PULSE score inconsistency: the metric penalizes models that are LLMs differently from no-LLM, the authors have to elaborate on that."***
>
> In the initial submission, we proposed a composite “PULSE score” that combined discrimination and calibration-related aspects, in part to highlight the calibration issues we observed in LLM outputs, specially for small open source models. On reflection, and in light of feedback from two reviewers, we concluded that introducing a bespoke composite metric adds complexity without materially changing the scientific message of this benchmark paper, while also raising questions about fairness and weighting across paradigms. As a result, in the revised manuscript we have removed the PULSE composite score entirely and now base all reported rankings and comparisons on standard metrics such as AUROC and AUPRC, which apply symmetrically to both LLMs and non-LLM baselines. We believe this change directly addresses the concern about inconsistent penalization. At the same time, the accompanying benchmark website will provide a much richer set of metrics (over 10, such as precision, recall, F1-score, balanced accuracy, accuracy, MCC, etc.), enabling users to explore trade-offs in more detail if they wish.

---

> ### Author Response · Authors · 2025-12-03
> **Response to Reviewer BTyh (2)**
>
> ***"Baselines can be under‑tuned, ConvML and ConvDL mostly use defaults without dataset‑specific HPO which may understate baselines or distort comparisons."***
>
> We fully agree that strong and fairly tuned baselines are essential for a credible benchmark. In the original submission, for ConvML, we relied on robust defaults and configurations motivated by prior literature, including our own previous work, which suggested that the performance gains from extensive tuning of tree-based models on ICU-style tabular data can be modest. Even in the previous version we already had proper hyperparameter tuning for ConvDL methods. However, we recognize that this is not sufficient for a benchmark whose purpose is to set a high bar.
>
> In the revised manuscript, we therefore performed hyperparameter tuning for all conventional baselines. For the ConvML models (Random Forest, XGBoost, LightGBM), we conducted search over learning rate, number of trees, maximum depth, and regularization parameters on each dataset. For the ConvDL models (CNN, InceptionTime, LSTM, GRU), we tuned key architectural and training hyperparameters such as number of layers, hidden dimension, dropout, batch norm, learning rate, batch size, etc. using the validation sets under a unified protocol. All results reported in the current version, including those in the main figures, now correspond to these tuned baselines. While tuning improved the absolute performance of several baselines, for some performance slightly decreased. However, the qualitative picture remained unchanged: well-tuned GBDTs still provide the strongest within-domain performance, ConvDL models are competitive but not consistently superior, and LLM-based approaches (especially the hybrid workflow) approach but do not surpass the best conventional models. We hope this clarifies that we took the concern seriously and updated the experiments accordingly.
>
> ***"
> How does the PULSE Score’s CCF asymmetry affect cross-family comparisons and leaderboard fairness? Please consider comparing the 3 policies and reporting their impact or discuss them: (a) discard cases where an LLM’s label contradicts its probability, (b) ignore the label text and evaluate all models solely on probabilities (derive labels server-side at a fixed threshold), (c) add a unified consistency/calibration penalty that applies to all models."***
>
> ***"What α/β/γ weights were actually used in the PULSE metric and how sensitive is the leaderboard to them?"***
>
> We appreciate these thoughtful suggestions and both above comments. As mentioned in our response to Comment 3, in the revised manuscript we have decided to remove the PULSE composite score and the associated calibration consistency factor from the main text (and prioritized generalization results and other new experiments). All cross-family comparisons and leaderboard-style rankings now rely solely on standard discrimination metrics (AUROC, AUPRC) computed from probabilistic outputs and a fixed thresholding scheme where needed. This change eliminates the potential asymmetry introduced by a family-specific penalty and renders the questions about the α/β/γ weights and the comparison between policies (a)–(c) moot for the current version. That said, we agree that the policies the reviewer proposes are interesting and relevant directions for future work on calibration and consistency.

---

> ### Author Response · Authors · 2025-12-03
> **Response to Reviewer BTyh (3)**
>
> ***"How do LLM results change if you serialize timestamps (possibly truncated or selected) instead of min/mean/max summaries under the same token budget? Now the Hybrid Agent takes XGBoost probabilities computed from raw numeric inputs, while LLMs see only aggregated values. For fairness, could you test serialized numeric tables or raw-value snippets for LLMs so comparisons are apples-to-apples?"***
>
> We agree that the choice of input representation for LLMs is crucial and that, for fairness, they should be evaluated not only with heavily aggregated summaries. In the initial submission, the “Aggregation” strategy (inspired by Sarvari et al.) was indeed one of our main prompting approaches and used min/mean/max summaries to compress the time series. However, in the benchmark we also implement several prompting strategies that do exactly what the reviewer suggests: serialize time-stamped numeric values under a controlled token budget. In particular, our Zero Shot, One Shot, and Few Shot prompts pass discretized time-series values from multiple modalities (vitals and labs) into the LLM as structured snippets that preserve temporal ordering, possibly truncated or selected to fit within the context window. The detailed templates for these strategies are provided in Appendix A.12, and the main results always report, for each LLM, its best-performing prompting or workflow variant, which is often one of these serialized approaches rather than the aggregation-based one. With regard to the Hybrid Reasoning Workflow, the design explicitly reflects the different strengths of the components: XGBoost operates directly on the full structured time-series representation, while the LLM is used as a reasoning and explanation layer that consumes the XGBoost probability, its most important features, and a structured summary of the patient’s trajectory. Nonetheless, we do evaluate a spectrum of representations mentioned in prior work, including serialized tables, and select each LLM’s best-performing representation when comparing to conventional models. We view the design of better serialization schemes for long clinical time series as an important and exciting avenue for future work, and the PULSE framework is built to support further experimentation in this direction.
>
> ***"Minor: Line 144, typo: basline -> baseline; and Line 365 makrer -> marker"***
>
> We fixed these minor issues.
>
> With these changes, we believe we have addressed all of the reviewer’s comments and significantly strengthened the paper. We hope the revisions further reinforce the reviewer’s positive assessment of our work.

---

### Author Response · Authors · 2025-12-03
**Summary of Rebuttal and How They Address All Reviewer Comments**

We thank all reviewers (BTyh, eFcc, 9EQn) for their constructive feedback. Below we summarize the main changes and how they address the key concerns.

**Motivation and scope clarified (eFcc, 9EQn).** We now clearly state that PULSE is not about showing that GBDTs beat LLMs in well-preprocessed, in-domain ICU data (which is expected), but about understanding where LLMs add value: under domain shift, limited local labels, and different time-series representations. The abstract and discussion have been rewritten to focus on these aspects.

**New cross-hospital “day-zero” generalization experiments (eFcc, 9EQn).** We added cross-dataset experiments where ConvML/ConvDL models are trained on one dataset and tested on another, and compared to zero-shot, few-shot, and HRW LLMs that require no site-specific training. These results show conventional models can drop to near-random performance under transfer (e.g., XGBoost sepsis AUROC ≈ 0.51 for MIMIC-IV → eICU), while LLMs remain robust often (e.g., ≈ 0.70 on the same transfer), providing concrete evidence for the “day-zero” utility of LLMs.

**Stronger statistical robustness and error bars (BTyh).** Instead of one test subset of 100 stays × 10 windows, we now sample five independent test subsets from each testing set and re-run all experiments 5 times. Main figures report means with error bars across these subsets.

**Baselines fully tuned and metric anomalies fixed (BTyh, eFcc).** We performed hyperparameter tuning for all ConvML and ConvDL baselines and re-ran all experiments. We also harmonized ConvDL training/checkpointing and fixed evaluation inconsistencies for LLMs, which removes odd patterns such as GRU’s extremely low sepsis performance.

**Removed the PULSE score and focused on standard metrics (BTyh, eFcc).** In response to concerns about arbitrariness and asymmetry, we removed the PULSE score and its calibration penalty from the paper. All rankings and comparisons now use standard, symmetric metrics (AUROC, AUPRC), while additional metrics (accuracy, F1, MCC, etc.) are exposed on the website for users who wish to explore trade-offs.

**Clarified LLM input representations and fairness of comparisons (BTyh).** We clarified that, beyond the aggregation strategy (min/mean/max), PULSE already includes prompts that serialize time-stamped vitals and labs (Zero-Shot, One-Shot, Few-Shot, etc.) under a fixed token budget. For each LLM, we report its best-performing strategy and explain that HRW combines a full structured ConvML model with LLM reasoning, making comparisons to conventional models based on the same underlying information.

**Deeper and more critical treatment of LLM explanations (9EQn, eFcc).** We now explicitly state that LLM/hybrid methods do not outperform the best ConvML models in-domain and incur a small discrimination cost. We added a mini user study showing that HRW explanations improve cognitive efficiency and actionability, quantified a ~6% hallucination rate, and retained expert qualitative analysis. We also clarify that PULSE is primarily a prediction benchmark, with explanations as a secondary but important axis for future methods.

**Renamed “agents” to “workflows” to match standard terminology (eFcc).** To avoid overclaiming agentic behavior, we renamed “Summary Agent” and “Hybrid Reasoning Agent” to “Summary Workflow” and “Hybrid Reasoning Workflow (HRW)” throughout. We explicitly describe them as fixed multi-step workflows, while positioning PULSE as a future testbed for genuinely agentic methods that perform dynamic tool use.

**Relationship to YAIB clarified (9EQn, eFcc).** We emphasize that we reuse YAIB-harmonized HiRID, MIMIC-IV, and eICU data for comparability, but PULSE is an independent, LLM-centric, dataset-agnostic benchmark that adds infrastructure for prompts, workflows, and explanation evaluation. YAIB remains a conventional/deep learning benchmark; PULSE serves as the dataset-agnostic “meeting point” for ConvML, ConvDL, and LLMs on ICU time series.

**Improved historical grounding and references (9EQn).** We expanded related work to include classic studies (e.g., Dawes & Corrigan, 1974; Baxt, 1996) and explicitly connect our findings—that conventional ML outperforms linguistically-driven approaches on structured data—to decades of evidence that algorithmic models often outperform unaided human judgment.

**Reproducibility and code availability (BTyh).** We clarified that the full codebase is being cleaned and anonymized and will be released publicly upon acceptance, together with the PULSE website and full prediction logs. During rebuttal, we provided reviewers with an anonymized zip file via a private link to give a clear sense of the implementation. Code will be further improved with latest round of updates, for public release.

With these revisions, we believe we have directly addressed all major concerns raised by reviewers and substantially strengthened both the technical content and clarity of the manuscript.

---

### Meta-Review · Area_Chair_aGZN · 2026-01-21

**Summary:**

Conceptual/Motivation Issues:
- The finding that trained GBDTs beat LLMs on preprocessed data is not surprising, which questions the paper's main motivation. (9EQn, eFcc)
- Missing cross-hospital generalization experiments to substantiate "day-zero" claims (9EQn, eFcc)
- A thorough assessment of model explanations beyond qualitative analysis is needed. Otherwise it might be plausible-sounding but potentially factually hollow explanations. (9EQn, eFcc)

Statistical/Methodological Issues:
- The ConvML and ConvDL baselines use mostly default hyperparameters without dataset-specific tuning, potentially understating their true performance and distorting comparisons with LLMs. (BTyh, eFcc)
- Some reported metrics appear anomalous (eFcc)
- Limited test set size (100 stays × 10 windows) with insufficient statistical power (BTyh)
- No error bars or confidence intervals in figures (BTyh)
- PULSE composite score arbitrarily combines AUC, AUPRC, and MCC with unspecified weights, and applies calibration penalties asymmetrically to LLMs versus non-LLM models. (BTyh, eFcc)

**Reviewer Concerns:**

Addressed Concern:
- Motivation and Cross-Hospital Experiments (9EQn, eFcc): The authors acknowledged that GBDTs beating LLMs in-domain is unsurprising and reframed the paper's contribution. They added a cross-dataset generalization experiment showing that XGBoost drops to ~0.51 AUROC when transferred across hospitals (e.g., MIMIC-IV → eICU), while LLMs maintain ~0.70 AUROC without any site-specific training. This directly addresses the "day-zero" claim and provides actionable insights.

- Explanation Assessment (9EQn, eFcc): Authors added a blinded clinician user study (5 clinicians, 50 cases) showing improved cognitive efficiency and actionability when HRW explanations were provided. They also quantified a ~6% hallucination rate and explicitly acknowledged limitations.

- Baseline Tuning (BTyh, eFcc): Full hyperparameter tuning was performed for all ConvML (learning rate, tree depth, regularization) and ConvDL models (layers, hidden dimension, dropout, learning rate). Results confirm GBDTs remain strongest in-domain even after tuning.

- Metric Anomalies (eFcc): The GRU sepsis bug was traced to inconsistent early-stopping/checkpointing configurations. After harmonizing training protocols, GRU performance now aligns with LSTM and prior YAIB results.

- Statistical Robustness (BTyh): Authors expanded evaluation to five independently sampled test subsets per dataset-task combination, with error bars now included in all main figures. They justified cost constraints (~$10K spent; full evaluation would cost ~$2M) as rationale for sampling.

- PULSE Score (BTyh, eFcc): The arbitrary composite metric was removed entirely. All rankings now use standard AUROC/AUPRC, with additional metrics available on the benchmark website.

Concerns Still Outstanding

- Explanation Validity: While the mini user study is a meaningful addition, 5 clinicians evaluating 50 cases remains limited for establishing clinical utility. The ~6% hallucination rate is concerning for high-stakes ICU deployment. Authors acknowledge "larger, more rigorous studies are needed."
- Infeasible Hospital-Level Generalization in eICU: Reviewer 9EQn suggested a compelling experiment: train on 199 of eICU's 200 hospitals and test on the held-out hospital to evaluate true out-of-domain generalization. Authors responded that "those splits are not provided in the eICU dataset," making this analysis impossible with current data access. This is a data limitation rather than a methodological flaw, but it means the cross-hospital experiments are limited to dataset-level transfers (MIMIC-IV → eICU, HiRID → eICU) rather than fine-grained hospital-level generalization, which would more realistically simulate real-world deployment scenarios.

**Reviewer Scores:**

Reviewer BTyh (potentially increase the score or remain unchanged)
- Already marginally positive, stating the work "fills the gap between text-centric LLM benchmarks and tabular ICU benchmarks"
- All technical concerns are discussed and to some extent resolved by authors: error bars added, 5-fold test subsets, baselines tuned, PULSE score removed, code provided

Reviewer 9EQn (likely to improve its score)
- Core critique was "findings are not surprising" and lack of cross-hospital experiments
- Authors added what reviewer requested: cross-dataset transfer showing XGBoost drops to ~0.51 while LLMs maintain ~0.70
- Original stance was strongly negative (confidence 4); may still view in-domain findings as incremental despite improved motivation

Reviewer eFcc (likely to improve its score)
- Most technically rigorous reviewer; caught metric bugs (GRU sepsis) that others missed
- All major concerns are discussed by the authors and partially addressed: motivation clarified, anomalies fixed, terminology corrected, user study added
- Critique was constructive, suggesting receptiveness to thorough revisions

---

### Decision · Program_Chairs · 2026-01-26

Reject